# Nonlinear Feature Aggregation:
# Two Algorithms driven by Theory

## Abstract

Many real-world machine learning applications are characterized by a huge number of features, leading to computational and memory issues, as well as the risk of overfitting. Ideally, only relevant and non-redundant features should be considered to preserve the complete information of the original data and limit the dimensionality. Dimensionality reduction and feature selection are common preprocessing techniques addressing the challenge of efficiently dealing with high-dimensional data. Dimensionality reduction methods control the number of features in the dataset while preserving its structure and minimizing information loss. Feature selection aims to identify the most relevant features for a task, discarding the less informative ones. Previous works have proposed approaches that aggregate features depending on their correlation without discarding any of them and preserving their interpretability through aggregation with the mean. A limitation of methods based on correlation is the assumption of linearity in the relationship between features and target. In this paper, we relax such an assumption in two ways. First, we propose a bias-variance analysis for general models with additive Gaussian noise, leading to a dimensionality reduction algorithm (NonLinCFA) which aggregates non-linear transformations of features with a generic aggregation function. Then, we extend the approach assuming that a generalized (non-)linear model regulates the relationship between features and target. A deviance analysis leads to a second dimensionality reduction algorithm (GenLinCFA), applicable to a larger class of regression problems and classification settings. Finally, we test the algorithms on synthetic and real-world datasets, performing regression and classification tasks, showing competitive performances.

## 1 Introduction

*Dimensionality reduction* is an essential technique in *machine learning* (ML), employed to limit the number of features or dimensions of datasets. It has been successfully applied in a large variety of fields where high-dimensional data needs to be analyzed, classified, visualized, or interpreted. For instance, in computer vision and in natural language processing, data is often represented as high-dimensional vectors; in bioinformatics, high-throughput biological data like DNA sequences are often represented as high-dimensional data; in earth sciences, meteorological variables have high-dimensional measurements in different locations. In these contexts, datasets are often characterized by a large number of highly correlated features. Projecting them into a lower dimensional space is crucial to simplify data representation and enhance the performance of models, mitigating overfitting, and limiting the computational complexity. However, dimensionality reduction may lead to loss of information and interpretability.

Another method to reduce the dimension of the feature space is *feature selection*, which aims to identify the most relevant features from a dataset. The importance of feature selection lies in its ability to improve model performance, reduce computational resources, and simplify the model's interpretation. Its main desirable property compared to dimensionality reduction is interpretability, since the reduced features are simply a subset of the original ones. However, these techniques usually discard features that may be exploited to reduce the variance.

In Bonetti et al. (2023), the authors propose a dimensionality reduction approach that *aggregates* subsets of features with their mean, if this is convenient in terms of bias-variance tradeoff. In this way, the algorithm preserves the interpretability of the transformed features, while exploiting the information of each feature. Assuming linearity in the underlying data generation process and applying linear regression, the asymptotic analysis on which the algorithm is based on suggests aggregating two features with their mean if their correlation is *sufficiently large*. Compared to other dimensionality reduction techniques, the aggregation based on the mean preserves the interpretability of the newly generated feature, since the mean function is a transformation that a domain expert can understand without any additional explanation by ML experts (see Kovalerchuk et al. (2021) for the definition of interpretability in these terms).

**Contributions**  In this work, after introducing the notation and formulation of the problem and reviewing the main dimensionality reduction methods (Section 2), we design principled algorithms that generalize this approach in two ways (Section 3). First, we relax the assumption of linearity of the relationship between the features and the target. In Section 4, we propose a dimensionality reduction technique that still considers linear regression as a supervised learning technique, but we allow the inputs to be *generic non-linear transformations* of the original features and the aggregation function to be a *generic non-linear function* (instead of the mean as in Bonetti et al. (2023)). This way, we allow the designer to ponder the suitable balance between the interpretability of the result (e.g., preferring simple feature and aggregation function) and the incorporation of complex non-linear relationships. Second, we analyse *generalized non-linear models*, assuming the expected value of the target to be a generic transformation of the features mapped through the link function (Section 5). We extend the proposed algorithm in this context, which is particularly useful when the Gaussianity assumption of the noise model is unrealistic or in the case of classification problems. Finally, we apply the two algorithms to classification and regression datasets, showing competitive results w.r.t. state-of-the-art methods (Section 6). We conclude the paper in Section 7 summarising its contributions and possible future developments. The paper is accompanied by an appendix. In particular, Appendix A and Appendix C respectively contain proofs and technical results related to Section 4 and Section 5. Appendix B shows an additional bi-variate analysis for generalized linear models, which is the starting point of the analysis that leads to the algorithm presented in the main paper. Finally, Appendix D presents additional experiments and gives more details on the experiments discussed in the main paper.

## 2  Preliminaries

**Notation**    Given $N$ samples and $D$ features, let $\mathbf{X} \in \mathbb{R}^{N \times D}$ and $\mathbf{y} \in \mathbb{R}^N$ (resp. $\mathbf{y} \in \{0, 1\}^N$ for classification) be the feature matrix and target vector, where $P_{X,Y}$ denotes the joint probability distribution. The $j$-th row of the matrix $\mathbf{X}$ is denoted with $\mathbf{x}_j$, while its $i$-th element is denoted with $x_i$, and it is called *feature*. Similarly, $y_j$ is the $j$-th element of the target vector $\mathbf{y}$. $\sigma_a^2$, $cov(a, b)$, $\rho_{a,b}$ and $\hat{\sigma}_a^2$, $\hat{cov}(a, b)$, $\hat{\rho}_{a,b}$ denote the variance of a random variable $a$, its covariance, and correlation with the random variable $b$ and their estimators, respectively. Finally, $\mathbb{E}_a[f(a)]$ and $var_a(f(a))$ are the expected value and the variance of a function $f(\cdot)$ of the random variable $a$ w.r.t. its distribution.

**Data Generation Processes**  We consider three scenarios for describing the data generation processes.

- *Non-linear model*: we consider a general non-linear relationship $f$ between the features $\mathbf{x}$ and the target $y$, with additive Gaussian noise:

$$y = f(\mathbf{x}) + \epsilon, \qquad \epsilon \sim \mathcal{N}(0, \sigma^2). \tag{1}$$

- *Generalized linear model*: we assume the distribution of the target to be part of the canonical exponential family:

$$Y|X \sim \exp\left(\frac{y\boldsymbol{\theta} - b(\boldsymbol{\theta})}{\phi}\right) + c(y, \phi), \tag{2}$$

  where $\boldsymbol{\theta} = \boldsymbol{X}\mathbf{w}$ is the parameter of the family, $\phi$, $b(\cdot)$ and $c(\cdot)$ are respectively a known scale parameter, a function of the parameter $\boldsymbol{\theta}$ characterizing the assumed distribution and a normalizing

function independent from $\boldsymbol{\theta}$. In this setting, the expected value of the target is $\mathbb{E}_{\boldsymbol{\theta}}[y] = b'(\boldsymbol{\theta})$, where we denote with $b'(\cdot)$ the first derivative of the function $b(\cdot)$. Moreover, in generalized linear models, the expected value of the target given the features is a nonlinear transformation of the linear combination of their values:

$$\mu(\mathbf{x}) := \mathbb{E}[y|\mathbf{x}] = g^{-1}(\mathbf{x}^\mathsf{T}\mathbf{w}) \iff g(\mu(\mathbf{x})) = \mathbf{x}^\mathsf{T}\mathbf{w}, \tag{3}$$

where $g(\cdot)$ is the *canonical link function*. A complete analysis of generalized linear models can be found in (McCullagh, 2019).

- *Generalized non-linear model*: to further generalize the approach, we consider a generic non-linear transformation $f(\cdot)$ inside the link function ($\boldsymbol{\theta} = f(\boldsymbol{x})$), such that the parameter $\boldsymbol{\theta}$ is not constrained to be a linear transformation of the features:

$$\mu(\mathbf{x}) := \mathbb{E}[y|\mathbf{x}] = g^{-1}(f(\mathbf{x})) \iff g(\mu(\mathbf{x})) = f(\mathbf{x}). \tag{4}$$

**Dimensionality Reduction via Aggregation**  A dimensionality reduction algorithm maps the original $D$-dimensional feature matrix $\mathbf{X}$ into a reduced dataset $\mathbf{U} = \boldsymbol{\psi}(\mathbf{X}) \in \mathbb{R}^{N \times d}$ with $d < D$, being $\boldsymbol{\psi}$ a transformation function designed to maximize a specific performance measure. In this paper we consider $D$ fixed non-linear transformations of the original features $\phi_i(X) : \mathbb{R}^D \to \mathbb{R}, \forall i \in \{1, \ldots, D\}$ as inputs of the proposed dimensionality reduction approaches, known as *basis functions* (Bishop, 2006)[1]. Then, a generic aggregation function $h(\cdot) : \mathbb{R}^2 \to \mathbb{R}$ is considered, and we aggregate couples of inputs through the aggregation function $h(\phi_i(X), \phi_j(X))$.

The choice to consider both non-linear transformation of the original features $\phi_i(\cdot)$ as inputs and a generic user-defined aggregation function $h(\cdot)$ is made to produce theoretical results that hold in a general setting. Some considerations follow:

- In line with the work of (Bonetti et al., 2023), the main applicative interest is to consider the $D$ features $\{x_1, \ldots, x_D\}$ as inputs and the mean as aggregation function. This way, the original (continuous) features are aggregated via a simple transformation, preserving the interpretability of the reduced features. However, in some applications, a non-linear relationship between some features and the target may be known, and the proposed algorithms can handle this situation as well. In the experimental section we will provide an example of this with synthetic data, where a quadratic relationship between the features and the target holds, by considering quadratic inputs $\phi_i(X) = x_i^2$ and the mean as aggregation function $h$.

- The choice to firstly consider non-linear transformations of the features and then an aggregation function may be replaced by a unique function that aggregates the original features, non-linearly combining them (e.g., when a quadratic relationship holds it is possible to directly consider the sum of squares as unique transformation function $\boldsymbol{\psi}$). However, this choice is made to underline the two distinct meanings of these functions. Firstly, the non-linear transformations $\phi_i(\cdot)$ of the inputs are user-defined modifications of the original features that allow a linear regression model to handle non-linear relationships, and are not aimed to reduce the dimension. Then, the aggregation function $h(\cdot)$ is a characteristic of the proposed algorithms that is responsible to perform the dimensionality reduction via aggregation. Indeed, since the aggregation function $h(\cdot)$ is an input of the algorithms, they allow the user to decide the aggregation function that is more meaningful for a specific application, or to test multiple aggregation functions to select the best performing one.

In this context, we assume $\epsilon$ to be a noise signal, independent of $X$, and we denote with $\hat{w}_i$ and $\hat{y}$ the estimated coefficients and the predicted target. Finally, to simplify the computations, we assume each expected value to be zero: $\mathbb{E}[\phi_i(X)] = \mathbb{E}[Y] = \mathbb{E}[h(\boldsymbol{\phi}(X))] = \mathbb{E}[f(X)] = 0$.

---

[1]We consider a number of basis functions $D$ equal to the number of the original features to simplify the notation, however a general set of $M$ basis functions can be considered.

**Performance Indexes** The Mean Squared Error (*MSE*), that is usually adopted as loss measure in regression problems that assume Gaussianity, can be decomposed into three terms (bias-variance decomposition (Hastie et al., 2009)):

$$
\underbrace{\mathbb{E}_{x,y,\mathcal{T}}[(\mathcal{M}_{\mathcal{T}}(x) - y)^2]}_{\text{MSE}} = \underbrace{\mathbb{E}_{x,\mathcal{T}}[(\mathcal{M}_{\mathcal{T}}(x) - \bar{\mathcal{M}}(x))^2]}_{\text{variance}}
$$
$$
+ \underbrace{\mathbb{E}_x[(\bar{\mathcal{M}}(x) - \bar{y}(x))^2]}_{\text{bias}} + \underbrace{\mathbb{E}_{x,y}[(\bar{y}(x) - y)^2]}_{\text{noise}},
\tag{5}
$$

with $x, y$ features and target of a test sample, $\mathcal{T}$ training set, $\mathcal{M}_{\mathcal{T}}(\cdot)$ ML model trained with $\mathcal{T}$, $\bar{\mathcal{M}}(\cdot)$ its expected value w.r.t. $\mathcal{T}$ and $\bar{y}$ expected value of the test target $y$ w.r.t. the input features $x$. We will consider the *MSE* to guarantee that the advantage in terms of reduction of variance (due to the dimensionality reduction) is larger than the disadvantage in terms of increase of bias.

The *deviance* is a measure of goodness of fit, usually considered in generalized linear models to extend the *MSE* to non-Gaussian settings (McCullagh, 2019). Recalling that in our setting $\boldsymbol{\theta} = f(\boldsymbol{X})$ (or $\boldsymbol{\theta} = \boldsymbol{X}\mathbf{w}$ assuming linearity), the deviance measures how the likelihood of the estimated model deviates from the best one:

$$
D^*(\boldsymbol{\theta}, \hat{\boldsymbol{\theta}}) := \frac{D(\boldsymbol{\theta}, \hat{\boldsymbol{\theta}})}{\phi} = -2\left[\ell(\hat{\boldsymbol{\theta}}) - \ell(\boldsymbol{\theta})\right] = \frac{2}{\phi}[y(\boldsymbol{\theta} - \hat{\boldsymbol{\theta}}) - (b(\boldsymbol{\theta}) - b(\hat{\boldsymbol{\theta}}))],
\tag{6}
$$

where $D$ is the deviance, $D^*$ is the scaled deviance, $\ell(\boldsymbol{\theta}), \ell(\hat{\boldsymbol{\theta}})$ are the log-likelihood of the model and of the estimated one, $\phi$ and $b(\cdot)$ are the known scale parameter and specific function of the distribution assumed. The last equation holds only if the distribution of the target belongs to the *canonical* exponential family, which will be assumed throughout the paper. When dealing with generalized (non-)linear models we will therefore consider the expected value of the scaled deviance to compare two different models. Performing dimensionality reduction, the deviance decreases when the reduction of variance due to the aggregation is convenient w.r.t. the loss of information.

## 2.1 Dimensionality Reduction Methods

This section contains a literature survey on dimensionality reduction. More extensive surveys can be found in (Van Der Maaten et al., 2009; Sorzano et al., 2014; Ayesha et al., 2020) and applicative results in (Zebari et al., 2020; Espadoto et al., 2021).

**Linear Dimensionality Reduction** Principal Components Analysis (PCA) (Pearson, 1901; Hotelling, 1933) is a popular linear dimensionality reduction method that embeds high dimensional data into a linear subspace, trying to preserve the variance of the original dataset. This method performs linear projections of possibly all the original features with different coefficients: interpretability can be difficult and it may suffer from the curse of dimensionality. Several linear dimensionality reduction approaches have been proposed to overcome the issues of PCA or to perform particular projections. Among these, svPCA (Ulfarsson & Solo, 2011) forces most of the weights of the projection to be zero, improving the interpretability but discarding the contribution of many features. Similar to feature selection, this does not lead to a loss of information in sparse environments but it may result in a loss of information when all features are informative. Independent Component Analysis (ICA) (Hyvarinen, 1999) is an information-theoretic approach that looks for independent components designed for multichannel data. Locality Preserving Projections (LPP) (He & Niyogi, 2003) is an unsupervised linear method that tries to preserve the local structure of the original data. Linear Discriminant Analysis (LDA) (Fisher, 1936) is a supervised technique that finds a linear combination of features that identifies the projection that separates the target classes. A broader overview of linear dimensionality reduction techniques can be found in (Cunningham & Ghahramani, 2015) and applicative results of linear methods in (Reddy et al., 2020).

**Non-linear Dimensionality Reduction** The limits of linear projections have been overcome by resorting to non-linear methods. Kernel PCA (Shawe-Taylor & Cristianini, 2004) is a variation of PCA that allows non-linearity by combining PCA with a kernel. Sammon Mapping (Sammon, 1969) is an unsupervised algorithm

that maps the data trying to preserve pairwise distance among data globally, similarly to Multidimensional scaling (Kruskal & Wish, 1978). Isomap (Tenenbaum et al., 2000) is an extension of MDS aimed to preserve the geodesic distance among features. Locally Linear Embedding (LLE) (Roweis & Saul, 2000) is a method aimed at preserving the distance among data, focusing on local similarity. More recently, many applications have embedded the dimensionality reduction phase in the learning process of a neural network, typically applying convolutional neural networks or autoencoders (Hinton & Salakhutdinov, 2006; Zebari et al., 2020). Among supervised methods (Chao et al., 2019), Supervised PCA (Bair et al., 2006) projects the data onto a subspace where the features are uncorrelated, simultaneously maximizing the dependency between the reduced dataset and the target. NMF-based algorithms (Jing et al., 2012; Lu et al., 2017) focus on the non-negativity property of features, which is not a general property of applicative problems. Finally, manifold-based methods perform supervised non-linear projections, usually adjusting an unsupervised approach to take into consideration the target (Neighborhood Components Analysis (NCA) (Goldberger et al., 2004), supervised Isomap (Ribeiro et al., 2008; Zhang et al., 2018) and supervised LLE (Zhang, 2009) are examples of these approaches). Additionally, similarly to the algorithms proposed in this paper, some dimensionality reduction algorithms optimize general non-linear dependency measures such as HSIC (Fukumizu et al., 2004; Masaeli et al., 2010; Wu et al., 2019; Barshan et al., 2011) or mutual information (Suzuki & Sugiyama, 2010), rather than simple correlation. However, in the most general case, these approaches produce a set of reduced features that are a combination of possibly all the original features, with different coefficients. This is in contrast with the main motivation of the work in (Bonetti et al., 2023) and its extensions provided in this paper. Indeed, the focus of the proposed algorithms is to identify a partition of the original features to aggregate with a user-defined function, which preserves the interpretability, for example in the sense of (Kovalerchuk et al., 2021).

**Group Regularization Methods**  We conclude the literature overview provided in this section by discussing feature grouping algorithms ((Park et al., 2007; She, 2008; Bondell & Reich, 2008; Kamkar et al., 2016)) in the context of sparse learning. These methods identify groups of variables and assign the same coefficient to variables belonging to the same group, which is similar to aggregating them with their mean. Additionally, some theoretical guarantees resemble the ones proposed in (Bonetti et al., 2023). However, these approaches are designed in a regularization setting, where the main motivation is to regularize a sparse linear problem by adding a penalty on the number of non-zero coefficients and another penalty term that focuses on assigning similar weights to groups of features, to further regularize the supervised learning task. This way, these group regularization methods are designed as multi-objective algorithms that simultaneously try to optimize the regression error, the coefficient sparsity and their similarity. On the contrary, the algorithm proposed in (Bonetti et al., 2023) is specifically designed for dimensionality reduction through feature aggregation, with the final purpose of obtaining a set of reduced features that preserves the majority of the information on the target, without discarding the contribution of each feature but reducing their dimensions in an interpretable way (the mean). In this context, this method is specifically focused in a single task (aggregating two features) providing some guarantees on the mean squared error after the aggregation, with no explicit regularization involved.

## 3  Proposed Algorithms: a Methodological Perspective

**Non-Linear Correlated Features Aggregation** Starting from the dimensionality reduction algorithm proposed in Bonetti et al. (2023), we introduce a similar approach relaxing the linearity assumptions, named *Non-Linear Correlated Features Aggregation (NonLinCFA)*. Let the relationship between the $D$ features $x_i$ and the target $y$ be non-linear ($y = f(\boldsymbol{x})$), with additive Gaussian noise (Equation 1). We iteratively compare, in terms of Mean Squared Error (MSE), two linear regression models. Given two non-linear transformations of the original features $\phi_1(\boldsymbol{x}), \phi_2(\boldsymbol{x}) : \mathbb{R}^D \to \mathbb{R}$, that allow to account for non-linearities in the linear regression, the first model considers them as separate inputs, while the second aggregates them with a generic aggregation function $h(\cdot) : \mathbb{R}^2 \to \mathbb{R}$:

$$\begin{cases} \hat{y} = \hat{w}_1 \phi_1(\mathbf{x}) + \hat{w}_2 \phi_2(\mathbf{x}) \\ \hat{y} = \hat{w} h(\phi_1(\mathbf{x}), \phi_2(\mathbf{x})) = \hat{w} h(\phi(\mathbf{x})), \end{cases} \tag{7}$$

where all the estimated coefficients $\hat{w}, \hat{w}_1, \hat{w}_2$ are least squares estimators. Intuitively, the second model has similar or better performances if the correlation between each input $\phi_1(\boldsymbol{x}), \phi_2(\boldsymbol{x})$ and the function $f(\boldsymbol{x})$ that generates the target $(\rho_{\phi_1,f}, \rho_{\phi_2,f})$ is not much larger than the one between their aggregation $h(\phi_1(\boldsymbol{x}), \phi_2(\boldsymbol{x}))$ and $f(\boldsymbol{x})$ $(\rho_{h(\phi_1,\phi_2),f})$. In this case, it is convenient to reduce the dimensionality of the inputs by aggregating them through the function $h(\cdot)$.

**Remark 1** (About linear features $\phi$)**.** *Considering the case $\phi_1(x) = x_i, \phi_2(x) = x_j, h(\phi_1, \phi_2) = \frac{\phi_1 + \phi_2}{2}$, the algorithm aggregates the two variables $x_i, x_j$ with their means if it is convenient in terms of MSE. In this case, the performance is preserved, and the aggregation keeps the reduced feature interpretable. Generally, the algorithm allows any zero-mean non-linear transformation of the input features and aggregation function, to balance the complexity and interpretability of the dimensionality reduction to be adapted to the specific problem.*

**Remark 2** (About considering two features at a time)**.** *Considering two inputs at a time can be convenient for huge-dimensional input spaces. The proposed algorithm iteratively compares couples of inputs, identifies a couple to aggregate and considers that aggregation as a single feature for the subsequent iterations.*

**Generalized-Linear Correlated Features Aggregation** The second algorithm we propose, *Generalized-Linear Correlated Features Aggregation (GenLinCFA)*, can be applied to problems where the target does not follow Gaussian distributions, including some classification problems. We assume the target to follow a distribution in the canonical exponential family, and we consider generalized linear models with canonical link function. Firstly, we compare the expected deviance of two models in the bivariate setting. Assuming that the expected value of the target is a linear combination of the two features, transformed by the canonical link $(\mathbb{E}[y|\mathbf{x}] = g^{-1}(w_1 x_1 + w_2 x_2))$, the first model considers the two features $x_1, x_2$ separately, while the second model considers their aggregation with the mean $\frac{x_1 + x_2}{2}$:

$$\begin{cases} \hat{y} = g^{-1}(\hat{w_1} x_1 + \hat{w_2} x_2) \\ \hat{y} = g^{-1}(\hat{w} \cdot \frac{x_1 + x_2}{2}), \end{cases} \tag{8}$$

where $\hat{w}_1, \hat{w}_2, \hat{w}$ are the coefficients estimated from data by the two models.

Then, we generalize this approach assuming a non-linear relationship between $D$ features and the expected value of the target $(\mathbb{E}[y|\mathbf{x}] = g^{-1}(f(\mathbf{x})))$. We analyse again the performances of two models, in terms of expected deviance:

$$\begin{cases} \hat{y} = g^{-1}(\hat{w_1}\phi_1(x) + \hat{w_2}\phi_2(x)) \\ \hat{y} = g^{-1}(\hat{w} \cdot h(\phi_1(x), \phi_2(x))). \end{cases} \tag{9}$$

As discussed for the first algorithm, we consider two nonlinear transformations $\phi_1(\cdot), \phi_2(\cdot)$ of the features as inputs and we compare a model that considers them separately with a model that aggregates them through a nonlinear function $h(\cdot)$. Again, the algorithm allows choosing different non-linear transformations of features and aggregation depending on the problem, and it iteratively compares couples of inputs.

## 4 Non-Linear Correlated Features Aggregation

This section describes the first proposed algorithm, *NonLinCFA*, designed to aggregate features in regression problems. We consider a problem with $D$ features and a non-linear relationship between the features and the target (Equation 1).

Given a training dataset with $N$ samples, $\mathbf{X} \in \mathbb{R}^{N \times D}$, we compare the *MSE* of the two models described in Equation 7. As discussed in Section 3, we, therefore, compare a bivariate with a univariate linear regression, where the two nonlinear functions of features $\phi_1(x), \phi_2(x)$ are aggregated through a function $h(\cdot)$. The variance decreases due to the reduction of the dimension of the hypothesis space and the bias increases due to the aggregation. Therefore, we analyse these two quantities, that together with the irreducible error compose the *MSE* (Equation 5), to guarantee that it does not increase significantly after the aggregation.

## 4.1 Theoretical Analysis

We firstly show the decrease of variance due to the aggregation in the asymptotic case. The asymptotic analysis is performed to identify an expression of the increase of variance that is manageable, without considering confidence intervals of each statistical estimator, as the sample size increases. The finite sample result and the proofs can be found in Appendix A.

**Theorem 1.** *Let the relationship between the features and the target be nonlinear with additive Gaussian noise (Equation 1). Let also each estimator to converge in probability to the quantity that it estimates. In the asymptotic case, let $\Delta_{var}^{n \to \infty}$ be the decrease of variance between a bivariate linear regression and the univariate case where the two features are aggregated (Equation 7), it is equal to:*

$$\Delta_{var}^{n \to \infty} = \frac{\sigma^2}{(n-1)}. \tag{10}$$

**Remark 3.** *The asymptotic result of Equation 10 follows the intuition that there is more variability in the prediction of a bivariate regression than in the univariate case if there is a small number of samples $n$ or a high variability of the noise $\sigma^2$.*

As a second result, we show the asymptotic increase of bias due to aggregation. A finite-sample result and related proofs are reported in Appendix A.

**Theorem 2.** *Let the relationship between features and target be nonlinear with additive Gaussian noise (Equation 1) and each estimator converge in probability. The asymptotic increase of bias $\Delta_{bias}^{n \to \infty}$ between bivariate and univariate linear regression, with the two features aggregated with a function $h(\cdot)$ (Equation 7), is equal to:*

$$\begin{aligned}
\Delta_{bias}^{n \to \infty} = &-\frac{cov(f, h(\phi_1, \phi_2))^2}{\sigma^2_{h(\phi_1,\phi_2)}} \\
&+ \frac{\sigma^2_{\phi_1} cov(\phi_2, f)^2 + \sigma^2_{\phi_2} cov(\phi_1, f)^2 - 2cov(\phi_1, f)cov(\phi_2, f)cov(\phi_1, \phi_2)}{\sigma^2_{\phi_1}\sigma^2_{\phi_2} - cov(\phi_1, \phi_2)^2}.
\end{aligned} \tag{11}$$

**Remark 4.** *Intuitively, the asymptotic result of Equation 11 suggests that it is convenient to aggregate the two features $\phi_1(\boldsymbol{x}), \phi_2(\boldsymbol{x})$ if there is large covariance $(cov(f, h(\phi_1, \phi_2)))$ between their aggregation and the target $f(\boldsymbol{x})$, or the covariance between the two features and the target $(cov(\phi_1, f), cov(\phi_2, f))$ is small, considering them singularly.*

We can now introduce the main theoretical result of this section: the asymptotic guarantee that ensures the *MSE* to not worsen after the aggregation.

**Theorem 3.** *Let the relationship between features and target be nonlinear with additive Gaussian noise (Equation 1) and each estimator converge in probability. The asymptotic* MSE *of a bivariate linear regression is not greater than the univariate case with the two inputs $\phi_1(\boldsymbol{x}), \phi_2(\boldsymbol{x})$ aggregated with a function $h(\cdot)$ (Equation 7) if and only if:*

$$\frac{\sigma^2}{\sigma_f^2(n-1)} \geq \frac{\rho^2_{\phi_1,f} + \rho^2_{\phi_2,f} - 2\rho_{\phi_1,f}\rho_{\phi_2,f}\rho_{\phi_1,\phi_2}}{1 - \rho^2_{\phi_1,\phi_2}} - \rho^2_{f,h(\phi_1,\phi_2)} = R^2_{f,\phi_1\phi_2} - R^2_{f,h(\phi_1,\phi_2)}. \tag{12}$$

*Proof.* The result follows imposing $\Delta_{var}^{n \to \infty} \geq \Delta_{bias}^{n \to \infty}$ from Equation 10 and 11. □

**Remark 5.** *Intuitively, the aggregation is convenient if:*

- *there is much noise in the model or a small number of samples (first term);*

- *real model and aggregated feature share a lot of information (third term);*

- *the two features do not share much information with the real model (small second term). Indeed, $R^2_{f,\phi_1\phi_2}$ is the coefficient of multiple correlation, indicating how well the target can be linearly predicted with a set of features (Keith, 2019).*

**Remark 6.** *In the bivariate linear case, where* $\phi_1 = x_1$, $\phi_2 = x_2$, $h(\phi_1, \phi_2) = \frac{x_1 + x_2}{2}$, $f(x) = w_1 x_1 + w_2 x_2$, *Equation 7 becomes* $\rho_{x_1, x_2} \geq 1 - \frac{2\sigma^2}{(n-1)(w_1 - w_2)^2}$. *This is in line with the result found in Bonetti et al. (2023) with linearity assumptions.*

**Remark 7.** *Assuming unitary variances* $\sigma_{\phi_1}^2 = \sigma_{\phi_2}^2 = 1$ *and uncorrelated combinations of features* $\rho_{\phi_1, \phi_2} = 0$, *the right hand side of Equation 12 becomes* $\rho_{\phi_1, f}^2 + \rho_{\phi_2, f}^2 - \rho_{f, h(\phi_1, \phi_2)}^2$. *This follows the intuition that the correlation between the real model and the aggregation should not be worse than the correlation between the real model and each individual transformation of features* $\phi_i(\boldsymbol{x})$.

### 4.2 Proposed algorithm: NonLinCFA

Starting from the result of Equation 12, this section introduces the algorithm proposed in this paper to perform dimensionality reduction in regression settings. The algorithm focuses on identifying if it is possible to add one feature in an aggregation, iteratively comparing a bivariate with a univariate approach. Algorithm 1 shows the pseudo-code of the proposed algorithm *Non-Linear Correlated Features Aggregation* (NonLinCFA). The main function of the algorithm (*NonLinCFA*) generates $d$ partitions of the $D$ inputs $\phi_1, \ldots, \phi_D$: at each iteration, it calls an auxiliary function (*compute_threshold*) to evaluate the performance in terms of coefficient of determination (*R2score*) of two models.

Firstly, the bivariate model is composed aggregating inputs already assigned to the current partition $h(\phi_{\mathcal{P}})$ and a feature not already assigned to any partition $\phi_j$. Then, the univariate linear regression considers the aggregation of the already selected inputs and the one under analysis ($h(\phi_{\mathcal{P}}, \phi_j)$) as a single feature. Their difference is an estimate of the right hand side of Equation 12: if it is *sufficiently small*, the algorithm adds the input $\phi_j$ to the current cluster. The hyperparameter $\epsilon$ regulates the propensity of the algorithm to aggregate features: when is large the algorithm is prone to aggregate, while small values are more conservative. This hyperparameter substitutes the left hand side term of Equation 12, which is difficult to be estimated in practice since it depends on the variance of the real model. Finally, the algorithm computes the aggregations of each element of the identified partition $\mathcal{P}$, returning the set of aggregated features $\{\bar{h}_\phi^1, \ldots, \bar{h}_\phi^d\}$.

As discussed in the previous sections, a particularly meaningful version of the algorithm can be obtained considering as inputs the $D$ features $\{x_1, \ldots, x_D\}$ and the mean as aggregation. This way, the algorithm identifies groups of features to aggregate with their mean, preserving the interpretability.

## 5 Generalized-Linear Correlated Features Aggregation

This section describes a second algorithm, *GenLinCFA*, that relaxes the Gaussianity assumption. We consider $D$ features and we assume the model in the canonical exponential family (Equation 2). The expected value of the target is a general function of the features $f(\boldsymbol{x})$, transformed by the linking function $g(\cdot)$ (Equation 4). In a first analysis we assume the function $f(\cdot)$ to be linear, modeling the expected value of the target as in Equation 3, and we compare the bivariate case with separate features and the univariate case with their aggregation (Equation 8). The bivariate analysis and some additional results are shown in Appendix B. As a second more general approach, given a training dataset with $N$ samples $\mathbf{X} \in \mathbb{R}^{N \times D}$, we compare the *expected deviance* of the two models described in Equation 9. We therefore compare a bivariate with a univariate regression, where two nonlinear functions of features $\phi_1(x), \phi_2(x)$ are aggregated through a function $h(\cdot)$. The main difference is the additional nonlinear transformation of the predicted linear model through the linking function $g(\cdot)$ (Equation 9), which characterizes the generalized non-linear model under analysis.

### 5.1 Theoretical analysis

Starting from Equation 6, given two estimators $\hat{\boldsymbol{\theta}}, \bar{\boldsymbol{\theta}}$ of the parameter $\boldsymbol{\theta}$, the increase of expected deviance between the two models is:

$$\mathbb{E}_{\mathbf{x}, y, \mathcal{T}}[D^*(\boldsymbol{\theta}, \hat{\boldsymbol{\theta}}) - D^*(\boldsymbol{\theta}, \bar{\boldsymbol{\theta}})] = \frac{2}{\phi} \mathbb{E}_{\mathbf{x}, y, \mathcal{T}}[y(\bar{\boldsymbol{\theta}} - \hat{\boldsymbol{\theta}}) - (b(\bar{\boldsymbol{\theta}}) - b(\hat{\boldsymbol{\theta}}))]. \tag{13}$$

Considering the two models under analysis (Equation 9), recalling that we defined $\boldsymbol{\theta} = f(\mathbf{x})$ and that the two estimators of $\boldsymbol{\theta}$ in the two cases are $\hat{\boldsymbol{\theta}} = \hat{w} h(\phi_1, \phi_2)$ and $\bar{\boldsymbol{\theta}} = \hat{w}_1 \phi_1 - \hat{w}_2 \phi_2$, the expected increase of

---

**Algorithm 1** NonLinCFA: Non-Linear Correlated Features Aggregation

---

**Input:** $D$ combinations of features $\{\phi_1(x), \ldots, \phi_D(x)\}$; target $y$; $N$ samples, aggregation function $h(\cdot)$, tolerance $\epsilon$

**Output:** reduced features $\{\bar{h}_\phi^1, \ldots, \bar{h}_\phi^d\}$, with $d \leq D$

    **function** COMPUTE_THRESHOLD($h, \phi_P, \phi_j, y, \epsilon$)             ▷ Threshold from Equation 12
        $R1 \leftarrow \text{R2score}(h(\phi_{\mathcal{P}}), \phi_j, y)$
        $R2 \leftarrow \text{R2score}(h(\phi_{\mathcal{P}}, \phi_j), y)$
    **return** $R2 - R1 - \epsilon$
    **end function**

    **function** NONLINCFA(*Input*)                                ▷ Main function
        $\mathcal{P} \leftarrow \{\}$                                 ▷ Partition of the features
        $\mathcal{V} \leftarrow \{\}$                     ▷ Set of already considered features
        **for each** $i \in \{1, \ldots, D\}$ **do**
            **if** $i \notin \mathcal{V}$ **then**
                $\mathcal{P} \leftarrow \{i\}$
                $\mathcal{V} \leftarrow \mathcal{V} \cup \{i\}$
                **for each** $j \in \{i+1, \ldots, D\}$ **do**
                    $threshold \leftarrow$ COMPUTE_THRESHOLD($h, \phi_P, \phi_j, y, \epsilon$)
                    **if** $threshold <= 0$ **then**             ▷ Aggregate the features
                        $\mathcal{P} \leftarrow \mathcal{P} \cup \{j\}$
                        $\mathcal{V} \leftarrow \mathcal{V} \cup \{j\}$
                    **end if**
                **end for**
                $\boldsymbol{\mathcal{P}} \leftarrow \boldsymbol{\mathcal{P}} \cup \{\mathcal{P}\}$
            **end if**
        **end for**
        $d \leftarrow |\boldsymbol{\mathcal{P}}|$
        **for each** $k \in \{1, \ldots, d\}$ **do**
            $\bar{h}_\phi^k = h(\phi_{\mathcal{P}_k})$
        **end for**
    **return** $\{\bar{h}_\phi^1, \ldots, \bar{h}_\phi^d\}$
    **end function**

---

deviance becomes:

$$\frac{2}{\phi}\Big\{\mathbb{E}_{\mathbf{x},y}\left[y \cdot (\mathbb{E}_{\mathcal{T}}[\hat{w}_1]\phi_1 + \mathbb{E}_{\mathcal{T}}[\hat{w}_2]\phi_2 - \mathbb{E}_{\mathcal{T}}[\hat{w}]h(\phi_1,\phi_2))\right]$$

$$- \mathbb{E}_{\mathbf{x},y,\mathcal{T}}\left[b\left(\hat{w}_1\phi_1 + \hat{w}_2\phi_2\right) - b(\hat{w}h(\phi_1,\phi_2))\right]\Big\}. \tag{14}$$

The following theorem provides the main theoretical result of this setting: a second-order approximated upper bound of the quantity under analysis, whose derivation can be found in Appendix C.

**Theorem 4.** *Let $M, m$ be the largest and smallest absolute value of the following expected values of coefficients: $\mathbb{E}_{\mathcal{T}}[\hat{w}_1]$, $\mathbb{E}_{\mathcal{T}}[\hat{w}_2]$, $\mathbb{E}_{\mathcal{T}}[\hat{w}]$, $\mathbb{E}_{\mathcal{T}}[\hat{w}_1^2]$, $\mathbb{E}_{\mathcal{T}}[\hat{w}_2^2]$, $\mathbb{E}_{\mathcal{T}}[\hat{w}_1\hat{w}_2]$, $\mathbb{E}_{\mathcal{T}}[\hat{w}^2]$. Let the real model belong to the canonical exponential family, denoting with $b''(\cdot)$ the second derivative of the function $b(\cdot)$ that characterizes the distribution (Equation 2). Considering a second order approximation, the expected increase of deviance $\Delta(D^*)$ due to the aggregation of the two transformed features $\phi_1(x), \phi_2(x)$ through an aggregation function $h(\cdot)$ is bounded by:*

$$\Delta(D^*) \lessapprox \frac{2}{\phi}\Bigg\{\left[M \cdot (|\operatorname{cov}(\phi_1, y)| + |\operatorname{cov}(\phi_2, y)|) - m \cdot |\operatorname{cov}(h(\phi_1, \phi_2), y)|\right]$$

$$- \frac{1}{2}b''(0) \cdot \left(m \cdot \sigma_{\phi_1+\phi_2}^2 - M \cdot \sigma_{h(\phi_1,\phi_2)}^2\right)\Bigg\}. \tag{15}$$

**Remark 8.** *Theorem 4 follows the intuition that it is convenient to aggregate two inputs when the variance of their sum is large or the variance of their aggregation $h(\phi_1, \phi_2)$ is small. Moreover, it is convenient to aggregate when the absolute value of the covariance between each feature and the target is small or the absolute value between the aggregated feature and the target is large.*

**Remark 9.** *Assuming a Gaussian distribution of the target given the features, the asymptotic increase of the expected deviance is equal to the asymptotic increase of* MSE *due to the aggregation. The proof of this considerations and additional technical results can be found in Appendix C.*

## 5.2 Proposed algorithm: GenLinCFA

Following the theoretical analysis performed, the second algorithm proposed in this paper is *GenLinCFA*: similarly to NonLinCFA, it iteratively compares tuples of inputs to decide if it is convenient, in terms of expected deviance, to substitute them with their aggregation through a user-defined function $h(\cdot)$.

The proposed algorithm is a variation of Algorithm 1. Its peculiarity is the different way to compute the threshold that suggests that it is convenient to aggregate two inputs. Algorithm 2 shows the pseudo-code of the function *compute_threshold*, which is the only difference w.r.t. Algorithm 1. Indeed, starting from the result of Equation 15, the algorithm aggregates two features if the upper bound of the increase of expected deviance due to the aggregation is smaller than 0. The variance and covariance of the quantities that appear in the upper bound are estimated from data, while the constant $\frac{m}{M}$ is replaced by an hyperparameter $\epsilon$, which regulates the propensity of the algorithm to aggregate. Large values give to the algorithm more propensity to aggregate, while small values require more information provided by the aggregation or more noise in the original features to perform the aggregation.

As for *NonLinCFA*, a specific version of the algorithm that preserves interpretability is to consider the inputs equal to the $D$ features $\{x_1, \ldots, x_D\}$ and the mean as aggregation function $h(x) = \frac{1}{|x|} \sum_{x_i \in x} x_i$.

---

**Algorithm 2** GenLinCFA: Generalized-Linear Correlated Features Aggregation

---

**Input:** $D$ combinations of features $\{\phi_1(x), \ldots, \phi_D(x)\}$; target $y$; $N$ samples, aggregation function $h(\cdot)$, parameter $\epsilon$
**Output:** reduced features $\{\bar{h}_\phi^1, \ldots, \bar{h}_\phi^d\}$, with $d \leq D$

    **function** COMPUTE_THRESHOLD$(h, \phi_P, \phi_j, y, \epsilon)$             ▷ Threshold from Equation 15
        $L \leftarrow |\,\hat{\mathrm{cov}}(\phi_P, y)| + |\,\hat{\mathrm{cov}}(\phi_j, y)| + \frac{1}{2} b''(0)\, \hat{\sigma}_{h(\phi)}^2$
        $R \leftarrow |\,\hat{\mathrm{cov}}(h(\phi), y)| + \frac{1}{2} b''(0)\, \hat{\sigma}_{\phi_P + \phi_j}^2$
    **return** $L - \epsilon R$
    **end function**

    **function** GENLINCFA$(Input)$                                  ▷ Main function
        Equal to NonLinCFA function in Algorithm 1
        The only difference is the renewed COMPUTE_THRESHOLD function
    **return** Reduced features as described in *Algorithm 1*
    **end function**

---

# 6 Experiments

This section describes the experiments performed on synthetic and real-world datasets to empirically validate the proposed algorithms. Additional details on datasets, methodologies and results can be found in Appendix D. Code and datasets can be found at the following anonimous link `https://www.dropbox.com/s/eoqyrs3o0ymh4o4/nonlinearFeatureAggregation.zip?dl=0` and they will be made publicly available on github upon acceptance.

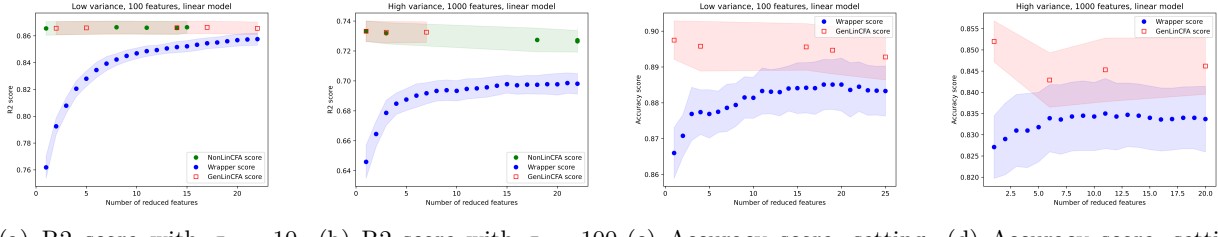

(a) R2 score with $\sigma = 10$ and $D = 100$ features.  (b) R2 score with $\sigma = 100$ and $D = 1000$.  (c) Accuracy score, setting of Figure 1a, binary target.  (d) Accuracy score, setting of Figure 1b, binary target.

Figure 1: Application of NonLinCFA and GenLinCFA in regression and classification. Confidence intervals show test performances with different numbers of reduced features considering different hyperparameters.

### 6.1 Synthetic experiments

In this subsection, we validate the proposed algorithms with synthetic experiments. We design two regression problems with $n = 3000$ samples (randomly considering 2000 samples for training and 1000 samples for testing) and we repeat the experiments ten times to produce confidence intervals. In the first regression setting, $D = 100$ linearly correlated features are considered, designing a target that linearly depends on each of them, with additive Gaussian noise with standard deviation $\sigma = 10$. In the second linear regression setting, more features are considered ($D = 1000$) and a more variable noise is added to the target ($\sigma = 100$). To validate GenLinCFA also in classification settings, we repeat the two experiments applying the sign function to the target, which transforms it into a binary variable. More details can be found in Appendix D.

Figure 1 shows the results of the application of the methods in terms of $R2$ score on the test set. In both settings, NonLinCFA and GenLinCFA have been applied, considering the mean as an aggregation function. Then, linear regression was performed on the reduced features to evaluate the performances. As a baseline, a wrapper forward feature selection has been applied, with a number of features up to the maximum number of features extracted by the two methods. The experiments were repeated 10 times to produce confidence intervals. From Figure 1a, it is possible to see that the two algorithms extract a comparable number of features, and the performance is already satisfactory with a small number of reduced features, while many features are needed for the wrapper method to reach similar results. Figure 1b reports the results in a more noisy environment with more features. With the same hyperparameters, NonLinCFA and GenLinCFA are respectively less and more prone to aggregate. As an additional comparison, LinCFA algorithm has been applied, since in this setting we are assuming the linearity of the model. The two proposed algorithms have similar performances w.r.t. LinCFA, but with the advantage to be more prone to further reduce the dimension.

To test the GenLinCFA algorithm also in a classification setting, the same two experiments are repeated, applying the sign function to the target and evaluating performances in terms of test accuracy (compared again with a wrapper forward feature selection). From Figure 1c-1d, it is possible to conclude that, again, the performances are similar or better w.r.t. the wrapper baseline and that, considering the same values of the hyperparameter, the algorithm is more prone to aggregate in a more complex and noisy context. Appendix D reports detailed results and confidence intervals of the four experiments performed as well as additional synthetic experiments showing that the two proposed algorithms can deal with non-linear transformations of the input features and with nonlinear aggregations (in particular, a quadratic relationship between the features and the target is analysed, both considering the original features and the sum of squares as aggregation function or the squared features and the mean as aggregation function).

### 6.2 Real World Experiments

To conclude the empirical analysis of the proposed algorithms, this section reports some experiments on real datasets. This analysis has been conducted evaluating test performances in terms of the coefficient of determination of the linear regression (or accuracy of the logistic regression for classification tasks),

considering the reduced features as inputs. Additionally, although the theoretical analysis has been performed considering linear methods, support vector machines for regression and classification have been considered to further empirically validate the approaches with non-linear supervised approaches. The related results in terms of coefficient of determination and accuracy can be found in Table 9 and 10 of Appendix D.

The two dimensionality reduction methods introduced in this paper have been performed with different hyperparameters, considering the original features as inputs ($\phi_i(\mathbf{x}) = x_i$) and performing the mean as aggregation function, which preserves the intepretability of the reduced features. The results are compared with different state-of-the-art dimensionality reduction methods with different characteristics (discussed in Section 2.1): linear unsupervised (PCA), linear supervised (LDA, LinCFA), nonlinear unsupervised (kernel PCA, Isomap, LLE, UMAP, t-SNE, autoencoders) and nonlinear supervised (supervised PCA, NCA). For all the approaches, the first 66% of the samples of each dataset has been considered to for training and validation purposes and the remaining 33% to test the results. Confidence intervals on the test performances have been obtained bootstrapping the training and validation set with five different seeds.

Four datasets from Kaggle and the UCI ML repository and four datasets extracted by the authors from meteorological data have been considered. In particular, The *Finance* dataset has been retrieved from Kaggle[2], while the *Bankruptcy*[3], *Parkinson*[4] and *GeneExprssion*[5] datasets have been retrieved from the UCI ML repository. The first climate dataset (*Climate, Climate(Class.)*) considers temperature, precipitation (retrieved from (Didan, 2015; Cornes et al., 2018)) and the state of vegetation (retrieved from (Zellner, 2022)) of neighbouring basins as features to predict the state of vegetation of a sub-basin of the Po River. The second dataset only focuses on temperature and precipitation features. For each climate dataset, two versions are considered: the original regression problem and a binary classification task obtained considering the values below and above the average respectively as class 0 and 1, representing conditions of water scarcity and abundance. These dataset are an example of the main applicative interest of the authors, which is to aggregate features representing measurements of the same variables at different locations through their mean, which remains interpretable (e.g., the mean of temperature measurements over a sub-region has a clear meaning for a climatologist) and significantly reduces the dimension and the autocorrelation among the reduced features. Additionally, the *GeneExpression* dataset is further relevant to empirically validate the algorithms, since it is composed by a very small number of sample (801) w.r.t. the number of features (19966).

Table 1 and 2 respectively report test performances of the four datasets from Kaggle and the UCI ML repository and meteorological data. In both tables are reported the number of reduced dimensions and the performance test score, considering the best validation hyperparameter for NonLinCFA and GenLinCFA as well as the performance of LinCFA and of the best performing algorithm among the aforementioned state-of-the-art methods considered. Additionally, the test score obtained from the initial full set of features is reported for comparison (row name *Full*). Some NA values associated with *NonLinCFA* and *LinCFA* are due to the fact that the two algorithms cannot be applied on classification tasks, which are considered to further investigate the *GenLinCFA* algorithm. The complete results related to all the baseline algorithms and to the different choices of hyperparameters can be found in Table 7 and 8 of Appendix D.

From the results it is possible to conclude that the proposed algorithms have competitive performances, with the advantage of preserving the interpretability of the reduced features, that are aggregated with the mean. In some cases (*Finance, Climate, Climate (Class.), GeneExpression*) the proposed algorithms outperform the compared methods, while in other situations they perform similarly (*Bankruptcy, Parkinson*). In the second meteorological case (*Climate II, Climate II (class.)*) the Kernel PCA algorithm (and the autoencoder with continuous target considering support vector regression) has better performances, showing that in some settings it is necessary to balance between the loss of information and the interpretability of the reduced features.

---

[2]https://www.kaggle.com/datasets/dgawlik/nyse

[3]https://archive.ics.uci.edu/ml/datasets/Polish+companies+bankruptcy+data

[4]https://archive.ics.uci.edu/dataset/470/parkinson+s+disease+classification

[5]https://archive.ics.uci.edu/dataset/401/gene+expression+cancer+rna+seq

Table 1: Experiments on real world datasets from Kaggle and UCI ML repository. Total number of samples $n$ splitted into train (66%) and test (33%) sets.

| Quantity | Finance | Bankruptcy (class.) | Parkinson (class.) | GeneExpression |
|---|---|---|---|---|
| # samples $n$ | 1299 | 1084 | 384 | 801 |
| # features $D$ | 75 | 65 | 753 | 19966 |
| **Reduced Dimension** | | | | |
| NonLinCFA | $7.4 \pm 0.4$ | NA | NA | $14.4 \pm 0.9$ |
| GenLinCFA | $8.0 \pm 1.4$ | $27.6 \pm 5.6$ | $23.4 \pm 1.1$ | $35.4 \pm 4.1$ |
| LinCFA | $11.4 \pm 0.7$ | NA | NA | $138.7 \pm 11.5$ |
| Best baseline | $36.0 \pm 10.8$ | $12.0 \pm 9.5$ | $28.2 \pm 15.3$ | $19.3 \pm 7.4$ |
| **Test performance** | **$R^2$ score** | **Accuracy score** | **Accuracy score** | |
| NonLinCFA | $0.8136 \pm 0.0036$ | NA | NA | $0.6089 \pm 0.0220$ |
| GenLinCFA | $0.8119 \pm 0.0010$ | $0.7503 \pm 0.0012$ | $0.8016 \pm 0.0069$ | $0.6209 \pm 0.0210$ |
| LinCFA | $0.8010 \pm 0.0128$ | NA | NA | $0.3135 \pm 0.1033$ |
| Best baseline | $0.7764 \pm 0.0118$ | $0.7637 \pm 0.0079$ | $0.7952 \pm 0.0181$ | $0.5639 \pm 0.0266$ |
| Full | $-5.9514 \pm 3.7166$ | $0.7446 \pm 0.0033$ | $0.7520 \pm 0.0258$ | $0.5385 \pm 0.0170$ |

Table 2: Experiments on climate datasets. The total number of samples $n$ has been splitted into train (66% of data) and test (33% of data) sets.

| | Climate | Climate (Class.) | Climate II | Climate II (class.) |
|---|---|---|---|---|
| # samples $n$ | 981 | 981 | 867 | 867 |
| # features $D$ | 1991 | 1991 | 2408 | 2408 |
| **Reduced Dimension** | | | | |
| NonLinCFA | $16.0 \pm 0.8$ | NA | $7.0 \pm 0.9$ | NA |
| GenLinCFA | $13.8 \pm 3.0$ | $17.5 \pm 3.3$ | $7.2 \pm 1.1$ | $26.0 \pm 6.1$ |
| LinCFA | $38.2 \pm 1.6$ | NA | $222.0 \pm 2.7$ | NA |
| Best baseline | $41.8 \pm 2.5$ | $37.0 \pm 6.6$ | $21.8 \pm 9.5$ | $11.1 \pm 2.3$ |
| **Test performance** | **$R^2$ score** | **Accuracy score** | **$R^2$ score** | **Accuracy score** |
| NonLinCFA | $0.9395 \pm 0.0125$ | NA | $0.2949 \pm 0.0156$ | NA |
| GenLinCFA | $0.9275 \pm 0.0004$ | $0.9107 \pm 0.0022$ | $0.2841 \pm 0.0051$ | $0.7127 \pm 0.0159$ |
| LinCFA | $0.9007 \pm 0.0310$ | NA | $-1.2861 \pm 0.2322$ | NA |
| Best baseline | $0.8454 \pm 0.0049$ | $0.8827 \pm 0.0098$ | $0.3889 \pm 0.0199$ | $0.7640 \pm 0.0062$ |
| Full | $0.7429 \pm 0.0228$ | $0.8428 \pm 0.0128$ | $-4.3511 \pm 0.9161$ | $0.6429 \pm 0.0205$ |

## 7    Conclusions

In this paper, we have deepened the study of dimensionality reduction to account for non-linear effects, focusing on preserving both information and interpretability. The non-linearity has been accounted for in both the deterministic mapping function and the noise model, considering the exponential family of distributions. The resulting algorithms aggregate, in the most general case, non-linear aggregation of non-linear features. Theoretical results have been provided to investigate the performance of the aggregation either in terms of MSE or increase of deviance. The experimental validation illustrates that our algorithms outperform the proposed baselines both in synthetically generated environments and in a real-world domain, or they have competitive results, with the advantage of letting the user define the most suitable aggregation function that, in most cases, has been selected as the mean to preserve the interpretability of the reduced features. Future works will include the consideration of additional indexes of statistical dependence, other than the correlation and covariance, to perform the aggregation (e.g., mutual information). Additionally, the proposed algorithms will be further applied to investigate their impact on the detection of the state of vegetation of European river basins.

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

# A  Non-Linear Correlated Features Aggregation: additional proofs and results

This section contains additional results and proofs related to Section 4 of the main paper.

Firstly, we introduce the finite-sample increase of variance due to the aggregation, of which we reported the asymptotic version in Theorem 1 of the main paper.

**Theorem 5.** *Let the relationship between the features and the target be nonlinear with additive Gaussian noise (Equation 1 of the main paper). The decrease of variance $\Delta_{var}$ between a bivariate linear regression and the univariate case where the two features are aggregated (Equation 7 of the main paper) is:*

$$\Delta_{var} = \frac{\sigma^2}{(n-1)}\left[\frac{\sigma_{\phi_1}^2\hat{\sigma}_{\phi_2}^2 + \sigma_{\phi_2}^2\hat{\sigma}_{\phi_1}^2 - 2cov(\phi_1,\phi_2)\hat{cov}(\phi_1,\phi_2)}{(\hat{\sigma}_{\phi_1}^2\hat{\sigma}_{\phi_2}^2 - \hat{cov}(\phi_1,\phi_2)^2)} - \frac{\sigma_{h(\phi_1,\phi_2)}^2}{\hat{\sigma}_{h(\phi_1,\phi_2)}^2}\right]. \tag{16}$$

**Proof of Theorem 5**

Recalling the result:

$$\Delta_{var} = \frac{\sigma^2}{(n-1)}\left[\frac{\sigma_{\phi_1}^2\hat{\sigma}_{\phi_2}^2 + \sigma_{\phi_2}^2\hat{\sigma}_{\phi_1}^2 - 2cov(\phi_1,\phi_2)\hat{cov}(\phi_1,\phi_2)}{(\hat{\sigma}_{\phi_1}^2\hat{\sigma}_{\phi_2}^2 - \hat{cov}(\phi_1,\phi_2)^2)} - \frac{\sigma_{h(\phi(x))}^2}{\hat{\sigma}_{h(\phi(x))}^2}\right],$$

we will compute the variance for the bivariate case and for the univariate case, and then compute their difference. To do so, we need to start by computing the variance of the estimators $(var_D(\hat{w}|\mathbf{X}))$.

**Lemma 1.** *In the one dimensional case $\hat{y} = \hat{w}h(\phi(x))$:*

$$var_D(\hat{w}|\mathbf{X}) = \frac{\sigma^2}{(n-1)\hat{\sigma}_{h(\phi(x))}^2}.$$

*In the two dimensional case $\hat{y} = \hat{w}_1\phi_1(x) + \hat{w}_2\phi_2(x)$:*

$$var_D(\hat{w}|\mathbf{X}) = \frac{\sigma^2}{(n-1)(\hat{\sigma}_{\phi_1}^2\hat{\sigma}_{\phi_2}^2 - \hat{cov}(\phi_1,\phi_2)^2)}\begin{bmatrix} \hat{\sigma}_{\phi_2}^2 & -\hat{cov}(\phi_1,\phi_2) \\ -\hat{cov}(\phi_1,\phi_2) & \hat{\sigma}_{\phi_1}^2 \end{bmatrix}.$$

*Proof.* For the one dimensional result, denoting with $\mathbf{h}(\phi)$ the $N$ dimensional vector of aggregated samples $[h(\phi_1(x^1),\phi_2(x^1)), ..., h(\phi_1(x^N),\phi_2(x^N))]$:

$$var_D(\hat{w}|\mathbf{X}) = var_D((\mathbf{h}(\phi)^\intercal\mathbf{h}(\phi))^{-1}\mathbf{h}(\phi)^\intercal y|\mathbf{X}) = (\mathbf{h}(\phi)^\intercal\mathbf{h}(\phi))^{-1}\sigma^2$$

$$= \begin{bmatrix} h(\phi(x^1)) & ... & h(\phi(x^n)) \end{bmatrix}\begin{bmatrix} h(\phi(x^1)) \\ ... \\ h(\phi(x^n)) \end{bmatrix}\sigma^2$$

$$= \frac{\sigma^2}{\sum_{i=1}^N h(\phi(x^i))^2} = \frac{\sigma^2}{(n-1)\hat{\sigma}_{h(\phi(x))}^2}.$$

In the first equality it is exploited the closed formula to estimate the linear regression coefficients in linear regression, the second equality exploits the property of the variance to extract the constant matrices and the definition of variance of the target $y$. Finally, the third and fourth equalities are simple algebraic computations.

In the two dimensional setting, denoting with $\mathbf{\Phi} = \begin{bmatrix} \phi_1(x^1) & \phi_2(x^1) \\ ... & ... \\ \phi_1(x^n) & \phi_2(x^n) \end{bmatrix}$ the $N \times 2$ matrix of samples:

$$var_D(\hat{w}|\mathbf{X}) = (\mathbf{\Phi}^\intercal\mathbf{\Phi})^{-1}\sigma^2 = \left(\begin{bmatrix} \phi_1(x^1) & ... & \phi_1(x^n) \\ \phi_2(x^1) & ... & \phi_2(x^n) \end{bmatrix}\begin{bmatrix} \phi_1(x^1) & \phi_2(x^1) \\ ... & ... \\ \phi_1(x^n) & \phi_2(x^n) \end{bmatrix}\right)^{-1}\sigma^2$$

$$= \left( \begin{bmatrix} \phi_1(x^1)^2 + ... + \phi_1(x^n)^2 & \phi_1(x^1)\phi_2(x^1) + ... + \phi_1(x^n)\phi_2(x^n) \\ \phi_1(x^1)\phi_2(x^1) + ... + \phi_1(x^n)\phi_2(x^n) & \phi_2(x^1)^2 + ... + \phi_2(x^n)^2 \end{bmatrix} \right)^{-1} \sigma^2$$

$$= \frac{\sigma^2}{(n-1)(\hat{\sigma}_{\phi_1}^2 \hat{\sigma}_{\phi_2}^2 - c\hat{o}v(\phi_1, \phi_2)^2)} \begin{bmatrix} \hat{\sigma}_{\phi_2}^2 & -c\hat{o}v(\phi_1, \phi_2) \\ -c\hat{o}v(\phi_1, \phi_2) & \hat{\sigma}_{\phi_1}^2 \end{bmatrix}.$$

The first equivalence follows again from the closed form solution of linear regression and from the variance of the target, while the others follow from algebraic computations. □

We are now ready to derive the expression of variance of the model $(\mathbb{E}[(h_D(x) - \bar{h}(x))^2 | \mathbf{X}])$ in the two cases.

**Theorem 6.** *In the one dimensional case $\hat{y} = \hat{w}g(\phi(x))$:*

$$\mathbb{E}[(h_D(x) - \bar{h}(x))^2 | \mathbf{X}] = \frac{\sigma^2}{(n-1)} \cdot \frac{\sigma_{h(\phi(x))}^2}{\hat{\sigma}_{h(\phi(x))}^2}.$$

*In the two dimensional case $\hat{y} = \hat{w}_1 \phi_1(x) + \hat{w}_2 \phi_2(x)$:*

$$\mathbb{E}[(h_D(x) - \bar{h}(x))^2 | \mathbf{X}] = \frac{\sigma^2}{(n-1)} \cdot \frac{\sigma_{\phi_1}^2 \hat{\sigma}_{\phi_2}^2 + \sigma_{\phi_2}^2 \hat{\sigma}_{\phi_1}^2 - 2cov(\phi_1, \phi_2)c\hat{o}v(\phi_1, \phi_2)}{(\hat{\sigma}_{\phi_1}^2 \hat{\sigma}_{\phi_2}^2 - c\hat{o}v(\phi_1, \phi_2)^2)}.$$

*Proof.* One dimensional, exploiting the independence between training, test samples and the definition of model variance and the assumption of expected values equal to zero of $h(\phi(x))$:

$$\mathbb{E}_{X,Y,D}[(\mathcal{M}_D(x) - \bar{\mathcal{M}}(x))^2] = \mathbb{E}_{X,Y,D}[(\hat{w}h(\phi(x)) - \mathbb{E}_D[\hat{w}h(\phi(x))])^2]$$

$$= \mathbb{E}_{X,Y,D}[(h(\phi(x))(\hat{w} - \mathbb{E}_D[\hat{w}]))^2]$$

$$= \mathbb{E}_{X,Y}[h(\phi(x))^2]\mathbb{E}_D[(\hat{w} - \mathbb{E}_D[\hat{w}])^2]$$

$$= var_X(h(\phi(x))) \cdot var_D(\hat{w}).$$

Conditioning on the features training set:

$$\mathbb{E}_{X,Y,D}[(\mathcal{M}_D(x) - \bar{\mathcal{M}}(x))^2 | \mathbf{X}] = \sigma_{h(\phi(x))}^2 \cdot \frac{\sigma^2}{(n-1)\hat{\sigma}_{h(\phi(x))}^2}.$$

Two dimensional, exploiting again the independence between train and test set and the assumption of expected values equal to zero of $\phi_1(x)$ and $\phi_2(x)$:

$$\mathbb{E}_{X,Y,D}[(\mathcal{M}_D(x) - \bar{\mathcal{M}}(x))^2] = \mathbb{E}_{X,Y,D}[(\hat{w}_1\phi_1 + \hat{w}_2\phi_2 - \mathbb{E}_D[\hat{w}_1\phi_1 + \hat{w}_2\phi_2])^2]$$

$$= \mathbb{E}_{X,Y,D}[(\phi_1(\hat{w}_1 - \mathbb{E}_D[\hat{w}_1]) + \phi_2(\hat{w}_2 - \mathbb{E}_D[\hat{w}_2]))^2]$$

$$= \mathbb{E}_{X,Y,D}[(\phi_1(\hat{w}_1 - \mathbb{E}_D[\hat{w}_1])^2] + \mathbb{E}_{X,Y,D}[(\phi_2(\hat{w}_2 - \mathbb{E}_D[\hat{w}_2]))^2]$$

$$+ 2\mathbb{E}_{X,Y,D}[\phi_1\phi_2(\hat{w}_1 - \mathbb{E}_D[\hat{w}_1])(\hat{w}_2 - \mathbb{E}_D[\hat{w}_2])]$$

$$= var_X(\phi_1)var_D(\hat{w}_1) + var_X(\phi_2)var_D(\hat{w}_2) + 2cov_X(\phi_1, \phi_2)cov_D(\hat{w}_1, \hat{w}_2).$$

Conditioning on the features training set:

$$\mathbb{E}_{X,Y,D}[(\mathcal{M}_D(x) - \bar{\mathcal{M}}(x))^2 | \mathbf{X}]$$

$$= var_X(\phi_1 | \mathbf{X})var_D(\hat{w}_1 | \mathbf{X}) + var_X(\phi_2 | \mathbf{X})var_D(\hat{w}_2 | \mathbf{X})$$

$$+ 2cov_X(\phi_1, \phi_2 | \mathbf{X})cov_D(\hat{w}_1, \hat{w}_2 | \mathbf{X})$$

$$= \frac{\sigma^2}{(n-1)} \cdot \frac{\sigma_{\phi_1}^2 \hat{\sigma}_{\phi_2}^2 + \sigma_{\phi_2}^2 \hat{\sigma}_{\phi_1}^2 - 2cov(\phi_1, \phi_2)c\hat{o}v(\phi_1, \phi_2)}{(\hat{\sigma}_{\phi_1}^2 \hat{\sigma}_{\phi_2}^2 - c\hat{o}v(\phi_1, \phi_2)^2)}.$$

□

In conclusion, Theorem 5 trivially follows by computing the difference between the two results of the theorem, i.e., between the variance of the two and the one dimensional settings.

Moreover Theorem 1, in the main paper, follows from this result substituting each estimator with the quantity that it estimates, since it is assume to be convergent in probability to its value.

**Proof of Theorem 2**

Recalling the asymptotic result of the theorem:

$$\Delta_{bias}^{n\to\infty} = -\frac{cov(f, h(\phi(x)))^2}{\sigma_{h(\phi(x))}^2}$$
$$+ \frac{\sigma_{\phi_1}^2 cov(\phi_2, f)^2 + \sigma_{\phi_2}^2 cov(\phi_1, f)^2 - 2cov(\phi_1, f)cov(\phi_2, f)cov(\phi_1, \phi_2)}{\sigma_{\phi_1}^2 \sigma_{\phi_2}^2 - cov(\phi_1, \phi_2)^2},$$

we firstly need to derive the expected value of the estimators $(\mathbb{E}_D[\hat{w}|\mathbf{X}])$.

**Lemma 2.** *The expected value of the coefficient of the one dimensional regression is:*

$$\mathbb{E}_D[\hat{w}|\mathbf{X}] = \frac{c\hat{o}v(h(\phi(x)), f(x))}{\hat{\sigma}_{h(\phi(x))}^2}.$$

*For the two dimensional regression:*

$$\mathbb{E}_D[\hat{w}|\mathbf{X}] = \frac{1}{(\hat{\sigma}_{\phi_1}^2 \hat{\sigma}_{\phi_2}^2 - c\hat{o}v(\phi_1, \phi_2)^2)} \begin{bmatrix} \hat{\sigma}_{\phi_2}^2 c\hat{o}v(\phi_1, f) - c\hat{o}v(\phi_1, \phi_2)c\hat{o}v(\phi_2, f) \\ \hat{\sigma}_{\phi_1}^2 c\hat{o}v(\phi_2, f) - c\hat{o}v(\phi_1, \phi_2)c\hat{o}v(\phi_1, f) \end{bmatrix}.$$

*Proof.* In the one-dimensional case, exploiting the closed form solution of the estimate of the linear regression coefficient:

$$\mathbb{E}_D[\hat{w}|\mathbf{X}] = (h(\mathbf{\Phi})^\intercal h(\mathbf{\Phi}))^{-1} h(\mathbf{\Phi})^\intercal f(\mathbf{X})$$
$$= \frac{1}{(n-1)\hat{\sigma}_{h(\phi(x))}^2} \begin{bmatrix} h(\phi(x^1)) & ... & h(\phi(x^n)) \end{bmatrix} \begin{bmatrix} f(x^1) \\ ... \\ f(x^n) \end{bmatrix}$$
$$= \frac{c\hat{o}v(h(\phi(x)), f(x))}{\hat{\sigma}_{h(\phi(x))}^2}.$$

In the two dimensional case, substituting the expression of the estimated coefficients:

$$\mathbb{E}_D(\hat{w}|\mathbf{X}) =$$
$$= (\mathbf{\Phi}^\intercal \mathbf{\Phi})^{-1} \mathbf{\Phi}^\intercal f(\mathbf{X})$$
$$= \left( \begin{bmatrix} \phi_1(x^1) & ... & \phi_1(x^n) \\ \phi_2(x^1) & ... & \phi_2(x^n) \end{bmatrix} \begin{bmatrix} \phi_1(x^1) & \phi_2(x^1) \\ ... & ... \\ \phi_1(x^n) & \phi_2(x^n) \end{bmatrix} \right)^{-1} \mathbf{\Phi}^\intercal f(\mathbf{X})$$
$$= \frac{1}{(n-1)(\hat{\sigma}_{\phi_1}^2 \hat{\sigma}_{\phi_2}^2 - c\hat{o}v(\phi_1, \phi_2)^2)}$$
$$\times \begin{bmatrix} \hat{\sigma}_{\phi_2}^2 & -c\hat{o}v(\phi_1, \phi_2) \\ -c\hat{o}v(\phi_1, \phi_2) & \hat{\sigma}_{\phi_1}^2 \end{bmatrix} \begin{bmatrix} \phi_1(x^1) & ... & \phi_1(x^n) \\ \phi_2(x^1) & ... & \phi_2(x^n) \end{bmatrix} \begin{bmatrix} f(x^1) \\ ... \\ f(x^n) \end{bmatrix}$$
$$= \frac{1}{(n-1)(\hat{\sigma}_{\phi_1}^2 \hat{\sigma}_{\phi_2}^2 - c\hat{o}v(\phi_1, \phi_2)^2)}$$
$$\times \begin{bmatrix} \hat{\sigma}_{\phi_2}^2 \sum_i f(x^i)\phi_1(x^i) - c\hat{o}v(\phi_1, \phi_2) \sum_i f(x^i)\phi_2(x^i) \\ -c\hat{o}v(\phi_1, \phi_2) \sum_i f(x^i)\phi_1(x^i) + \hat{\sigma}_{\phi_1}^2 \sum_i f(x^i)\phi_2(x^i) \end{bmatrix}$$

$$= \frac{1}{(\hat{\sigma}_{\phi_1}^2 \hat{\sigma}_{\phi_2}^2 - c\hat{o}v(\phi_1,\phi_2)^2)} \begin{bmatrix} \hat{\sigma}_{\phi_2}^2 c\hat{o}v(\phi_1,f) - c\hat{o}v(\phi_1,\phi_2)c\hat{o}v(\phi_2,f) \\ \hat{\sigma}_{\phi_1}^2 c\hat{o}v(\phi_2,f) - c\hat{o}v(\phi_1,\phi_2)c\hat{o}v(\phi_1,f) \end{bmatrix}.$$

□

The next step of the proof is to derive the expression of (squared) bias of the two linear regression settings $(\mathbb{E}[(\bar{\mathcal{M}}(x) - \bar{y})^2|\mathbf{X}])$.

**Theorem 7.** *Let* $h = h(\phi(x))$, $f = f(x)$.
*In the one dimensional linear regression* $\hat{y} = \hat{w}h$ *the squared bias is:*

$$\mathbb{E}[(\bar{\mathcal{M}}(x) - \bar{y})^2|\mathbf{X}] = \sigma_f^2 + \frac{\sigma_h^2 c\hat{o}v(f,h)^2 - 2cov(f,h)c\hat{o}v(f,h)\hat{\sigma}_h^2}{\hat{\sigma}_h^4}.$$

*In the two dimensional case* $\hat{y} = \hat{w}_1\phi_1 + \hat{w}_2\phi_2$ *the squared bias is:*

$$\begin{aligned}
\mathbb{E}[(\bar{\mathcal{M}}(x) - \bar{y})^2|\mathbf{X}] &= \sigma_{\phi_1}^2 \mathbb{E}_D[\hat{w}_1|\mathbf{X}]^2 + \sigma_{\phi_2}^2 \mathbb{E}_D[\hat{w}_2|\mathbf{X}]^2 \\
&+ 2cov(\phi_1,\phi_2)\mathbb{E}_D[\hat{w}_1|\mathbf{X}]\mathbb{E}_D[\hat{w}_2|\mathbf{X}] + \sigma_f^2 \\
&- 2cov(\phi_1,f)\mathbb{E}_D[\hat{w}_1|\mathbf{X}] - 2cov(\phi_2,f)\mathbb{E}_D[\hat{w}_2|\mathbf{X}].
\end{aligned}$$

*Proof.* In the one dimensional case, exploiting the independence between the train and test set:

$$\begin{aligned}
\mathbb{E}_{X,Y,D}[(\bar{\mathcal{M}}(x) - \bar{y})^2|\mathbf{X}] &= \mathbb{E}_{X,Y,D}[(\mathbb{E}_D[\hat{w}h(\phi(x))] - f(x))^2|\mathbf{X}] \\
&= \mathbb{E}_X[(h(\phi(x))\mathbb{E}_D[\hat{w}|\mathbf{X}] - f(x))^2] \\
&= \mathbb{E}_X[h(\phi(x))^2]\mathbb{E}_D[\hat{w}|\mathbf{X}]^2 + \mathbb{E}_X[f(x)^2] - 2\mathbb{E}_X[f(x)h(\phi(x))]\mathbb{E}_D[\hat{w}|\mathbf{X}] \\
&= \frac{\sigma_h^2}{\hat{\sigma}_h^4} c\hat{o}v(f,h)^2 + \sigma_f^2 - 2cov(f,h)\frac{c\hat{o}v(f,h)}{\hat{\sigma}_h^2},
\end{aligned}$$

where the last equation follows substituting the expected value of the coefficient with its expression derived in the previous lemma and exploiting the assumption of null expected value of the aggregation function $h$.

In the two dimensional setting, exploiting again the independence between train and test set and the null assumption of the expected value of the two functions of features $\phi_1, \phi_2$:

$$\begin{aligned}
\mathbb{E}_{X,Y,D}[(\bar{\mathcal{M}}(x) - \bar{y})^2|\mathbf{X}] &= \mathbb{E}_{X,Y}[(\phi_1(x)\mathbb{E}_D[\hat{w}_1] + \phi_2(x)\mathbb{E}_D[\hat{w}_2] - f(x))^2|\mathbf{X}] \\
&= \sigma_{\phi_1}^2 \mathbb{E}_D[\hat{w}_1|\mathbf{X}]^2 + \sigma_{\phi_2}^2 \mathbb{E}_D[\hat{w}_2|\mathbf{X}]^2 + 2cov(\phi_1,\phi_2)\mathbb{E}_D[\hat{w}_1|\mathbf{X}]\mathbb{E}_D[\hat{w}_2|\mathbf{X}] \\
&+ \sigma_f^2 - 2\mathbb{E}[f(x)(\phi_1\mathbb{E}_D[\hat{w}_1] + \phi_2\mathbb{E}_D[\hat{w}_2])|\mathbf{X}] \\
&= \sigma_{\phi_1}^2 \mathbb{E}_D[\hat{w}_1|\mathbf{X}]^2 + \sigma_{\phi_2}^2 \mathbb{E}_D[\hat{w}_2|\mathbf{X}]^2 + 2cov(\phi_1,\phi_2)\mathbb{E}_D[\hat{w}_1|\mathbf{X}]\mathbb{E}_D[\hat{w}_2|\mathbf{X}] \\
&+ \sigma_f^2 - 2cov(\phi_1,f)\mathbb{E}_D[\hat{w}_1|\mathbf{X}] - 2cov(\phi_2,f)\mathbb{E}_D[\hat{w}_2|\mathbf{X}].
\end{aligned}$$

□

**Remark 10.** *In the two dimensional result of Theorem 7 the expected values of the coefficients, found in Lemma 2, are not explicitly inserted, to improve readability. The full expression of squared bias, in the bivariate case, would be:*

$$\begin{aligned}
&\mathbb{E}[(\bar{\mathcal{M}}(x) - \bar{y})^2|\mathbf{X}] \\
&= \sigma_{\phi_1}^2 \left( \frac{\hat{\sigma}_{\phi_2}^2 c\hat{o}v(\phi_1,f) - c\hat{o}v(\phi_1,\phi_2)c\hat{o}v(\phi_2,f)}{(\hat{\sigma}_{\phi_1}^2 \hat{\sigma}_{\phi_2}^2 - c\hat{o}v(\phi_1,\phi_2)^2)} \right)^2 \\
&+ \sigma_{\phi_2}^2 \left( \frac{\hat{\sigma}_{\phi_1}^2 c\hat{o}v(\phi_2,f) - c\hat{o}v(\phi_1,\phi_2)c\hat{o}v(\phi_1,f)}{(\hat{\sigma}_{\phi_1}^2 \hat{\sigma}_{\phi_2}^2 - c\hat{o}v(\phi_1,\phi_2)^2)} \right)^2
\end{aligned}$$

$$+ 2cov(\phi_1, \phi_2)$$

$$\times \left( \frac{(\hat{\sigma}_{\phi_2}^2 c\hat{o}v(\phi_1, f) - c\hat{o}v(\phi_1, \phi_2)c\hat{o}v(\phi_2, f))(\hat{\sigma}_{\phi_1}^2 c\hat{o}v(\phi_2, f) - c\hat{o}v(\phi_1, \phi_2)c\hat{o}v(\phi_1, f))}{(\hat{\sigma}_{\phi_1}^2 \hat{\sigma}_{\phi_2}^2 - c\hat{o}v(\phi_1, \phi_2)^2)^2} \right)$$

$$+ \sigma_f^2$$

$$- 2cov(\phi_1, f)\left( \frac{\hat{\sigma}_{\phi_2}^2 c\hat{o}v(\phi_1, f) - c\hat{o}v(\phi_1, \phi_2)c\hat{o}v(\phi_2, f)}{(\hat{\sigma}_{\phi_1}^2 \hat{\sigma}_{\phi_2}^2 - c\hat{o}v(\phi_1, \phi_2)^2)} \right)$$

$$- 2cov(\phi_2, f)\left( \frac{\hat{\sigma}_{\phi_1}^2 c\hat{o}v(\phi_2, f) - c\hat{o}v(\phi_1, \phi_2)c\hat{o}v(\phi_1, f)}{(\hat{\sigma}_{\phi_1}^2 \hat{\sigma}_{\phi_2}^2 - c\hat{o}v(\phi_1, \phi_2)^2)} \right).$$

*That, after algebraic computations, is equal to:*

$$\mathbb{E}[(\bar{\mathcal{M}}(x) - \bar{y})^2 | \mathbf{X}]$$

$$= \sigma_f^2 + \frac{1}{(\hat{\sigma}_{\phi_1}^2 \hat{\sigma}_{\phi_2}^2 - c\hat{o}v(\phi_1, \phi_2)^2)^2} \times \Bigg\{$$

$$\sigma_{\phi_1}^2 \hat{\sigma}_{\phi_2}^4 c\hat{o}v(\phi_1, f)^2 + \sigma_{\phi_1}^2 c\hat{o}v(\phi_1, \phi_2)^2 c\hat{o}v(\phi_2, f)^2$$

$$- 2\sigma_{\phi_1}^2 \hat{\sigma}_{\phi_2}^2 c\hat{o}v(\phi_1, f)c\hat{o}v(\phi_2, f)c\hat{o}v(\phi_1, \phi_2)$$

$$+ \sigma_{\phi_2}^2 \hat{\sigma}_{\phi_1}^4 c\hat{o}v(\phi_2, f)^2 + \sigma_{\phi_2}^2 c\hat{o}v(\phi_1, \phi_2)^2 c\hat{o}v(\phi_1, f)^2$$

$$- 2\sigma_{\phi_2}^2 \hat{\sigma}_{\phi_1}^2 c\hat{o}v(\phi_1, f)c\hat{o}v(\phi_2, f)c\hat{o}v(\phi_1, \phi_2)$$

$$+ 2cov(\phi_1, \phi_2)\hat{\sigma}_{\phi_1}^2 \hat{\sigma}_{\phi_2}^2 c\hat{o}v(\phi_1, f)c\hat{o}v(\phi_2, f)$$

$$+ 2cov(\phi_1, \phi_2)c\hat{o}v(\phi_1, f)c\hat{o}v(\phi_2, f)c\hat{o}v(\phi_1, \phi_2)^2$$

$$- 2cov(\phi_1, \phi_2)\hat{\sigma}_{\phi_2}^2 c\hat{o}v(\phi_1, f)^2 c\hat{o}v(\phi_1, \phi_2)$$

$$- 2cov(\phi_1, \phi_2)\hat{\sigma}_{\phi_1}^2 c\hat{o}v(\phi_2, f)^2 c\hat{o}v(\phi_1, \phi_2)$$

$$- 2cov(\phi_1, f)\hat{\sigma}_{\phi_1}^2 \hat{\sigma}_{\phi_2}^4 c\hat{o}v(\phi_1, f)$$

$$+ 2cov(\phi_1, f)\hat{\sigma}_{\phi_2}^2 c\hat{o}v(\phi_1, f)c\hat{o}v(\phi_1, \phi_2)^2$$

$$+ 2cov(\phi_1, f)\hat{\sigma}_{\phi_1}^2 \hat{\sigma}_{\phi_2}^2 c\hat{o}v(\phi_2, f)c\hat{o}v(\phi_1, \phi_2)$$

$$- 2cov(\phi_1, f)c\hat{o}v(\phi_2, f)c\hat{o}v(\phi_1, \phi_2)^3$$

$$- 2cov(\phi_2, f)\hat{\sigma}_{\phi_2}^2 \hat{\sigma}_{\phi_1}^4 c\hat{o}v(\phi_2, f)$$

$$+ 2cov(\phi_2, f)\hat{\sigma}_{\phi_1}^2 c\hat{o}v(\phi_2, f)c\hat{o}v(\phi_1, \phi_2)^2$$

$$+ 2cov(\phi_2, f)\hat{\sigma}_{\phi_1}^2 \hat{\sigma}_{\phi_2}^2 c\hat{o}v(\phi_1, f)c\hat{o}v(\phi_1, \phi_2)$$

$$- 2cov(\phi_2, f)c\hat{o}v(\phi_1, f)c\hat{o}v(\phi_1, \phi_2)^3$$

$$\Bigg\}.$$

**Lemma 3.** *In the asmyptotic case, considering each estimator convergent in probability to the quantity that it estimates, the (squared) bias for the univariate* $(bias_{1D}^{n \to \infty})$ *and bivariate* $(bias_{2D}^{n \to \infty})$ *linear regression under analysis is respectively:*

$$bias_{1D}^{n \to \infty} = \sigma_f^2 - \frac{cov(f, h)^2}{\sigma_h^2},$$

$$bias_{2D}^{n \to \infty} = \sigma_f^2 + \frac{2cov(\phi_1, f)cov(\phi_2, f)cov(\phi_1, \phi_2) - \sigma_{\phi_1}^2 cov(\phi_2, f)^2 - \sigma_{\phi_2}^2 cov(\phi_1, f)^2}{\sigma_{\phi_1}^2 \sigma_{\phi_2}^2 - cov(\phi_1, \phi_2)^2}.$$

*Proof.* To prove the two results it is enough to start from the results of Theorem 7 and Remark 10 and substitute each estimator with the quantity that it estimates, to which it converges in probability. $\square$

Theorem 2, that is reported in the main paper, is finally proved from the two asymptotic quantities derived in Lemma 3, subtracting the univariate to the bivariate result.

## B    Generalized-Linear Correlated Features Aggregation: bivariate analysis

In this section we show a first bivariate result related to generalized linear models. As in the general case, we assume the conditional distribution of the target given the feature to belong to the canonical exponential family:

$$f_{\boldsymbol{\theta}}(y) = exp(\frac{y\boldsymbol{\theta} - b(\boldsymbol{\theta})}{\phi}) + c(y, \phi).$$

Moreover, we assume that the expected value of the target is a linear combination of the inputs, subsequently transformed with the canonical link function:

$$\mathbb{E}[y|x] = g^{-1}(w_1 x_1 + w_2 x_2).$$

In this setting we compare a bivariate model with the univariate one that substitutes the two features with their aggregation through a zero-mean function $h(\cdot)$:

$$\begin{cases} \hat{y} = g^{-1}(\hat{w_1} x_1 + \hat{w_2} x_2) \\ \hat{y} = g^{-1}(\hat{w} \cdot \frac{x_1 + x_2}{2}). \end{cases}$$

As a first result, we prove the following lemma that justifies the adoption of the expected deviance as a goodness of fit measure.

**Lemma 4.** *Assuming a bivariate generalized linear model and considering a Gaussian distribution of the target given the features $y \sim \mathbb{N}(\mu, \sigma^2) \implies f_{\boldsymbol{\theta}}(y) = exp\{\frac{y\boldsymbol{\theta} - \boldsymbol{\theta}^2/2}{\phi^2} + c(y, \phi)\}, \boldsymbol{\theta} = \mu, \phi = \sigma$ the difference of performance in terms of* MSE *between the bivariate linear regression and the univariate one that aggregates the two input features with their mean is equivalent to the difference of deviance between the two models.*

*Proof.* Recalling the definition of deviance:

$$D^*(\boldsymbol{\theta}, \hat{\boldsymbol{\theta}}) := \frac{D(\boldsymbol{\theta}, \hat{\boldsymbol{\theta}})}{\phi} = -2\left[\ell(\hat{\boldsymbol{\theta}}) - \ell(\boldsymbol{\theta})\right] = \frac{2}{\phi}[y(\boldsymbol{\theta} - \hat{\boldsymbol{\theta}}) - (b(\boldsymbol{\theta}) - b(\hat{\boldsymbol{\theta}}))],$$

in the Gaussian setting the parameter $\boldsymbol{\theta} = \mu = w_1 x_1 + w_2 x_2$ represents the mean of the distribution of the target $y$. Defining $\hat{\boldsymbol{\theta}} = \hat{\mu}_1$ the mean of the target estimated with the univariate regression and $\bar{\boldsymbol{\theta}} = \hat{\mu}_2$ the one of the bivariate case, the increase of expected deviance between the two models is:

$$\mathbb{E}_{x,y,\mathcal{T}}[D^*(\boldsymbol{\theta}, \hat{\boldsymbol{\theta}}) - D^*(\boldsymbol{\theta}, \bar{\boldsymbol{\theta}})] = \mathbb{E}_{x,y,\mathcal{T}}[D^*(\mu, \hat{\mu}_1) - D^*(\mu, \hat{\mu}_2)].$$

Moreover, assuming Gaussianity we have $b(\boldsymbol{\theta}) = \frac{\boldsymbol{\theta}^2}{2}$ and the link function $g(\cdot)$ is the identity function (indeed, in the bivariate linear regression we just compare the prediction $\hat{y} = \hat{w}_1 x_1 + \hat{w}_2 x_2$ with the univariate $\hat{y} = \hat{w}\frac{x_1 + x_2}{2}$).

Therefore, the expected increase of deviance becomes:

$$\mathbb{E}_{x,y,\mathcal{T}}[D^*\left(\mu, \hat{\mu}_1\right) - D^*\left(\mu, \hat{\mu}_2\right)] = \frac{2}{\phi}\mathbb{E}_{x,y,\mathcal{T}}[y(\hat{\mu}_2 - \hat{\mu}_1) - (b(\hat{\mu}_2) - b(\hat{\mu}_1))]$$

$$= \frac{2}{\sigma^2}\mathbb{E}_{x,y,\mathcal{T}}\left[y(\hat{w}_1 x_1 + \hat{w}_2 x_2 - \hat{w}\frac{x_1 + x_2}{2}) - \frac{(\hat{w}_1 x_1 + \hat{w}_2 x_2)^2}{2} - \frac{(\hat{w}\bar{x})^2}{2}\right]$$

$$= \frac{2}{\sigma^2}\Bigg\{\mathbb{E}_{x,y}\Bigg[$$

$$y \cdot \left(w_1 x_1 + w_2 x_2 - \frac{2\left(w_1\hat{\sigma}_{x_1}^2 + w_2\hat{\sigma}_{x_2}^2 + (w_1 + w_2)\,c\hat{o}v\,(x_1, x_2)\right)}{\hat{\sigma}_{x_1}^2 + \hat{\sigma}_x^2 + 2c\hat{o}v\,(x_1, x_2)} \cdot \frac{x_1 + x_2}{2}\right)\Bigg]$$

$$-\mathbb{E}_{x,y}\mathbb{E}_{\mathcal{T}}\left[\frac{(\hat{w}_1 x_1 + \hat{w}_2 x_2)^2}{2} - \frac{(\hat{w}\bar{x})^2}{2}\right]\Bigg\}.$$

The last equation follows from the independence of train and test set and the expression of the expected value of the regression coefficients (Equation A2,A3 of Bonetti et al. (2023)).

Asymptotically, the first expected value is:

$$\mathbb{E}_{x,y}\left[y \cdot \left(w_1 x_1 + w_2 x_2 - \frac{2\left(w_1\sigma_{x_1}^2 + w_2\sigma_{x_2}^2 + (w_1 + w_2)\,cov\,(x_1, x_2)\right)}{\sigma_{x_1}^2 + \sigma_{x_2}^2 + 2cov\,(x_1, x_2)} \cdot \frac{x_1 + x_2}{2}\right)\right]$$

$$= \mathbb{E}_{x,y}\left[y \cdot \left(w_1 x_1 + w_2 x_2 - \frac{\left(w_1\sigma_{x_1}^2 + w_2\sigma_{x_2}^2 + (w_1 + w_2)\,cov\,(x_1, x_2)\right)}{\sigma_{x_1}^2 + \sigma_{x_2}^2 + 2\,cov\,(x_1, x_2)} \cdot (x_1 + x_2)\right)\right]$$

$$= w_1 E_{x,y}\left[x_1 y\right] + w_2\mathbb{E}_{x,y}\left[x_2 y\right]$$

$$-\frac{\left(w_1\sigma_{x_1}^2 + w_2\sigma_{x_2}^2 + (w_1 + w_2)\,cov\,(x_1 x_2)\right)}{\sigma_{x_1}^2 + \sigma_{x_2}^2 + 2\,cov\,(x_1, x_2)} \cdot (\mathbb{E}_{x,y}\left[x_1 y\right] + E_{x,y}\left[x_2 y\right]).$$

Recalling the definition of covariance and the zero-mean assumption of the expected values:

$$\mathbb{E}_{x,y}\left[x_1 y\right] = cov\,(x_1, y) - \mathbb{E}_x\left[x_1\right]\mathbb{E}_y[y]$$

$$= cov\,(x_1, w_1 x_1 + w_2 x_2 + \varepsilon) - 0 = w_1\sigma_{x_1}^2 + w_2\,cov\,(x_1, x_2).$$

A similar argumentation holds for $\mathbb{E}_{x,y}\left[x_2 y\right]$, therefore the expected value under analysis becomes:

$$w_1^2\sigma_{x_1}^2 + w_2^2\sigma_{x_2}^2 + 2w_1 w_2\,cov\,(x_1, x_2)$$

$$-\frac{\left(w_1\sigma_{x_1}^2 + w_2\sigma_{x_2}^2 + (w_1 + w_2)\,cov\,(x_1, x_2)\right)^2}{\sigma_{x_1}^2 + \sigma_{x_2}^2 + 2\,cov\,(x_1 x_2)}.$$

The second expected value that appears in the equation representing the increase of deviance is asymptotically equal to:

$$\mathbb{E}_{x,y}\mathbb{E}_{\mathcal{T}}\left[\frac{(\hat{w}_1 x_1 + \hat{w}_2 x_2)^2}{2} - \frac{(\hat{w}\bar{x})^2}{2}\right] = \frac{1}{2}\mathbb{E}_{x,y}\mathbb{E}_{\mathcal{T}}\left[\hat{w}_1^2 x_1^2 + \hat{w}_2^2 x_2^2 + 2w_1 w_2 x_1 x_2 - \hat{w}^2\bar{x}^2\right]$$

$$= \mathbb{E}_{\mathcal{T}}\left[\hat{w}_1^2\right]\mathbb{E}_x\left[x_1^2\right] + \mathbb{E}_{\mathcal{T}}\left[\hat{w}_2^2\right]\mathbb{E}_x\left[x_2^2\right] + 2\mathbb{E}_{\mathcal{T}}\left[\hat{w}_1\hat{w}_2\right]\mathbb{E}_x\left[x_1 x_2\right] - \mathbb{E}_{\mathcal{T}}\left[\hat{w}^2\right]\mathbb{E}_x\left[\bar{x}^2\right]$$

$$= \sigma_{x_1}^2\left(w_1^2 + \frac{\sigma^2\sigma_{x_2}^2}{(n-1)\left(\sigma_{x_1}^2\sigma_{x_2}^2 - cov^2\,(x_1, x_2)\right)}\right)$$

$$+ \sigma_{x_2}^2\left(w_2^2 + \frac{\sigma^2\sigma_{x_1}^2}{(n-1)\left(\sigma_{x_1}^2\sigma_{x_2}^2 - cov^2\,(x_2, x_2)\right)}\right)$$

$$+ 2cov(x_1, x_2) \left( w_1 w_2 - \frac{\sigma^2 cov(x_1, x_2)}{(n-1)\left(\sigma_{x_1}^2 \sigma_{x_2}^2 - cov^2(x_1 x_2)\right)} \right)$$

$$- \frac{1}{4} \left[ \left(\sigma_{x_1}^2 + \sigma_{x_2}^2 + 2cov(x_1, x_2)\right) \right.$$

$$\times \left( \frac{\sigma^2}{(n-1) \cdot \frac{1}{4}\left(\sigma_{x_1}^2 + \sigma_{x_2}^2 + 2cov(x_1, x_2)\right)} \right.$$

$$\left. \left. + \left( \frac{2\left(w_1 \sigma_{x_1}^2 + w_2 \sigma_{x_2}^2 + (w_1 + w_2) cov(x_1, x_2)\right)}{\sigma_{x_1}^2 + \sigma_{x_2}^2 + 2cov(x_1, x_2)} \right)^2 \right) \right]$$

$$= w_1^2 \sigma_{x_1}^2 + w_2^2 \sigma_{x_2}^2 + 2\,\text{cov}(x_1 x_2) w_1 w_2 + \frac{2\sigma^2\left(\sigma_{x_1}^2 \sigma_{x_2}^2 - cov^2(x_1 x_2)\right)}{(n-1)\left(\sigma_{x_1}^2 \sigma_{x_2}^2 - cov^2(x_1, x_2)\right)}$$

$$- \left[ \frac{\sigma^2}{n-1} + \frac{\left(w_1 \sigma_{x_1}^2 + w_2 \sigma_{x_2}^2 + (w_1 + w_2) \cdot \text{cov}(x_1, x_2)\right)^2}{\sigma_{x_1}^2 + \sigma_{x_2}^2 + 2\,\text{cov}(x_1, x_2)} \right]$$

$$= w_1^2 \sigma_{x_1}^2 + w_2^2 \sigma_{x_2}^2 + 2\,\text{cov}(x_1, x_2) w_1 w_2 + \frac{\sigma^2}{n-1}$$

$$- \frac{\left(w_1 \sigma_{x_1}^2 + w_2 \sigma_{x_2}^2 + (w_1 + w_2) \text{cov}(x_1, x_2)\right)^2}{\sigma_{x_1}^2 + \sigma_{x_2}^2 + 2\,\text{cov}(x_1, x_2)},$$

where the last two equalities follow again from the expression of expected value and variance of the regression coefficients.

Combining the two asymptotic terms, the expected increase of deviance becomes:

$$\mathbb{E}_{x,y,\mathcal{T}}[D^*(\mu, \hat{\mu}_1) - D^*(\mu, \hat{\mu}_2)] = \frac{2}{\sigma^2}\left( w_1^2 \sigma_{x_1}^2 + w_2^2 \sigma_{x_2}^2 + 2w_1 w_2 \,\text{cov}(x_1, x_2) \right.$$

$$\left. - \frac{\left(w_1 \sigma_{x_1}^2 + w_2 \sigma_{x_2}^2 + (w_1 + w_2)\,\text{cov}(x_1, x_2)\right)^2}{\sigma_{x_1}^2 + \sigma_{x_2}^2 + 2\,\text{cov}(x_1, x_2)} \right)$$

$$- \frac{1}{\sigma^2}\left( w_1^2 \sigma_{x_1}^2 + w_2^2 \sigma_{x_2}^2 + 2\,\text{cov}(x_1, x_2) w_1 w_2 + \frac{\sigma^2}{n-1} \right.$$

$$\left. - \frac{\left(w_1 \sigma_{x_1}^2 + w_2 \sigma_{x_2}^2 + (w_1 + w_2)\,\text{cov}(x_1 x_2)\right)^2}{\sigma_{x_1}^2 + \sigma_{x_2}^2 + 2\,\text{cov}(x_1 x_2)} \right)$$

$$= \frac{1}{\sigma^2}\left( w_1^2 \sigma_{x_1}^2 + w_2^2 \sigma_{x_2}^2 + 2w_1 w_2 \,\text{cov}(x_1, x_2) \right.$$

$$\left. - \frac{\left(w_1 \sigma_{x_1}^2 + w_2 \sigma_{x_2}^2 + (w_1 + w_2)\,\text{cov}(x_1, x_2)\right)^2}{\sigma_{x_1}^2 + \sigma_{x_2}^2 + 2\,\text{cov}(x_1 x_2)} - \frac{\sigma^2}{n-1} \right)$$

$$= \frac{1}{\sigma^2(\sigma_{x_1}^2 + \sigma_{x_2}^2 + 2\,\text{cov}(x_1 x_2))}\left( w_1^2 \sigma_{x_1}^2 \sigma_{x_2}^2 + w_2^2 \sigma_{x_1}^2 \sigma_{x_2}^2 \right.$$

$$\left. - w_1^2 \,\text{cov}^2(x_1, x_2) - w_2^2 \,\text{cov}^2(x_1, x_2) - 2w_1 w_2 \sigma_{x_1}^2 \sigma_{x_2}^2 + 2w_1 w_2 \,\text{cov}^2(x_1, x_2) \right)$$

$$- \frac{1}{\sigma^2} \cdot \frac{\sigma^2}{n-1}.$$

Recalling that the asymptotic increase of bias in the linear bivariate setting is:

$$\frac{\sigma_{x_1}^2 \sigma_{x_2}^2 (1 - \rho_{x_1,x_2}^2)(w_1 - w_2)^2}{\sigma_{x_1+x_2}^2},$$

and the asymptotic decrease of variance is $\frac{\sigma^2}{n-1}$ (respectively Equation 9 and 13 of Bonetti et al. (2023)), the first is equal to the first term found comparing the deviance and the second one is equal to the second one, proving the equivalence.

$\square$

After having verified that, in linear contexts, the analysis of deviance is tight w.r.t. the mean squared error, we are now ready to introduce the main result of this section.

**Theorem 8.** *Let $M$ be the maximum between the following differences: $\mathbb{E}_{\mathcal{T}}[\hat{w}_1] - \frac{1}{2}\mathbb{E}_{\mathcal{T}}[\hat{w}]$, $\mathbb{E}_{\mathcal{T}}[\hat{w}_2] - \frac{1}{2}\mathbb{E}_{\mathcal{T}}[\hat{w}]$, $\mathbb{E}_{\mathcal{T}}[\hat{w}_1^2] - \frac{1}{4}\mathbb{E}_{\mathcal{T}}[\hat{w}^2]$, $\mathbb{E}_{\mathcal{T}}[\hat{w}_2^2] - \frac{1}{4}\mathbb{E}_{\mathcal{T}}[\hat{w}^2]$, $\mathbb{E}_{\mathcal{T}}[\hat{w}_1\hat{w}_2] - \frac{1}{4}\mathbb{E}_{\mathcal{T}}[\hat{w}^2]$, and let $m$ be the minimum of the same quantities. Moreover, let the real model belong to the canonical exponential family.*
*Defining $\Delta(D^*)$ as the expected increase of deviance due to the aggregation of the two features $x_1, x_2$ with their mean in the bivariate setting, it is equal to:*

$$\Delta(D^*) \leq \frac{2}{\phi}\left\{ M \cdot (|\operatorname{cov}(x_1, y)| + |\operatorname{cov}(x_2, y)|) - \frac{1}{2}m \cdot b''(0) \cdot (\sigma_{x_1+x_2}^2) \right\}. \tag{17}$$

*Proof.* In the bivariate setting, the expected increase of deviance under analysis is equal to:

$$\frac{2}{\phi}\{E_{x,y}\left[y \cdot (\mathbb{E}_{\mathcal{T}}[\hat{w}_1]x_1 + \mathbb{E}_{\mathcal{T}}[\hat{w}_2]x_2 - \mathbb{E}_{\mathcal{T}}[\hat{w}]\tilde{x})\right]$$
$$- (\mathbb{E}_{x,y}\mathbb{E}_{\mathcal{T}}[b(\hat{w}_1 x_1 + \hat{w}_2 x_2) - b(\hat{w}\tilde{x})])\}.$$

**First expected value** The first expected value of the previous equation can be rewritten as:

$$E_{x,y}\left[y \cdot (\mathbb{E}_{\mathcal{T}}[\hat{w}_1]x_1 + \mathbb{E}_{\mathcal{T}}[\hat{w}_2]x_2 - \mathbb{E}_{\mathcal{T}}[\hat{w}]\tilde{x})\right]$$
$$= E_{x,y}[yx_1]\mathbb{E}_{\mathcal{T}}[\hat{w}_1] + E_{x,y}[yx_2]\mathbb{E}_{\mathcal{T}}[\hat{w}_2] - E_{x,y}\left[y\frac{x_1 + x_2}{2}\right]\mathbb{E}_{\mathcal{T}}[\hat{w}]$$
$$= \operatorname{cov}(x_1, y)\mathbb{E}_{\mathcal{T}}[\hat{w}_1] + \operatorname{cov}(x_2, y)\mathbb{E}_{\mathcal{T}}[\hat{w}_2] - \frac{1}{2}\operatorname{cov}(x_1, y)\mathbb{E}_{\mathcal{T}}[\hat{w}] - \frac{1}{2}\operatorname{cov}(x_2, y)\mathbb{E}_{\mathcal{T}}[\hat{w}]$$
$$= \operatorname{cov}(x_1, y)\left(\mathbb{E}_{\mathcal{T}}[\hat{w}_1] - \frac{1}{2}\mathbb{E}_{\mathcal{T}}[\hat{w}]\right) + \operatorname{cov}(x_2, y)\left(\mathbb{E}_{\mathcal{T}}[\hat{w}_2] - \frac{1}{2}\mathbb{E}_{\mathcal{T}}[\hat{w}]\right).$$

**Second expected value** The second expected value that appears in the expected increase of deviance under analysis, $\mathbb{E}_{x,y}\mathbb{E}_{\mathcal{T}}[b(\hat{w}_1 x_1 + \hat{w}_2 x_2) - b(\hat{w}\tilde{x})]$, depends on the function $b(\cdot)$, that is a specific parameter of the distribution. In order to derive a result that holds for any distribution belonging to the canonical exponential family, we use a second order Taylor approximation of the function, centered in $\boldsymbol{\theta}_0 = 0$.

$$\begin{cases} b(\hat{w}_1 x_1 + \hat{w}_2 x_2) \simeq b(0) + b'(0) \cdot (\hat{w}_1 x_1 + \hat{w}_2 x_2) + \frac{1}{2}b''(0) \cdot (\hat{w}_1 x_1 + \hat{w}_2 x_2)^2 \\ b(\hat{w}\tilde{x}) \simeq b(0) + b'(0) \cdot (\hat{w}\tilde{x}) + \frac{1}{2}b''(0) \cdot (\hat{w}\tilde{x})^2. \end{cases}$$

Since the first-order terms vanish because of the zero-mean assumption, the approximated expected value is therefore:

$$\mathbb{E}_{x,y,\mathcal{T}}[b(\hat{w}_1 x_1 + \hat{w}_2 x_2) - b(\hat{w}\tilde{x})] \simeq \frac{1}{2}b''(0) \cdot \mathbb{E}_{x,y,\mathcal{T}}\left[\left((\hat{w}_1 x_1 + \hat{w}_2 x_2)^2 - (\hat{w}\tilde{x})^2\right)\right]$$
$$= \frac{1}{2}b''(0) \cdot \left[\sigma_{x_1}^2 \cdot \mathbb{E}_{\mathcal{T}}[\hat{w}_1^2 - \frac{\hat{w}^2}{4}] + \sigma_{x_2}^2 \cdot \mathbb{E}_{\mathcal{T}}[\hat{w}_2^2 - \frac{\hat{w}^2}{4}]\right.$$

$$+ 2\operatorname{cov}(x_1, x_2) \cdot \mathbb{E}_{\mathcal{T}}[\hat{w}_1 \hat{w}_2 - \frac{\hat{w}^2}{4}]\Big],$$

$$\iff \frac{1}{2}b''(0) \cdot [\operatorname{var}(\hat{w}_1 x_1 + \hat{w}_2 x_2) - \operatorname{var}(\hat{w}\bar{x})].$$

**Expected increase of deviance** We are now ready to combine the two expressions:

$$\Delta(D^*) = \frac{2}{\phi}\Big\{ E_{x,y}\left[y \cdot (\mathbb{E}_{\mathcal{T}}[\hat{w}_1]\, x_1 + \mathbb{E}_{\mathcal{T}}[\hat{w}_2]\, x_2 - \mathbb{E}_{\mathcal{T}}[\hat{w}]\bar{x})\right]$$

$$- (\mathbb{E}_{x,y}\mathbb{E}_{\mathcal{T}}\left[b\left(\hat{w}_1 x_1 + \hat{w}_2 x_2\right) - b(\hat{w}\bar{x})\right]) \Big\}$$

$$= \frac{2}{\phi}\Big\{ \operatorname{cov}(x_1, y)\left(\mathbb{E}_{\mathcal{T}}[\hat{w}_1] - \frac{1}{2}\mathbb{E}_{\mathcal{T}}[\hat{w}]\right) + \operatorname{cov}(x_2, y)\left(\mathbb{E}_{\mathcal{T}}[\hat{w}_2] - \frac{1}{2}\mathbb{E}_{\mathcal{T}}[\hat{w}]\right)$$

$$- \frac{1}{2}b''(0)\left[\sigma_{x_1}^2\left(\mathbb{E}_{\mathcal{T}}[\hat{w}_1^2] - \frac{1}{4}\mathbb{E}_{\mathcal{T}}[\hat{w}^2]\right) + \sigma_{x_2}^2\left(\mathbb{E}_{\mathcal{T}}[\hat{w}_2^2] - \frac{1}{4}\mathbb{E}_{\mathcal{T}}[\hat{w}^2]\right)\right.$$

$$\left.+ 2\operatorname{cov}(x_1, x_2)\left(\mathbb{E}_{\mathcal{T}}[\hat{w}_1 \hat{w}_2] - \frac{1}{4}\mathbb{E}_{\mathcal{T}}[\hat{w}^2]\right)\right]\Big\}.$$

Finally, substituting with: $M$ the maximum difference of the expected value of the coefficients that appear in the equation ($\mathbb{E}_{\mathcal{T}}[\hat{w}_1] - \frac{1}{2}\mathbb{E}_{\mathcal{T}}[\hat{w}]$, $\mathbb{E}_{\mathcal{T}}[\hat{w}_2] - \frac{1}{2}\mathbb{E}_{\mathcal{T}}[\hat{w}]$, $\mathbb{E}_{\mathcal{T}}[\hat{w}_1^2] - \frac{1}{4}\mathbb{E}_{\mathcal{T}}[\hat{w}^2]$, $\mathbb{E}_{\mathcal{T}}[\hat{w}_2^2] - \frac{1}{4}\mathbb{E}_{\mathcal{T}}[\hat{w}^2]$, $\mathbb{E}_{\mathcal{T}}[\hat{w}_1 \hat{w}_2] - \frac{1}{4}\mathbb{E}_{\mathcal{T}}[\hat{w}^2]$), and with $m$ the minimum of the same quantities, the result follows. □

We conclude this bivariate analysis with a justification of the choice to center in 0 the Taylor expansion in the proof above and providing an intuitive interpretation of the result of the theorem.

**Remark 11.** *Considering the center in zero ($\theta_0 = 0$) for the second-order Taylor expansion of the function $b(\cdot)$ in the analysis of deviance, in the linear asymptotic case, the second order Taylor expansion is an exact approximation of the term.*

*Proof.* In the linear case, recalling that $b(\theta) = \frac{\theta^2}{2}$, the expected value that contains the function $b(\cdot)$ in the proof, is:

$$\mathbb{E}\left[b\left(\hat{w}_1 x_1 + \hat{w}_2 x_2\right) - b(\hat{w}\bar{x})\right] = \frac{1}{2}\mathbb{E}\left[(\hat{w}_1 x_1 + \hat{w}_2 x_2)^2 - (\hat{w}\bar{x})^2\right]$$

$$= \frac{1}{2}\mathbb{E}\left[\hat{w}_1^2 x_1^2 + \hat{w}_2^2 x_2^2 + 2\hat{w}_1 \hat{w}_2 x_1 x_2 - \frac{\hat{w}^2 x_1^2}{4} - \frac{\hat{w}^2 x_2^2}{4} - \frac{\hat{w}^2 x_1 x_2}{2}\right]$$

$$= \frac{1}{2}\left[\sigma_{x_1}^2 \cdot \mathbb{E}_{\mathcal{T}}\left[\hat{w}_1^2 - \frac{\hat{w}^2}{4}\right] + \sigma_{x_2}^2 \cdot \mathbb{E}_{\mathcal{T}}\left[\hat{w}_2^2 - \frac{\hat{w}^2}{4}\right]\right.$$

$$\left.+ 2\operatorname{cov}(x_1 x_2) \cdot \mathbb{E}_{\mathcal{T}}\left[\hat{w}_1 \hat{w}_2 - \frac{\hat{w}^2}{4}\right]\right].$$

□

*Moreover, $b''(0) = 1$. Therefore, this quantity is equal to the general expression of the second order Taylor expansion centered in $0$.*

**Remark 12.** *The result of the theorem suggests that it is convenient to aggregate two variables when the variance of the sum between the two variables is large or the absolute value of the covariance between each feature and the target is small. Indeed, this implies respectively that there is much noise or that each feature shares a lot of information with the target individually.*

## C   Generalized-Linear Correlated Features Aggregation: additional proofs and results

**Proof of Theorem 4**

Recalling the expression of the expected increase of deviance:

$$\frac{2}{\phi}\Big\{ \mathbb{E}_{x,y}\left[y \cdot (\mathbb{E}_{\mathcal{T}}[\hat{w}_1]\phi_1 + \mathbb{E}_{\mathcal{T}}[\hat{w}_2]\phi_2 - \mathbb{E}_{\mathcal{T}}[\hat{w}]h(\phi_1,\phi_2))\right] \tag{18}$$
$$- \mathbb{E}_{x,y,\mathcal{T}}\left[b\left(\hat{w}_1\phi_1 + \hat{w}_2\phi_2\right) - b(\hat{w}h(\phi_1,\phi_2))\right] \Big\},$$

we analyse the two expected values separately.

Exploiting the zero-mean assumption, the first expected value is equal to:

$$E_{x,y}\left[y \cdot (\mathbb{E}_{\mathcal{T}}\left[\hat{w}_1\right]\phi_1 + \mathbb{E}_{\mathcal{T}}\left[\hat{w}_2\right]\phi_2 - \mathbb{E}_{\mathcal{T}}[\hat{w}]h(\phi_1,\phi_2))\right]$$
$$= E_{x,y}\left[y\phi_1\right]\mathbb{E}_{\mathcal{T}}\left[\hat{w}_1\right] + E_{x,y}\left[y\phi_2\right]\mathbb{E}_{\mathcal{T}}\left[\hat{w}_2\right] - E_{x,y}\left[yh(\phi_1,\phi_2)\right]\mathbb{E}_{\mathcal{T}}[\hat{w}]$$
$$= \operatorname{cov}(\phi_1, y)\mathbb{E}_{\mathcal{T}}\left[\hat{w}_1\right] + \operatorname{cov}(\phi_2, y)\mathbb{E}_{\mathcal{T}}\left[\hat{w}_2\right] - \operatorname{cov}(h(\phi_1,\phi_2), y)\mathbb{E}_{\mathcal{T}}[\hat{w}].$$

The second expectation, on the other hand, is:

$$\mathbb{E}_{x,y,\mathcal{T}}\left[b\left(\hat{w}_1\phi_1 + \hat{w}_2\phi_2\right) - b(\hat{w}h(\phi_1,\phi_2))\right].$$

In order to obtain an expression that holds for any distribution in the canonical exponential family, similarly to the bivatiate case of the previous section, we perform a second order Taylor approximation of the function $b(\cdot)$:

$$\begin{cases} b\left(\hat{w}_1\phi_1 + \hat{w}_2\phi_2\right) \simeq b\left(0\right) + b'\left(0\right) \cdot \left(\hat{w}_1\phi_1 + \hat{w}_2\phi_2\right) + \frac{1}{2}b''\left(0\right) \cdot \left(\hat{w}_1\phi_1 + \hat{w}_2\phi_2\right)^2 \\ b(\hat{w}h(\phi_1,\phi_2)) \simeq b\left(0\right) + b'\left(0\right) \cdot \left(\hat{w}h(\phi_1,\phi_2)\right) + \frac{1}{2}b''\left(0\right) \cdot \left(\hat{w}h(\phi_1,\phi_2)\right)^2. \end{cases}$$

The expected value under analysis therefore becomes:

$$\mathbb{E}_{x,y,\mathcal{T}}\left[b\left(\hat{w}_1\phi_1 + \hat{w}_2\phi_2\right) - b(\hat{w}h(\phi_1,\phi_2))\right]$$
$$\simeq \frac{1}{2}b''\left(0\right) \cdot \mathbb{E}_{x,y,\mathcal{T}}\left[\left(\hat{w}_1\phi_1 + \hat{w}_2\phi_2\right)^2 - \left(\hat{w}h(\phi_1,\phi_2)\right)^2\right]$$
$$= \frac{1}{2}b''\left(0\right) \cdot [\sigma_{\phi_1}^2 \mathbb{E}_{\mathcal{T}}[\hat{w}_1^2] + \sigma_{\phi_2}^2 \mathbb{E}_{\mathcal{T}}[\hat{w}_2^2]$$
$$+ 2\operatorname{cov}(\phi_1,\phi_2)\mathbb{E}_{\mathcal{T}}[\hat{w}_1\hat{w}_2] - \sigma_{h(\phi_1,\phi_2)}^2 \mathbb{E}_{\mathcal{T}}[\hat{w}^2]]$$
$$= \frac{1}{2}b''\left(0\right) \cdot \left[\operatorname{var}(\hat{w}_1\phi_1 + \hat{w}_2\phi_2) - \operatorname{var}(\hat{w}h(\phi_1,\phi_2))\right].$$

Merging the two expected values, the expected increase of deviance finally becomes:

$$\Delta(D^*) = \frac{2}{\phi}\Big\{ \operatorname{cov}(\phi_1, y)\mathbb{E}_{\mathcal{T}}\left[\hat{w}_1\right] + \operatorname{cov}(\phi_2, y)\mathbb{E}_{\mathcal{T}}\left[\hat{w}_2\right] - \operatorname{cov}(h(\phi_1,\phi_2), y)\mathbb{E}_{\mathcal{T}}[\hat{w}]$$
$$- \frac{1}{2}b''\left(0\right) \cdot [\sigma_{\phi_1}^2 \mathbb{E}_{\mathcal{T}}[\hat{w}_1^2] + \sigma_{\phi_2}^2 \mathbb{E}_{\mathcal{T}}[\hat{w}_2^2]$$
$$+ 2\operatorname{cov}(\phi_1,\phi_2)\mathbb{E}_{\mathcal{T}}[\hat{w}_1\hat{w}_2] - \sigma_{h(\phi_1,\phi_2)}^2 \mathbb{E}_{\mathcal{T}}[\hat{w}^2]]\Big\}$$
$$\leq \frac{2}{\phi}\Big\{ \left[M \cdot (|\operatorname{cov}(\phi_1, y)| + |\operatorname{cov}(\phi_2, y)|) - m \cdot |\operatorname{cov}(h(\phi), y)|\right]$$
$$- \frac{1}{2}b''\left(0\right) \cdot \left(m \cdot \sigma_{x_1+x_2}^2 - M \cdot \sigma_{h(\phi)}^2\right)\Big\},$$

where $M, m$ are respectively the maximum and minimum absolute values of the expected values of the coefficients and their squared values. This concludes the proof, since the second expression of the theorem follows by rearranging the terms.

### C.1 Additional Gaussian considerations

In this section we prove the statement of Remark 9, assuming that the generalized linear model under analysis follows a Gaussian distribution.

**Lemma 5.** *Let a generalized linear model have Gaussian distribution of the target given the features ($Y|X \sim \mathbb{N}(\mu, \sigma^2)$) and compare the bivariate linear regression having as inputs two zero-mean transformations of the features $\phi_1(x), \phi_2(x)$ with a univariate linear regression having as input the aggregation of $\phi_1, \phi_2$ through a function $h(\cdot)$ (Equation 7 of the main paper). The increase of* expected MSE *due to the aggregation is equal to the increase of the* expected deviance.

*Proof.* Recalling that, with the Gaussianity assumption, the link function $g(\cdot)$ is the identity and the function $b(\cdot) = \frac{\theta^2}{2}$, the expression of the expected increase of deviance (Equation 18) becomes:

$$
\frac{2}{\sigma^2} \Big\{ \mathbb{E}_{x,y} \left[ y \cdot (\mathbb{E}_{\mathcal{T}}[\hat{w}_1]\phi_1 + \mathbb{E}_{\mathcal{T}}[\hat{w}_2]\phi_2 - \mathbb{E}_{\mathcal{T}}[\hat{w}]h(\phi_1, \phi_2)) \right]
$$
$$
- \frac{1}{2} \mathbb{E}_{x,y,\mathcal{T}} \left[ (\hat{w}_1\phi_1 + \hat{w}_2\phi_2)^2 - (\hat{w}h(\phi_1, \phi_2))^2 \right] \Big\}
$$
$$
= \Big\{ \mathbb{E}_{x,y} \left[ y \cdot \left( \frac{\phi_1(\sigma_{\phi_2}^2 cov(\phi_1, f) - cov(\phi_1, \phi_2)cov(\phi_2, f)) + \phi_2(\sigma_{\phi_1}^2 cov(\phi_2, f) - cov(\phi_1, \phi_2)cov(\phi_1, f))}{\sigma_{\phi_1}^2 \sigma_{\phi_2}^2 - cov^2(\phi_1, \phi_2)} \right. \right.
$$
$$
\left. \left. - \frac{cov(h(\phi_1, \phi_2), f)}{\sigma_h^2(\phi_1, \phi_2)} \cdot h(\phi_1, \phi_2) \right) \right]
$$
$$
- \frac{1}{2} E_{x,y,\mathcal{T}} \left[ \hat{w}_1^2\phi_1^2 + \hat{w}_2^2\phi_2^2 + 2\hat{w}_1\hat{w}_2\phi_1\phi_2 - \hat{w}^2\hat{h}^2(\phi_1, \phi_2) \right] \Big\},
$$

where the equation holds substituting the expression of the expected value of the linear regression coefficients.

*First expected value.*
Exploiting the independence between train and test data and the zero mean assumption, the first expected value in the equation above becomes:

$$
\frac{cov(\phi_1, y) \cdot \left[ \sigma_{\phi_2}^2 cov(\phi_1, f) - cov(\phi_1, \phi_2) cov(\phi_2, f) \right] + cov(\phi_2, y) \cdot \left[ \sigma_{\phi_1}^2 cov(\phi_2, f) - cov(\phi_1, \phi_2) cov(\phi_1, f) \right]}{\sigma_{\phi_1}^2 \sigma_{\phi_2}^2 - cov^2(\phi_1, \phi_2)}
$$
$$
- \frac{cov(h(\phi_1, \phi_2), f)}{\sigma_{h(\phi_1,\phi_2)}^2} \cdot cov(h(\phi_1, \phi_2), y)
$$
$$
\frac{\sigma_{\phi_2}^2 cov^2(\phi_1, f) - cov(\phi_1, \phi_2) cov(\phi_1, f) cov(\phi_2, f) + \sigma_{\phi_1}^2 cov^2(\phi_2, f) - cov(\phi_1, \phi_2) cov(\phi_1, f) cov(\phi_2, f)}{\sigma_{\phi_1}^2 \sigma_{\phi_2}^2 - cov^2(\phi_1, \phi_2)}
$$
$$
- \frac{cov^2(h(\phi_1, \phi_2), f)}{\sigma_{h(\phi_1,\phi_2)}^2}
$$
$$
= \frac{\sigma_{\phi_2}^2 cov^2(\phi_1, f) + \sigma_{\phi_1}^2 cov^2(\phi_2, f) - 2 cov(\phi_1, \phi_2) cov(\phi_1, f) cov(\phi_2, f)}{\sigma_{\phi_1}^2 \sigma_{\phi_2}^2 - cov^2(\phi_1, \phi_2)} - \frac{cov^2(h(\phi_1, \phi_2), f)}{\sigma_{h(\phi_1,\phi_2)}^2}
$$
$$
= \Delta_{\text{bias}}^{n\to\infty},
$$

where the last expression is exactly the increase of (squared) asymptotic bias found in Equation 11.

*Second expected value*
Substituting the expected values of the estimates of the regression coefficients and their variances, the second expected value of the expression of the increase of deviance becomes:

$$
\mathbb{E}_{\mathcal{T}} \left[ \hat{w}_1^2 \right] \sigma_{\phi_1}^2 + \mathbb{E}_{\mathcal{T}} \left[ \hat{w}_2^2 \right] \sigma_{\phi_2}^2 + 2\mathbb{E}_{\mathcal{T}} \left[ \hat{w}_1\hat{w}_2 \right] cov(\phi_1, \phi_2) - \mathbb{E}_{\mathcal{T}} \left[ \hat{w}^2 \right] \sigma_{h(\phi_1,\phi_2)}^2
$$
$$
= \left( var(\hat{w}_1) + \mathbb{E}_{\mathcal{T}} \left[ \hat{w}_1 \right]^2 \right) \sigma_{\phi_1}^2 + \left( var(\hat{w}_2) + \mathbb{E}_{\mathcal{T}} \left[ \hat{w}_2 \right]^2 \right) \sigma_{\phi_2}^2
$$

$$+ 2\left(\text{cov}\left(\hat{w}_1, \hat{w}_2\right) + \mathbb{E}_{\mathcal{T}}\left[\hat{w}_1\right]\mathbb{E}_{\mathcal{T}}\left[\hat{w}_2\right]\right)\text{cov}\left(\phi_1, \phi_2\right) - \left(\text{var}(\hat{w}) + \mathbb{E}_{\mathcal{T}}[\hat{w}]^2\right)\sigma^2_{h(\phi_1, \phi_2)}$$

$$= \left[\frac{\sigma^2 \cdot \sigma^2_{\phi_2}}{(n-1)\left(\sigma^2_{\phi_1}\sigma^2_{\phi_2} - \text{cov}^2\left(\phi_1, \phi_2\right)\right)} + \left(\frac{\sigma^2_{\phi_2}\text{cov}\left(\phi_1, f\right) - \text{cov}\left(\phi_1, \phi_2\right)\text{cov}\left(\phi_2, f\right)}{\sigma^2_{\phi_1}\sigma^2_{\phi_2} - \text{cov}^2\left(\phi_1, \phi_2\right)}\right)^2\right]\sigma^2_{\phi_1}$$

$$+ \left[\frac{\sigma^2 \cdot \sigma^2_{\phi_1}}{(n-1)\left(\sigma^2_{\phi_1}\sigma^2_{\phi_2} - \text{cov}^2\left(\phi_1, \phi_2\right)\right)} + \left(\frac{\sigma^2_{\phi_1}\text{cov}\left(\phi_2, f\right) - \text{cov}\left(\phi_1, \phi_2\right)\text{cov}\left(\phi_1, f\right)}{\sigma^2_{\phi_1}\sigma^2_{\phi_2} - \text{cov}^2\left(\phi_1, \phi_2\right)}\right)^2\right]\sigma^2_{\phi_2}$$

$$+ 2\,\text{cov}\left(\phi_1, \phi_2\right)\left[\frac{-\sigma^2 \cdot \text{cov}\left(\phi_1, \phi_2\right)}{(n-1)\left(\sigma^2_{\phi_1}\sigma^2_{\phi_2} - \text{cov}^2\left(\phi_1\phi_2\right)\right)}\right.$$

$$\left. + \left(\frac{\left(\sigma^2_{\phi_2}\text{cov}\left(\phi_1, f\right) - \text{cov}\left(\phi_1, \phi_2\right)\text{cov}\left(\phi_2, f\right)\right)\cdot\left(\sigma^2_{\phi_1}\text{cov}\left(\phi_2, f\right) - \text{cov}\left(\phi_1, \phi_2\right)\text{cov}\left(\phi_1, f\right)\right)}{\left(\sigma^2_{\phi_1}\sigma^2_{\phi_2} - \text{cov}^2\left(\phi_1, \phi_2\right)\right)^2}\right)\right]$$

$$- \left(\frac{\sigma^2}{n-1}\cdot\frac{1}{\sigma^2_{h(\phi_1, \phi_2)}} + \left(\frac{\text{cov}\left(h\left(\phi_1, \phi_2\right), f\right)}{\sigma^2_{h(\phi_1, \phi_2)}}\right)^2\right)\sigma^2_{h(\phi_1, \phi_2)}.$$

Considering the terms with $n-1$ in the denominator, they are equal to the asymptotic decrease of variance found in Equation 10:

$$\frac{\sigma^2}{n-1} \cdot \left[\frac{\sigma^2_{\phi_2}\sigma^2_{\phi_1} - \sigma^2_{\phi_1}\sigma^2_{\phi_2} - 2\,\text{cov}^2\left(\phi_1, \phi_2\right)}{\sigma^2_{\phi_1}\sigma^2_{\phi_2} - \text{cov}^2\left(\phi_1, \phi_2\right)} - \frac{\sigma^2_{h(\phi_1, \phi_2)}}{\sigma^2_{h(\phi_1, \phi_2)}}\right]$$

$$= \frac{\sigma^2}{n-1} \cdot (2-1) = \frac{\sigma^2}{n-1} = \Delta^{n\to\infty}_{\text{var}}.$$

The remaining terms are:

$$\sigma^2_{\phi_1}\left(\frac{\sigma^2_{\phi_2}\text{cov}\left(\phi_1, f\right) - \text{cov}\left(\phi_1, \phi_2\right)\text{cov}\left(\phi_2, f\right)}{\sigma^2_{\phi_1}\sigma^2_{\phi_2} - \text{cov}^2\left(\phi_1, \phi_2\right)}\right)^2 + \sigma^2_{\phi_2}\left(\frac{\sigma^2_{\phi_1}\text{cov}\left(\phi_2, f\right) - \text{cov}\left(\phi_1, \phi_2\right)\text{cov}\left(\phi_1, f\right)}{\sigma^2_{\phi_1}\sigma^2_{\phi_2} - \text{cov}^2\left(\phi_1, \phi_2\right)}\right)^2$$

$$+ 2\,\text{cov}\left(\phi_1, \phi_2\right)\left(\frac{\left(\sigma^2_{\phi_2}\text{cov}\left(\phi_1, f\right) - \text{cov}\left(\phi_1, \phi_2\right)\text{cov}\left(\phi_2, f\right)\right)\cdot\left(\sigma^2_{\phi_1}\text{cov}\left(\phi_2, f\right) - \text{cov}\left(\phi_1, \phi_2\right)\text{cov}\left(\phi_1, f\right)\right)}{\left(\sigma^2_{\phi_1}\sigma^2_{\phi_2} - \text{cov}^2\left(\phi_1, \phi_2\right)\right)^2}\right)$$

$$- \frac{\text{cov}^2\left(h\left(\phi_1, \phi_2\right), f\right)}{\sigma^2_{h(\phi_1, \phi_2)}}.$$

The first three terms of this expression have the same denominator, therefore we can focus on their numerators:

$$\sigma^2_{\phi_1}\left(\sigma^4_{\phi_2}\text{cov}^2\left(\phi_1, f\right) + \text{cov}^2\left(\phi_1, \phi_2\right)\text{cov}^2\left(\phi_2, f\right) - 2\sigma^2_{\phi_2}\text{cov}\left(\phi_1, f\right)\text{cov}\left(\phi_2, f\right)\text{cov}\left(\phi_1, \phi_2\right)\right)$$

$$+ \sigma^2_{\phi_2}\left(\sigma^4_{\phi_1}\text{cov}^2\left(\phi_2, f\right) + \text{cov}^2\left(\phi_1, \phi_2\right)\text{cov}^2\left(\phi_1, f\right) - 2\sigma^2_{\phi_1}\text{cov}\left(\phi_1, f\right)\text{cov}\left(\phi_2, f\right)\text{cov}\left(\phi_1, \phi_2\right)\right)$$

$$+ 2\,\text{cov}\left(\phi_1, \phi_2\right)\left(\sigma^2_{\phi_1}\sigma^2_{\phi_2}\text{cov}\left(\phi_1, f\right)\text{cov}\left(\phi_2, f\right) - \sigma^2_{\phi_2}\text{cov}^2\left(\phi_1, f\right)\text{cov}\left(\phi_1, \phi_2\right)\right.$$

$$\left. + \text{cov}^2\left(\phi_1, \phi_2\right)\text{cov}\left(\phi_1, f\right)\text{cov}\left(\phi_2, f\right) - \sigma^2_{\phi_1}\text{cov}^2\left(\phi_2, f\right)\text{cov}\left(\phi_1, \phi_2\right)\right)$$

$$= \sigma^2_{\phi_1}\sigma^4_{\phi_2}\text{cov}^2\left(\phi_1, f\right) + \sigma^4_{\phi_1}\sigma^2_{\phi_2}\text{cov}^2\left(\phi_2, f\right) - \sigma^2_{\phi_1}\text{cov}^2\left(\phi_2, f\right)\text{cov}^2\left(\phi_1, f\right)$$

$$-\sigma_{\phi_2}^2 \operatorname{cov}^2(\phi_1, f) \operatorname{cov}^2(\phi_1, \phi_2) - 2\sigma_{\phi_1}^2 \sigma_{\phi_2}^2 \operatorname{cov}(\phi_1, f) \operatorname{cov}(\phi_2, f) \operatorname{cov}(\phi_1, \phi_2)$$

$$+2\operatorname{cov}^3(\phi_1, \phi_2) \operatorname{cov}(\phi_1, f) \operatorname{cov}(\phi_2, f)$$

$$=\sigma_{\phi_1}^2 \operatorname{cov}^2(\phi_2, f) \cdot \left[\sigma_{\phi_1}^2 \sigma_{\phi_2}^2 - \operatorname{cov}^2(\phi_1, \phi_2)\right] + \sigma_{\phi_2}^2 \operatorname{cov}^2(\phi_1, f) \cdot \left[\sigma_{\phi_1}^2 \sigma_{\phi_2}^2 - \operatorname{cov}^2(\phi_1, \phi_2)\right]$$

$$-2\operatorname{cov}(\phi_1, f) \operatorname{cov}(\phi_2, f) \operatorname{cov}(\phi_1\phi_2) \cdot \left[\sigma_{\phi_1}^2 \sigma_{\phi_2}^2 - \operatorname{cov}^2(\phi_1, \phi_2)\right]$$

$$= \left(\sigma_{\phi_1}^2 \operatorname{cov}^2(\phi_2, f) + \sigma_{\phi_2}^2 \operatorname{cov}^2(\phi_1, f) - 2\operatorname{cov}(\phi_1, f) \operatorname{cov}(\phi_2, f) \operatorname{cov}(\phi_1, \phi_2)\right) \cdot \left(\sigma_{\phi_1}^2 \sigma_{\phi_2}^2 - \operatorname{cov}^2(\phi_1, \phi_2)\right).$$

The full term is therefore:

$$\frac{\left(\sigma_{\phi_1}^2 \operatorname{cov}^2(\phi_2, f) + \sigma_{\phi_2}^2 \operatorname{cov}^2(\phi_1, f) - 2\operatorname{cov}(\phi_1, f) \operatorname{cov}(\phi_2, f) \operatorname{cov}(\phi_1, \phi_2)\right)}{\left(\sigma_{\phi_1}^2 \sigma_{\phi_2}^2 - \operatorname{cov}^2(\phi_1, \phi_2)\right)^2}$$

$$\times \left(\sigma_{\phi_1}^2 \sigma_{\phi_2}^2 - \operatorname{cov}^2(\phi_1, \phi_2)\right) - \frac{\operatorname{cov}^2(h(\phi_1, \phi_2), f)}{\sigma_{h(\phi_1, \phi_2)}^2} = \Delta_{\text{bias}}^{n \to \infty}.$$

Summing up, the expected deviance is therefore equal to:

$$\frac{2}{\phi}\left(\Delta_{\text{bias}}^{n \to \infty} - \frac{1}{2}(\Delta_{\text{var}}^{n \to \infty} + \Delta_{\text{bias}}^{n \to \infty})\right) = \frac{1}{\phi}\left(\Delta_{\text{bias}}^{n \to \infty} - \Delta_{\text{var}}^{n \to \infty}\right),$$

that is exactly the increase of MSE in terms of increase of bias and reduction of variance due to the aggregation found in the analysis of the NonLinCFA algorithm. $\square$

A second result assuming Gaussianity justifies the choice of centering the second order Taylor expansion in $\boldsymbol{\theta}_0 = 0$ to approximate the function $b(\cdot)$ for any generalized linear model.

**Lemma 6.** *Considering the center in zero ($\boldsymbol{\theta}_0 = 0$), in the Gaussian asymptotic case, the second order Taylor expansion leads to an exact approximation of the function $b(\cdot)$.*

*Proof.* The function $b(\cdot)$ appears in the increase of deviance analysis performed in the previous subsection in the following equation:

$$\mathbb{E}_{x,y,\mathcal{T}}\left[b\left(\hat{w}_1\phi_1 + \hat{w}_2\phi_2\right) - b(\hat{w}h(\phi_1, \phi_2))\right].$$

In the linear case, recalling that $b(\boldsymbol{\theta}) = \frac{\theta^2}{2}$, the expected value under analysis is equal to:

$$\frac{1}{2} \cdot \mathbb{E}_{x,y,D}\left[(\hat{w}_1\phi_1 + \hat{w}_2\phi_2)^2 - (\hat{w}h(\phi_1, \phi_2))^2\right]$$

$$=\frac{1}{2} \cdot \left[\sigma_{\phi_1}^2 \mathbb{E}_D[\hat{w}_1^2] + \sigma_{\phi_2}^2 \mathbb{E}_D[\hat{w}_2^2] + 2\operatorname{cov}(\phi_1, \phi_2)\mathbb{E}_D[\hat{w}_1\hat{w}_2] - \sigma_{h(\phi_1, \phi_2)}^2 \mathbb{E}_D[\hat{w}^2]\right].$$

Moreover, $b''(0) = 1$. Therefore, this quantity is equal to the general expression of the second order Taylor expansion centered in 0 (Equation C). $\square$

## D   Experiments

### D.1   Synthetic experiments

This subsection provides more details on the synthetic experiments introduced in the main paper. The first synthetic problem that we designed is composed of $D = 100$ features and standard deviation of the noise $\sigma = 10$. The first independent variable $x_1$ follows a uniform distribution in the interval $[0, 1]$. Any other feature $x_i$, $i \in \{2, .., 100\}$, is a linear combination between a randomly chosen previous features $x_j$, $j < i$ and a random variable that follows a uniform distribution in the interval $[0, 1]$ (specifically $x_i = 0.7x_j + 0.3u$, $u \sim$

$\mathcal{U}([0, 1])$). The target variable $y$ is finally a linear combination between the $D$ features $x_1, .., x_{100}$, randomly sampling the coefficients from a uniform distribution in $[0, 1]$. Moreover, a Gaussian noise with standard deviation $\sigma$ is added to the target. The same experiment is repeated in a more complex setting, considering $D = 1000$ features and $\sigma = 100$ as standard deviation of the additive noise.

To test the GenLinCFA algorithm in classification settings, the same two experiment described in this section have been repeated applying the sign function to the target, in order to transform the two problems into classification tasks.

In Table 3 and Table 4 it is possible to find the detailed results of the experiments performed respectively with continuous and binary target. As discussed in the main paper, also LinCFA algorithm has been applied to compare the regression results. Since it has no hyperparameters to tune, it is not reported in the tables. In the setting with $D = 100$ features it leads to $d = 39 \pm 1.52$ features and an $R2score = 0.8659 \pm 0.0049$. In the setting with $D = 1000$ features, it produces $d = 193.2 \pm 2.37$ features with an $R2score = 0.7070 \pm 0.0074$.

Table 3: 95% confidence intervals, linear synthetic setting: different hyperparameters, continuous target.

| | **Noise variance $\sigma = 10$, number of features D = 100** | | | | |
|---|---|---|---|---|---|
| | | | NonLinCFA | | |
| Hyperparameter | $\epsilon = 0.01$ | $\epsilon = 0.001$ | $\epsilon = 0.0001$ | $\epsilon = 0.00001$ | $\epsilon = 0.000001$ |
| Number reduced features $d$ | $1.0 \pm 0.0$ | $8.0 \pm 0.88$ | $11.4 \pm 1.45$ | $14.4 \pm 1.28$ | $14.7 \pm 1.21$ |
| $R2$ test score | $0.8655 \pm 0.0051$ | $0.8664 \pm 0.0048$ | $0.8661 \pm 0.0049$ | $0.8659 \pm 0.0050$ | $0.8664 \pm 0.0043$ |
| | | | GenLinCFA | | |
| Hyperparameter | $\epsilon = 0.76$ | $\epsilon = 0.77$ | $\epsilon = 0.78$ | $\epsilon = 0.79$ | $\epsilon = 0.80$ |
| Number reduced features $d$ | $21.6 \pm 1.47$ | $16.6 \pm 0.63$ | $13.6 \pm 1.31$ | $5.4 \pm 1.15$ | $2.0 \pm 0.39$ |
| $R2$ test score | $0.8656 \pm 0.0047$ | $0.8663 \pm 0.0046$ | $0.8660 \pm 0.0046$ | $0.8659 \pm 0.0049$ | $0.8657 \pm 0.0049$ |
| | **Noise variance $\sigma = 100$, number of features D = 1000** | | | | |
| | | | NonLinCFA | | |
| Hyperparameter | $\epsilon = 0.01$ | $\epsilon = 0.001$ | $\epsilon = 0.0001$ | $\epsilon = 0.00001$ | $\epsilon = 0.000001$ |
| Number reduced features $d$ | $1.0 \pm 0.0$ | $3.1 \pm 0.65$ | $18.0 \pm 1.92$ | $21.9 \pm 2.10$ | $22.3 \pm 1.90$ |
| $R2$ test score | $0.7332 \pm 0.0069$ | $0.7318 \pm 0.0071$ | $0.7274 \pm 0.0079$ | $0.7265 \pm 0.0072$ | $0.7267 \pm 0.0076$ |
| | | | GenLinCFA | | |
| Hyperparameter | $\epsilon = 0.76$ | $\epsilon = 0.77$ | $\epsilon = 0.78$ | $\epsilon = 0.79$ | $\epsilon = 0.80$ |
| Number reduced features $d$ | $7.3 \pm 1.08$ | $3.4 \pm 0.63$ | $1.0 \pm 0.0$ | $1.0 \pm 0.0$ | $1.0 \pm 0.0$ |
| $R2$ test score | $0.7326 \pm 0.0069$ | $0.7325 \pm 0.0072$ | $0.7332 \pm 0.0069$ | $0.7332 \pm 0.0069$ | $0.7332 \pm 0.0069$ |

Table 4: 95% confidence intervals, linear synthetic setting: different hyperparameters, binary target.

| | **Noise variance $\sigma = 10$, number of features D = 100** | | | | |
|---|---|---|---|---|---|
| | | | GenLinCFA | | |
| Hyperparameter | $\epsilon = 0.71$ | $\epsilon = 0.72$ | $\epsilon = 0.73$ | $\epsilon = 0.75$ | $\epsilon = 0.77$ |
| Number reduced features $d$ | $25.2 \pm 1.59$ | $19.4 \pm 1.69$ | $15.6 \pm 1.39$ | $4.3 \pm 1.21$ | $1.0 \pm 0.0$ |
| *Accuracy* test score | $0.8928 \pm 0.0064$ | $0.8947 \pm 0.0066$ | $0.8956 \pm 0.0065$ | $0.8958 \pm 0.0069$ | $0.8975 \pm 0.0054$ |
| | **Noise variance $\sigma = 100$, number of features D = 1000** | | | | |
| | | | GenLinCFA | | |
| Hyperparameter | $\epsilon = 0.71$ | $\epsilon = 0.72$ | $\epsilon = 0.73$ | $\epsilon = 0.75$ | $\epsilon = 0.77$ |
| Number reduced features $d$ | $20.0 \pm 3.54$ | $11.1 \pm 2.06$ | $5.7 \pm 0.88$ | $1.0 \pm 0.0$ | $1.0 \pm 0.0$ |
| *Accuracy* test score | $0.8462 \pm 0.0067$ | $0.8453 \pm 0.0075$ | $0.8429 \pm 0.0064$ | $0.8520 \pm 0.0048$ | $0.8520 \pm 0.0048$ |

The two regression and the two classification synthetic experiments that have been described in this section have been repeated considering a nonlinear relationship between the features and the target. In particular, the datasets have been generated exactly in the same way, with the only difference to define the target variable as a linear combination of the squared value of the input features, with additive Gaussian noise. In this setting, a wrapper feature selection has been again considered as baseline, considering the squared of the inputs as candidate features to select. NonLinCFA and GenLinCFA have been applied in two different ways: considering as features the squared values of the original features ($\phi_i(x) = x_i^2$) and selecting the mean as aggregation function, or considering the original features as inputs ($\phi_i(x) = x_i$) and performing the

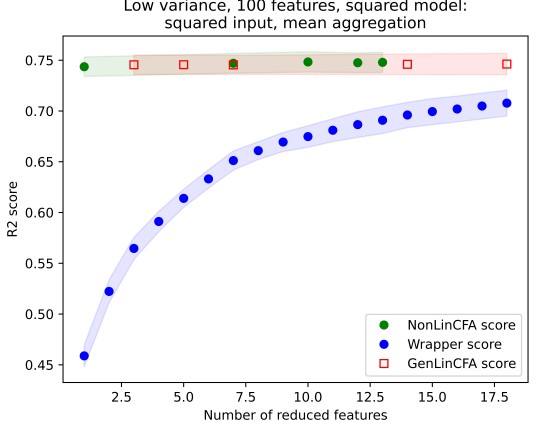

(a) Results considering standard deviation $\sigma = 10$, $D = 100$ features, squared inputs and mean aggregations.

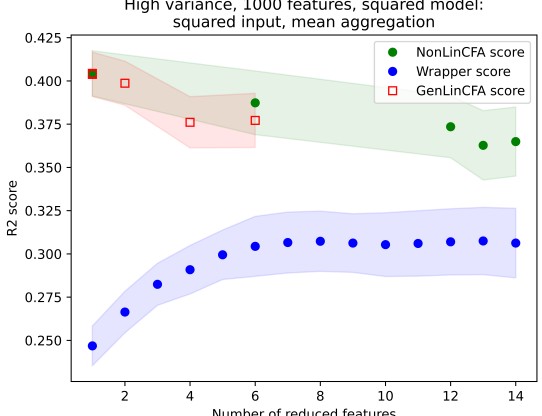

(b) Results considering standard deviation $\sigma = 100$, $D = 1000$ features, squared inputs and mean aggregations.

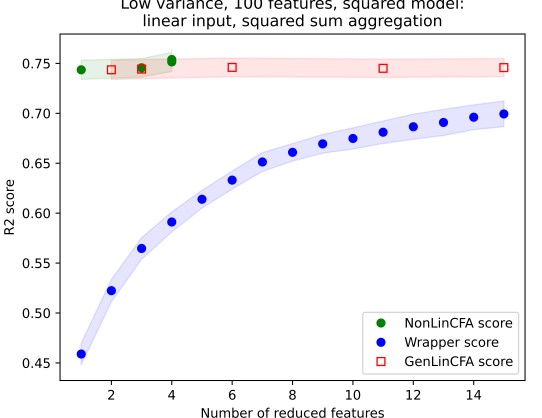

(c) Results considering standard deviation $\sigma = 10$, $D = 100$ features, linear inputs and sum of square aggregations.

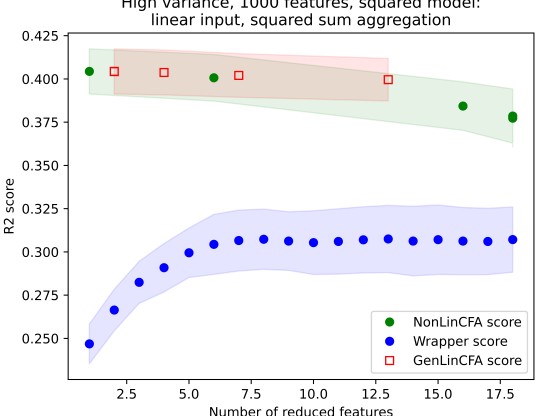

(d) Results considering standard deviation $\sigma = 100$, $D = 1000$ features, linear inputs and sum of square aggregations.

Figure 2: Application of the two proposed algorithm. Test performances and number of reduced features considering different hyperparameters, transformations of features and aggregation functions in a quadratic setting.

squared sum as aggregation function. In this way, the two methods have been tested considering nonlinear transformations of features or nonlinear aggregation functions.

Figure 2 shows the results in terms of coefficient of determination for the two regression problems with low and high number of features and variance. In all the four cases the hyperparaneter $\epsilon$ of NonLinCFA has been considered in the range $\{0.01, 0.001, 0.0001, 1e - 05, 1e - 06\}$. Moreover, for GenLinCFA it has been considered in the range $\{0.68, 0.7, 0.72, 0.74, 0.76\}$. All experiments have been repeated ten times to produce confidence intervals. Figure 2a and Figure 2c show the test regression performance considering respectively squared input features and the mean and aggregation function and linear input features and the squared sum as aggregation function, in the low dimensional and low variance scenario. The same results obtained with the same hyperparameters are reported in Figure 2b and Figure 2d for the high dimensional with high variance regression setting. As already discussed in the main paper for linear settings, the linear regression applied on the features aggregated by the two algorithms perform better than the linear regression performed on the features selected by the wrapper approach, that needs more features to obtain similar performances. Moreover, it is possible to observe that GenLinCFA is more prone to aggregate features in

high dimensional and noisy problems. On the contrary, NonLinCFA, is less prone to aggregate the features in high dimensional and noisy problems. Finally, considering linear inputs and squared sum aggregation or squared inputs and linear aggregations the two algorithms have similar performances, showing that they can be applied in both contexts. As a further comparison, LinCFA algorithm has also been applied, considering the square of the features as inputs. The linear regression on the reduced features have similar performances in the two settings, with a test score respectively of $0.7503 \pm 0.0080$ and $0.3619 \pm 0.0186$ for the low and high dimensional problems, but a larger number of reduced features ($d = 34.3 \pm 0.91$ and $d = 120.5 \pm 1.69$).

To further analyse GenLinCFA algorithm in classification contexts, the two experiments that consider a quadratic relationship between the features and the target have been repeated in classification, applying the sign function to the target. Again, mean aggregations of the squared features and squared sums of the features have been considered. The accuracy scores and number of reduced features are reported in Figure 3. The figure compares again the performance of the algorithm with a wrapper feature selection, logistic regression is the supervised model applied to produce the scores. Again, the performances of the proposed algorithm are similar or better w.r.t. the wrapper baseline and, considering the same values of hiperparameters in the two settings, the algorithm is more prone to aggregate in the more complex and noisy setting. The detailed results and confidence intervals for different hyperparameters that have been discussed in this setting can be found in Table 5 and Table 6, respectively for regression and classification.

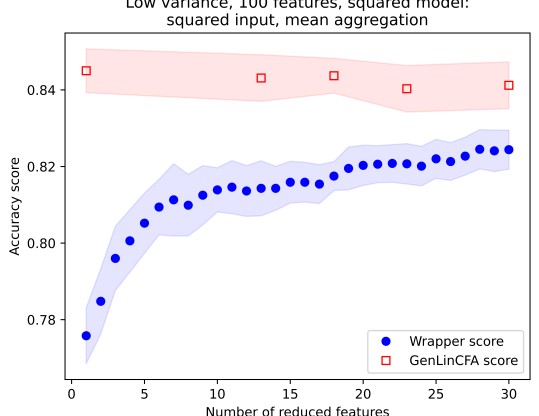

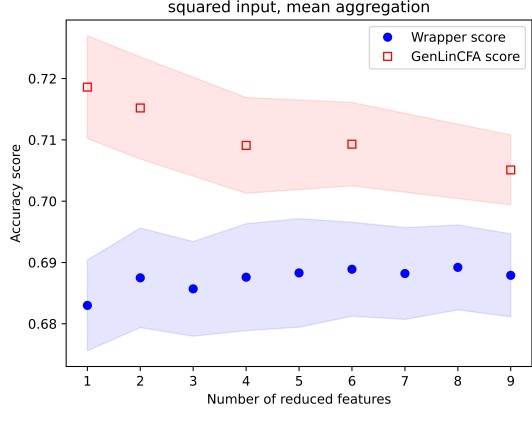

(a) Classification results with standard deviation $\sigma = 10$, $D = 100$ features, squared inputs and mean aggregations.

(b) Classification results with standard deviation $\sigma = 100$, $D = 1000$ features, squared inputs, mean aggregations.

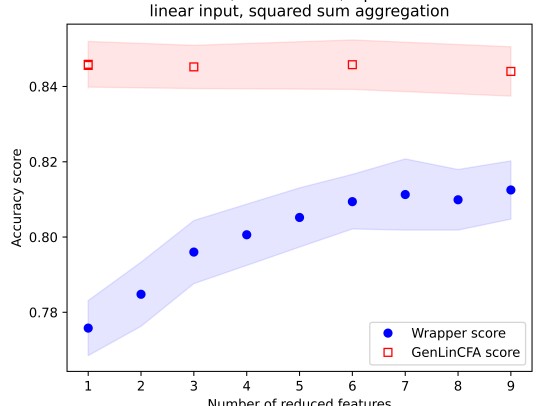

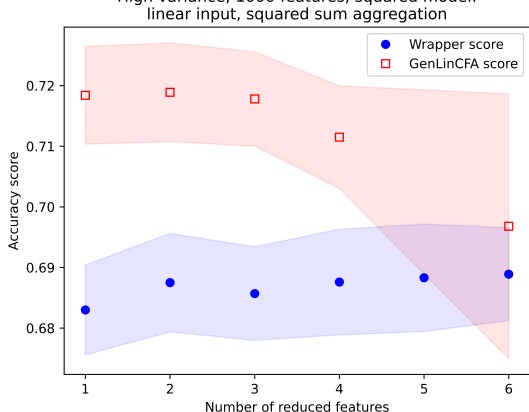

(c) Classification results with standard deviation $\sigma = 10$, $D = 100$ features, linear inputs, sum of square aggregations.

(d) Classification results with standard deviation $\sigma = 100$, $D = 1000$ features, linear inputs, sum of square aggregations.

Figure 3: Application of GenLinCFA in classification. Test performances and number of reduced features considering different hyperparameters, transformations of features and aggregation functions.

### D.1.1 Real-World experiments

Table 7 and Table 8 report the confidence intervals associated to five different repetitions of the experiments, in terms of coefficient of determination for regression and accuracy for classification with linear and logistic regression. The same results are reported in Table 9 and Table 10, considering support vector machines for regression and classification as supervised learning methods. The *Finance* dataset has been retrieved from Kaggle (https://www.kaggle.com/datasets/dgawlik/nyse). The *Bankruptcy* and *Parkinson* datasets have been retrieved from the UCI ML repository (https://archive.ics.uci.edu/ml/datasets/Parkinson%27s+Disease+Classification, https://archive.ics.uci.edu/ml/datasets/Polish+companies+bankruptcy+data). To further test the proposed algorithms on a dataset with a huge number of features and a small number of samples, also a gene expression dataset from the UCI ML repository has been considered (https://archive.ics.uci.edu/dataset/401/gene+expression+cancer+rna+seq.)

Table 5: 95% confidence intervals for quadratic synthetic experiments: different hyperparameters, continuous target.

| | | | | | |
|---|---|---|---|---|---|
| **Noise variance $\sigma = 10$, number of features D $= 100$** | | | | | |
| NonLinCFA, squared input, mean aggregation | | | | | |
| Hyperparameter | $\epsilon = 0.01$ | $\epsilon = 0.001$ | $\epsilon = 0.0001$ | $\epsilon = 0.00001$ | $\epsilon = 0.000001$ |
| Number reduced features $d$ | $1.0 \pm 0.0$ | $7.1 \pm 0.85$ | $10.3 \pm 1.64$ | $12.9 \pm 1.58$ | $12.4 \pm 1.39$ |
| $R2$ test score | $0.7436 \pm 0.0096$ | $0.7469 \pm 0.0097$ | $0.7483 \pm 0.0099$ | $0.7479 \pm 0.0099$ | $0.7475 \pm 0.0096$ |
| GenLinCFA, squared input, mean aggregation | | | | | |
| Hyperparameter | $\epsilon = 0.68$ | $\epsilon = 0.70$ | $\epsilon = 0.72$ | $\epsilon = 0.74$ | $\epsilon = 0.76$ |
| Number reduced features $d$ | $17.6 \pm 0.85$ | $14.5 \pm 1.64$ | $7.2 \pm 1.58$ | $4.9 \pm 1.39$ | $3.0 \pm 1.39$ |
| $R2$ test score | $0.7462 \pm 0.0104$ | $0.7459 \pm 0.0101$ | $0.7453 \pm 0.0093$ | $0.7455 \pm 0.0094$ | $0.7455 \pm 0.0096$ |
| NonLinCFA, linear input, square sum aggregation | | | | | |
| Hyperparameter | $\epsilon = 0.01$ | $\epsilon = 0.001$ | $\epsilon = 0.0001$ | $\epsilon = 0.00001$ | $\epsilon = 0.000001$ |
| Number reduced features $d$ | $1.0 \pm 0.0$ | $2.8 \pm 0.46$ | $3.8 \pm 1.37$ | $4.1 \pm 1.43$ | $4.1 \pm 1.43$ |
| $R2$ test score | $0.7457 \pm 0.0090$ | $0.7450 \pm 0.0091$ | $0.7460 \pm 0.0094$ | $0.7442 \pm 0.0096$ | $0.7435 \pm 0.0096$ |
| GenLinCFA, linear input, square sum aggregation | | | | | |
| Hyperparameter | $\epsilon = 0.68$ | $\epsilon = 0.70$ | $\epsilon = 0.72$ | $\epsilon = 0.74$ | $\epsilon = 0.76$ |
| Number reduced features $d$ | $15.4 \pm 0.46$ | $10.9 \pm 0.37$ | $6.3 \pm 0.41$ | $3.3 \pm 0.43$ | $2.1 \pm 0.43$ |
| $R2$ test score | $0.7462 \pm 0.0104$ | $0.7459 \pm 0.0101$ | $0.7453 \pm 0.0093$ | $0.7455 \pm 0.0094$ | $0.7455 \pm 0.0096$ |
| **Noise variance $\sigma = 100$, number of features D $= 1000$** | | | | | |
| NonLinCFA, squared input, mean aggregation | | | | | |
| Hyperparameter | $\epsilon = 0.01$ | $\epsilon = 0.001$ | $\epsilon = 0.0001$ | $\epsilon = 0.00001$ | $\epsilon = 0.000001$ |
| Number reduced features $d$ | $1.0 \pm 0.0$ | $5.9 \pm 0.75$ | $11.6 \pm 0.69$ | $13.2 \pm 0.61$ | $13.7 \pm 0.62$ |
| $R2$ test score | $0.4044 \pm 0.0130$ | $0.3873 \pm 0.0184$ | $0.3735 \pm 0.0179$ | $0.3627 \pm 0.0201$ | $0.3649 \pm 0.0199$ |
| GenLinCFA, squared input, mean aggregation | | | | | |
| Hyperparameter | $\epsilon = 0.68$ | $\epsilon = 0.70$ | $\epsilon = 0.72$ | $\epsilon = 0.74$ | $\epsilon = 0.76$ |
| Number reduced features $d$ | $5.5 \pm 0.76$ | $3.5 \pm 0.69$ | $2.0 \pm 0.60$ | $1.2 \pm 0.63$ | $1.1 \pm 0.69$ |
| $R2$ test score | $0.3772 \pm 0.0156$ | $0.3760 \pm 0.0148$ | $0.3986 \pm 0.0127$ | $0.4037 \pm 0.0128$ | $0.4043 \pm 0.0130$ |
| NonLinCFA, linear input, square sum aggregation | | | | | |
| Hyperparameter | $\epsilon = 0.01$ | $\epsilon = 0.001$ | $\epsilon = 0.0001$ | $\epsilon = 0.00001$ | $\epsilon = 0.000001$ |
| Number reduced features $d$ | $1.0 \pm 0.0$ | $6.3 \pm 0.78$ | $15.5 \pm 1.55$ | $17.7 \pm 1.92$ | $18.0 \pm 1.84$ |
| $R2$ test score | $0.4043 \pm 0.0130$ | $0.4006 \pm 0.0134$ | $0.3843 \pm 0.0140$ | $0.3785 \pm 0.0156$ | $0.3772 \pm 0.0166$ |
| GenLinCFA, linear input, square sum aggregation | | | | | |
| Hyperparameter | $\epsilon = 0.68$ | $\epsilon = 0.70$ | $\epsilon = 0.72$ | $\epsilon = 0.74$ | $\epsilon = 0.76$ |
| Number reduced features $d$ | $12.7 \pm 0.78$ | $7.2 \pm 1.55$ | $4.1 \pm 1.92$ | $2.4 \pm 1.83$ | $1.7 \pm 1.61$ |
| $R2$ test score | $0.3995 \pm 0.0122$ | $0.4020 \pm 0.0126$ | $0.4037 \pm 0.0129$ | $0.4043 \pm 0.0130$ | $0.4042 \pm 0.0130$ |

The climate datasets are composed of continuous climatological features and a scalar target which represents the state of vegetation of a basin of Po river. The first one (*Climate, Climate(Class.)*) also considers the state of the vegetation of neighbouring basins as inputs, while the second one is a more difficult problem since it tries to predict it only from temperature and precipitation features. These datasets have been composed by the authors merging different sources for the vegetation index, temperature and precipitation over different basins (see (Didan, 2015; Cornes et al., 2018; Zellner, 2022)), and they are available in the repository of this work.

The two tables show the performances of the NonLinCFA algorithm, considering five different hyperparameters ($\epsilon \in \{0.01, 0.001, 0.0001, 0.00001, 0.000001\}$). The same holds for the GenLinCFA algorithm, where the hyperparameters have been selected in order to show some aggregations that are not too small (everything is aggregated) or is too large (more than 50 reduced features). The proposed algorithms are shown in comparison with the LinCFA method, that they generalize. Additionally, state of the art methods (PCA, LDA, Kernel PCA, Isomap, LLE, Supervised PCA, NCA, UMAP, t-SNE) have been applied to compare the proposed methods, reporting in the related tables the best validation performance between 1 and 50 reduced features, considering default hyperparameters. Given the huge amount of features, for the *GeneExpression* dataset, the number of reduced components tested are the following set: $\{5, 10, 15, 20, 25, 30, 40, 50, 100, 200\}$. Additionally, Autoencoders have been also considered as baselines, testing lantent space dimensions in the following set: $\{2, 4, 8, 16, 32, 64\}$ (and also considering $\{128, 256\}$ for the *GeneExpression* dataset). The au-

Table 6: 95% confidence intervals for quadratic synthetic experiments: different hyperparameters, binary target.

| | **Noise variance $\sigma = 10$, number of features D $= 100$** | | | | |
|---|---|---|---|---|---|
| | GenLinCFA, squared input, mean aggregation | | | | |
| Hyperparameter | $\epsilon = 0.77$ | $\epsilon = 0.79$ | $\epsilon = 0.81$ | $\epsilon = 0.85$ | $\epsilon = 0.95$ |
| Number reduced features $d$ | $30.2 \pm 1.97$ | $22.9 \pm 2.01$ | $18.5 \pm 0.84$ | $13.0 \pm 1.17$ | $1.0 \pm 0.0$ |
| *Accuracy* test score | $0.8412 \pm 0.0061$ | $0.8403 \pm 0.0060$ | $0.8436 \pm 0.0045$ | $0.8431 \pm 0.0061$ | $0.8450 \pm 0.0057$ |
| | GenLinCFA, linear input, square sum aggregation | | | | |
| Hyperparameter | $\epsilon = 0.77$ | $\epsilon = 0.79$ | $\epsilon = 0.81$ | $\epsilon = 0.85$ | $\epsilon = 0.95$ |
| Number reduced features $d$ | $8.8 \pm 1.38$ | $5.5 \pm 1.57$ | $3.2 \pm 1.03$ | $1.4 \pm 0.41$ | $1.0 \pm 0.0$ |
| *Accuracy* test score | $0.8440 \pm 0.0065$ | $0.8457 \pm 0.0066$ | $0.8452 \pm 0.0057$ | $0.8459 \pm 0.0061$ | $0.8455 \pm 0.0060$ |
| | **Noise variance $\sigma = 100$, number of features D $= 1000$** | | | | |
| | GenLinCFA, squared input, mean aggregation | | | | |
| Hyperparameter | $\epsilon = 0.77$ | $\epsilon = 0.79$ | $\epsilon = 0.81$ | $\epsilon = 0.85$ | $\epsilon = 0.95$ |
| Number reduced features $d$ | $9.4 \pm 1.15$ | $5.9 \pm 0.65$ | $3.5 \pm 0.50$ | $1.6 \pm 0.41$ | $1.0 \pm 0.0$ |
| *Accuracy* test score | $0.7051 \pm 0.0056$ | $0.7093 \pm 0.0068$ | $0.7091 \pm 0.0078$ | $0.7152 \pm 0.0083$ | $0.7186 \pm 0.0083$ |
| | GenLinCFA, linear input, square sum aggregation | | | | |
| Hyperparameter | $\epsilon = 0.77$ | $\epsilon = 0.79$ | $\epsilon = 0.81$ | $\epsilon = 0.85$ | $\epsilon = 0.95$ |
| Number reduced features $d$ | $6.1 \pm 2.22$ | $3.7 \pm 1.33$ | $2.6 \pm 0.63$ | $1.6 \pm 0.30$ | $1.3 \pm 0.28$ |
| *Accuracy* test score | $0.6968 \pm 0.0218$ | $0.7114 \pm 0.0084$ | $0.7178 \pm 0.0077$ | $0.7189 \pm 0.0081$ | $0.7184 \pm 0.0080$ |

toencoders considered have two hidden layers both in the encoder and in the decoder, respectively with four and two times the number of nodes w.r.t. the latent space dimension (for example, for the architecture with latent dimension 2, the autoencoder is made of five layers of $8, 4, 2, 4, 8$ nodes respectively).

The results, discussed also in the main paper, show better or competitive performances with respect to the baselines. To further inspect the reduced features obtained, considering the climate dataset where the features are temperature and precipitation measurements at different locations and with different time aggregations, Figure 4 reports some interpretable reduced features in practice. Considering five different time aggregations of features related to temperature and precipitation, we applied the NonLinCFA algorithm, constraining the algorithm to aggregate features only if they are at neighboring locations and they represent the same variable with the same time scale. In the figure is reported, represented with the same color, each set of variables that is aggregated with the mean by the algorithm. This is clearly interpretable, since the resulting features are averages of the original features, obtained by the algorithm through partitioning the geographical area of interest into sub-regions, and averaging on them.

Table 7: Experiments on climate datasets. The total number of samples $n$ has been divided into train (66% of data) and test (33% of data) sets.

| Quantity | Climate | Climate (Class.) | Climate II | Climate II (class.) |
|---|---|---|---|---|
| # samples $n$ | 981 | 981 | 867 | 867 |
| # features $D$ | 1991 | 1991 | 2408 | 2408 |
| **Reduced Dimension** | | | | |
| NonLinCFA ($\epsilon = 0.01$) | $7.4 \pm 1.3$ | NA | $7.0 \pm 0.9$ | NA |
| NonLinCFA ($\epsilon = 0.001$) | $\mathbf{16.0 \pm 0.8}$ | NA | $11.4 \pm 0.9$ | NA |
| NonLinCFA ($\epsilon = 0.0001$) | $19.4 \pm 1.4$ | NA | $12.2 \pm 0.6$ | NA |
| NonLinCFA ($\epsilon = 0.00001$) | $19.6 \pm 0.9$ | NA | $12.4 \pm 0.7$ | NA |
| NonLinCFA ($\epsilon = 0.000001$) | $19.6 \pm 1.3$ | NA | $12.4 \pm 0.7$ | NA |
| GenLinCFA ($\epsilon = \epsilon_1$) | $31.6 \pm 10.9$ | $33.75 \pm 11.5$ | $13.6 \pm 4.8$ | $34.0 \pm 6.1$ |
| GenLinCFA ($\epsilon = \epsilon_2$) | $26.0 \pm 9.1$ | $27.0 \pm 8.2$ | $7.2 \pm 1.1$ | $26.0 \pm 6.1$ |
| GenLinCFA ($\epsilon = \epsilon_3$) | $18.6 \pm 5.4$ | $22.3 \pm 5.7$ | $4.6 \pm 0.9$ | $18.7 \pm 5.4$ |
| GenLinCFA ($\epsilon = \epsilon_4$) | $15.0 \pm 4.3$ | $21.3 \pm 5.2$ | $2.4 \pm 0.4$ | $11.0 \pm 4.5$ |
| GenLinCFA ($\epsilon = \epsilon_5$) | $13.8 \pm 3.0$ | $\mathbf{17.5 \pm 3.3}$ | $2 \pm 0$ | $2.5 \pm 0.4$ |
| LinCFA | $38.2 \pm 1.6$ | NA | $222.0 \pm 2.7$ | NA |
| PCA | $18.2 \pm 0.7$ | $18.2 \pm 0.7$ | $29.4 \pm 0.1$ | $29.4 \pm 0.4$ |
| LDA | NA | 1 | NA | 1 |
| Kernel PCA | $35.6 \pm 7.9$ | $27.0 \pm 4.8$ | $\mathbf{21.8 \pm 9.5}$ | $\mathbf{11.2 \pm 2.3}$ |
| Isomap | $2.6 \pm 0.7$ | $12.6 \pm 10.8$ | $42.4 \pm 13.3$ | $16.6 \pm 9.2$ |
| LLE | $15.4 \pm 13.8$ | $28.2 \pm 12.9$ | $35.0 \pm 10.7$ | $25.8 \pm 13.1$ |
| Supervised PCA | $41.8 \pm 2.5$ | $37.0 \pm 6.6$ | $7.4 \pm 3.1$ | $13.8 \pm 4.9$ |
| NCA | NA | $24.2 \pm 2.9$ | NA | $7.0 \pm 0.6$ |
| UMAP | $14.6 \pm 5.6$ | $19.6 \pm 16.2$ | $19.5 \pm 11.2$ | $26.8 \pm 13.1$ |
| t-SNE | $2.2 \pm 1.2$ | $12.1 \pm 11.9$ | $7.8 \pm 1.7$ | $10.4 \pm 3.3$ |
| Autoencoders | $12.0 \pm 9.8$ | $30.8 \pm 24.1$ | $9.6 \pm 4.7$ | $10.4 \pm 4.2$ |
| **Test performance** | $\mathbf{R^2}$ **score** | **Accuracy score** | $\mathbf{R^2}$ **score** | **Accuracy score** |
| NonLinCFA ($\epsilon = 0.01$) | $0.8524 \pm 0.0407$ | NA | $0.2949 \pm 0.0156$ | NA |
| NonLinCFA ($\epsilon = 0.001$) | $\mathbf{0.9395 \pm 0.0125}$ | NA | $0.2547 \pm 0.0182$ | NA |
| NonLinCFA ($\epsilon = 0.0001$) | $0.9124 \pm 0.0055$ | NA | $0.2541 \pm 0.0121$ | NA |
| NonLinCFA ($\epsilon = 0.00001$) | $0.9113 \pm 0.0056$ | NA | $0.2529 \pm 0.0148$ | NA |
| NonLinCFA ($\epsilon = 0.000001$) | $0.9121 \pm 0.0047$ | NA | $0.2530 \pm 0.0146$ | NA |
| GenLinCFA ($\epsilon = \epsilon_1$) | $0.9194 \pm 0.0039$ | $0.9068 \pm 0.0037$ | $0.2439 \pm 0.0246$ | $0.6820 \pm 0.0199$ |
| GenLinCFA ($\epsilon = \epsilon_2$) | $0.9226 \pm 0.0026$ | $0.9075 \pm 0.0029$ | $0.2841 \pm 0.0051$ | $0.7127 \pm 0.0159$ |
| GenLinCFA ($\epsilon = \epsilon_3$) | $0.9207 \pm 0.0052$ | $0.9056 \pm 0.0039$ | $0.2764 \pm 0.0069$ | $0.7094 \pm 0.0181$ |
| GenLinCFA ($\epsilon = \epsilon_4$) | $0.9269 \pm 0.0012$ | $0.9062 \pm 0.0036$ | $0.2493 \pm 0.0125$ | $0.7061 \pm 0.0105$ |
| GenLinCFA ($\epsilon = \epsilon_5$) | $0.9275 \pm 0.0004$ | $\mathbf{0.9107 \pm 0.0022}$ | $0.2269 \pm 0.0157$ | $0.6776 \pm 0.0159$ |
| LinCFA | $0.9007 \pm 0.031$ | NA | $-1.2861 \pm 0.2322$ | NA |
| PCA | $0.7536 \pm 0.019$ | $0.8515 \pm 0.0054$ | $0.1917 \pm 0.0395$ | $0.6868 \pm 0.0147$ |
| LDA | NA | $0.7357 \pm 0.0188$ | NA | $0.5526 \pm 0.0224$ |
| Kernel PCA | $0.7990 \pm 0.0061$ | $0.8698 \pm 0.0081$ | $\mathbf{0.3889 \pm 0.0199}$ | $\mathbf{0.7640 \pm 0.0062}$ |
| Isomap | $0.1354 \pm 0.0118$ | $0.6443 \pm 0.0041$ | $0.3216 \pm 0.0146$ | $0.7360 \pm 0.0145$ |
| LLE | $0.1149 \pm 0.0139$ | $0.6444 \pm 0.0052$ | $0.3102 \pm 0.0367$ | $0.7456 \pm 0.0140$ |
| Supervised PCA | $0.8454 \pm 0.0049$ | $0.8827 \pm 0.0098$ | $0.3835 \pm 0.0230$ | $0.7482 \pm 0.0067$ |
| NCA | NA | $0.8776 \pm 0.0086$ | NA | $0.7638 \pm 0.0051$ |
| UMAP | $0.1307 \pm 0.0125$ | $0.6540 \pm 0.0118$ | $0.2998 \pm 0.0107$ | $0.7008 \pm 0.0078$ |
| t-SNE | $0.1366 \pm 0.0193$ | $0.6791 \pm 0.0508$ | $0.3465 \pm 0.0171$ | $0.7403 \pm 0.0028$ |
| Autoencoders | $0.1363 \pm 0.0234$ | $0.6515 \pm 0.0108$ | $0.3637 \pm 0.0043$ | $0.7280 \pm 0.0139$ |
| Full | $0.7429 \pm 0.0228$ | $0.8428 \pm 0.0128$ | $-4.3511 \pm 0.9161$ | $0.6429 \pm 0.0205$ |

Table 8: Experiments on real world datasets. The total number of samples $n$ has been divided into train (66% of data) and test (33% of data) sets.

| Quantity | Finance | Bankruptcy (class.) | Parkinson (class.) | GeneExpression |
|---|---|---|---|---|
| # samples $n$ | 1299 | 1084 | 384 | 801 |
| # features $D$ | 75 | 65 | 753 | 19966 |
| **Reduced Dimension** | | | | |
| NonLinCFA ($\epsilon = 0.01$) | $5.6 \pm 0.4$ | NA | NA | $14.4 \pm 0.9$ |
| NonLinCFA ($\epsilon = 0.001$) | $7.2 \pm 0.7$ | NA | NA | $31.0 \pm 1.8$ |
| NonLinCFA ($\epsilon = 0.0001$) | $7.4 \pm 0.4$ | NA | NA | $37.0 \pm 1.6$ |
| NonLinCFA ($\epsilon = 0.00001$) | $7.4 \pm 0.4$ | NA | NA | $37.8 \pm 1.3$ |
| NonLinCFA ($\epsilon = 0.000001$) | $\mathbf{7.4 \pm 0.4}$ | NA | NA | $39.8 \pm 0.9$ |
| GenLinCFA ($\epsilon = \epsilon_1$) | $15.2 \pm 1.2$ | $27.6 \pm 5.6$ | $53.4 \pm 5.9$ | $\mathbf{35.4 \pm 4.1}$ |
| GenLinCFA ($\epsilon = \epsilon_2$) | $8.0 \pm 1.4$ | $16.4 \pm 3.4$ | $26.4 \pm 1.2$ | $23.6 \pm 2.5$ |
| GenLinCFA ($\epsilon = \epsilon_3$) | $4.8 \pm 0.4$ | $9.8 \pm 1.5$ | $\mathbf{23.4 \pm 1.1}$ | $17.4 \pm 2.3$ |
| GenLinCFA ($\epsilon = \epsilon_4$) | $4.2 \pm 0.4$ | $8.2 \pm 0.4$ | $20.2 \pm 1.5$ | $11.6 \pm 1.8$ |
| GenLinCFA ($\epsilon = \epsilon_5$) | $3.8 \pm 0.3$ | $7.0 \pm 0.6$ | $14.2 \pm 1.3$ | $7.6 \pm 0.8$ |
| LinCFA | $11.4 \pm 0.7$ | NA | NA | $138.7 \pm 11.5$ |
| PCA | $26.6 \pm 0.4$ | $4.8 \pm 0.9$ | $77.6 \pm 3.8$ | 200 |
| LDA | NA | 1 | 1 | NA |
| Kernel PCA | $36.0 \pm 10.8$ | $13.8 \pm 5.1$ | $32.4 \pm 7.7$ | $180.0 + -35.1$ |
| Isomap | $27.2 \pm 6.8$ | $21.2 \pm 9.8$ | $14.8 \pm 9.3$ | $20.0 \pm 7.3$ |
| LLE | $44.0 \pm 7.3$ | 1 | $39.6 \pm 4.6$ | $180.0 \pm 35.1$ |
| Supervised PCA | $31.0 \pm 13.8$ | $16.2 \pm 7.5$ | $21.7 \pm 3.4$ | $19.3 \pm 7.4$ |
| NCA | NA | $\mathbf{12.0 \pm 9.5}$ | $28.2 \pm 15.3$ | NA |
| UMAP | $44.8 \pm 4.2$ | $4.6 \pm 3.4$ | $26.2 \pm 13.9$ | $41.0 \pm 9.6$ |
| t-SNE | $2.6 \pm 2.1$ | $1.6 \pm 0.7$ | $6.4 \pm 3.1$ | $8.8 \pm 3.8$ |
| Autoencoders | $40.0 \pm 18.8$ | $51.2 \pm 13.7$ | $17.2 \pm 11.3$ | $57.6 \pm 11.2$ |
| **Test performance** | $\mathbf{R^2}$ **score** | **Accuracy score** | **Accuracy score** | |
| NonLinCFA ($\epsilon = 0.01$) | $0.8131 \pm 0.0032$ | NA | NA | $0.6089 \pm 0.0220$ |
| NonLinCFA ($\epsilon = 0.001$) | $0.8061 \pm 0.0076$ | NA | NA | $0.5902 \pm 0.0157$ |
| NonLinCFA ($\epsilon = 0.0001$) | $0.8133 \pm 0.0037$ | NA | NA | $0.5750 \pm 0.0155$ |
| NonLinCFA ($\epsilon = 0.00001$) | $0.8133 \pm 0.0037$ | NA | NA | $0.5716 \pm 0.0105$ |
| NonLinCFA ($\epsilon = 0.000001$) | $\mathbf{0.8136 \pm 0.0036}$ | NA | NA | $0.5629 \pm 0.0198$ |
| GenLinCFA ($\epsilon = \epsilon_1$) | $0.8104 \pm 0.0015$ | $0.7503 \pm 0.0012$ | $0.7984 \pm 0.0112$ | $\mathbf{0.6209 \pm 0.0210}$ |
| GenLinCFA ($\epsilon = \epsilon_2$) | $0.8119 \pm 0.0010$ | $0.7480 \pm 0.0018$ | $0.7647 \pm 0.0130$ | $0.6168 \pm 0.0213$ |
| GenLinCFA ($\epsilon = \epsilon_3$) | $0.8101 \pm 0.0003$ | $0.7430 \pm 0.0044$ | $\mathbf{0.8016 \pm 0.0069}$ | $0.6018 \pm 0.0371$ |
| GenLinCFA ($\epsilon = \epsilon_4$) | $0.8114 \pm 0.0003$ | $0.7413 \pm 0.0025$ | $0.7808 \pm 0.0071$ | $0.5801 \pm 0.0349$ |
| GenLinCFA ($\epsilon = \epsilon_5$) | $0.8107 \pm 0.0009$ | $0.7408 \pm 0.0024$ | $0.7712 \pm 0.0095$ | $0.4848 \pm 0.0890$ |
| LinCFA | $0.8010 \pm 0.0128$ | NA | NA | $0.3153 \pm 0.1033$ |
| PCA | $0.7559 \pm 0.0027$ | $0.7413 \pm 0.0037$ | $0.7840 \pm 0.0117$ | $0.5380 \pm 0.0099$ |
| LDA | NA | $0.7587 \pm 0.0065$ | $0.7632 \pm 0.0489$ | NA |
| Kernel PCA | $0.7764 \pm 0.0118$ | $0.7592 \pm 0.0061$ | $0.7920 \pm 0.0177$ | $0.5353 \pm 0.0095$ |
| Isomap | $0.2610 \pm .0458$ | $0.7463 \pm 0.0039$ | $0.7856 \pm 0.0112$ | $0.2307 \pm 0.0586$ |
| LLE | $0.7281 \pm 0.0267$ | $0.7402 \pm 0$ | $0.7696 \pm 0.0525$ | $0.3424 \pm 0.0116$ |
| Supervised PCA | $0.7731 \pm 0.0126$ | $0.7508 \pm 0.0018$ | $0.7913 \pm 0.0069$ | $0.5639 \pm 0.0266$ |
| NCA | NA | $\mathbf{0.7637 \pm 0.0079}$ | $0.7952 \pm 0.0181$ | NA |
| UMAP | $0.6268 \pm 0.0248$ | $0.7385 \pm 0.0059$ | $0.7535 \pm 0.0240$ | $0.2798 \pm 0.0090$ |
| t-SNE | $0.3258 \pm 0.0167$ | $0.7335 \pm 0.0057$ | $0.7504 \pm 0.0174$ | $0.2291 \pm 0.1432$ |
| Autoencoders | $0.5854 \pm 0.0330$ | $0.7435 \pm 0.0039$ | $0.7440 \pm 0.0062$ | $0.3922 \pm 0.0173$ |
| Full | $-5.9514 \pm 3.7166$ | $0.7446 \pm 0.0033$ | $0.7520 \pm 0.0258$ | $0.5385 \pm 0.0170$ |

Table 9: Experiments on climate datasets with Support Vector Machines. The total number of samples $n$ has been divided into train (66% of data) and test (33% of data) sets.

| Quantity | Climate | Climate (Class.) | Climate II | Climate II (class.) |
|---|---|---|---|---|
| # samples $n$ | 981 | 981 | 867 | 867 |
| # features $D$ | 1991 | 1991 | 2408 | 2408 |
| **Test performance** | $\mathbf{R^2}$ **score** | **Accuracy score** | $\mathbf{R^2}$ **score** | **Accuracy score** |
| NonLinCFA ($\epsilon = 0.01$) | $0.8741 \pm 0.0137$ | NA | $0.3045 \pm 0.0117$ | NA |
| NonLinCFA ($\epsilon = 0.001$) | $0.8649 \pm 0.0135$ | NA | $0.3027 \pm 0.0263$ | NA |
| NonLinCFA ($\epsilon = 0.0001$) | $0.8587 \pm 0.0084$ | NA | $0.2869 \pm 0.0211$ | NA |
| NonLinCFA ($\epsilon = 0.00001$) | $0.8623 \pm 0.0058$ | NA | $0.3004 \pm 0.0242$ | NA |
| NonLinCFA ($\epsilon = 0.000001$) | $0.8620 \pm 0.0056$ | NA | $0.2902 \pm 0.0333$ | NA |
| GenLinCFA ($\epsilon = \epsilon_1$) | $0.8964 \pm 0.0011$ | $0.8979 \pm 0.0014$ | $0.1932 \pm 0.0501$ | $0.6853 \pm 0.0153$ |
| GenLinCFA ($\epsilon = \epsilon_2$) | $0.8974 \pm 0.0012$ | $0.8954 \pm 0.0031$ | $0.2129 \pm 0.0470$ | $0.6721 \pm 0.0336$ |
| GenLinCFA ($\epsilon = \epsilon_3$) | $\mathbf{0.8988 \pm 0.0017}$ | $0.8989 \pm 0.0017$ | $0.2253 \pm 0.0487$ | $0.6601 \pm 0.0559$ |
| GenLinCFA ($\epsilon = \epsilon_4$) | $0.8975 \pm 0.0012$ | $0.8990 \pm 0.0010$ | $0.2422 \pm 0.0541$ | $0.6842 \pm 0.0286$ |
| GenLinCFA ($\epsilon = \epsilon_5$) | $0.8986 \pm 0.0024$ | $\mathbf{0.9015 \pm 0.0010}$ | $0.2681 \pm 0.0644$ | $0.6557 \pm 0.0144$ |
| LinCFA | $0.8566 \pm 0.0031$ | NA | $-2.3201 \pm 0.5375$ | NA |
| PCA | $0.5267 \pm 0.0292$ | $0.7137 \pm 0.0126$ | $0.3018 \pm 0.0257$ | $0.7245 \pm 0.0089$ |
| LDA | NA | $0.7301 \pm 0.0174$ | NA | $0.5359 \pm 0.0196$ |
| Kernel PCA | $0.5513 \pm 0.0150$ | $0.8698 \pm 0.0081$ | $0.3888 \pm 0.0199$ | $\mathbf{0.7760 \pm 0.0061}$ |
| Isomap | $0.1202 \pm 0.0067$ | $0.6428 \pm 0.0080$ | $0.2905 \pm 0.0230$ | $0.7192 \pm 0.0105$ |
| LLE | $0.1225 \pm 0.0192$ | $0.6454 \pm 0.0073$ | $0.3085 \pm 0.0229$ | $0.7350 \pm 0.0052$ |
| Supervised PCA | $0.5137 \pm 0.0091$ | $0.8212 \pm 0.0125$ | $0.3721 \pm 0.0148$ | $0.7103 \pm 0.0089$ |
| NCA | NA | $0.7689 \pm 0.0150$ | NA | $0.7482 \pm 0.0052$ |
| UMAP | $0.1219 \pm 0.0118$ | $0.6357 \pm 0.0090$ | $0.2890 \pm 0.0086$ | $0.7114 \pm 0.0115$ |
| t-SNE | $0.2036 \pm 0.0735$ | $0.6786 \pm 0.0320$ | $0.3623 \pm 0.0563$ | $0.7423 \pm 0.0270$ |
| Autoencoders | $0.1480 \pm 0.0288$ | $0.6510 \pm 0.0117$ | $\mathbf{0.3892 \pm 0.0223}$ | $0.7325 \pm 0.0114$ |
| Full | $0.5572 \pm 0.0172$ | $0.7261 \pm 0.0167$ | $0.3253 \pm 0.0132$ | $0.7028 \pm 0.0037$ |

Table 10: Experiments on real world datasets with Support Vector Machines. The total number of samples $n$ has been divided into train (66% of data) and test (33% of data) sets.

| Quantity | Finance | Bankruptcy (class.) | Parkinson (class.) | GeneExpression |
|---|---|---|---|---|
| # samples $n$ | 1299 | 1084 | 384 | 801 |
| # features $D$ | 75 | 65 | 753 | 19966 |
| **Test performance** | **$R^2$ score** | **Accuracy score** | **Accuracy score** | |
| NonLinCFA ($\epsilon = 0.01$) | $0.7489 \pm 0.0127$ | NA | NA | $0.5799 \pm 0.0512$ |
| NonLinCFA ($\epsilon = 0.001$) | $0.7558 \pm 0.0091$ | NA | NA | $0.5649 \pm 0.0178$ |
| NonLinCFA ($\epsilon = 0.0001$) | $0.7624 \pm 0.0127$ | NA | NA | $0.5571 \pm 0.0070$ |
| NonLinCFA ($\epsilon = 0.00001$) | $0.7624 \pm 0.0127$ | NA | NA | $0.5589 \pm 0.0033$ |
| NonLinCFA ($\epsilon = 0.000001$) | $0.7626 \pm 0.0124$ | NA | NA | $0.5609 \pm 0.0195$ |
| GenLinCFA ($\epsilon = \epsilon_1$) | $0.7396 \pm 0.0010$ | $0.7486 \pm 0.0010$ | $0.7344 \pm 0.0082$ | $\mathbf{0.5892 \pm 0.0244}$ |
| GenLinCFA ($\epsilon = \epsilon_2$) | $0.7702 \pm 0.0203$ | $0.7474 \pm 0.0012$ | $0.7424 \pm 0.0174$ | $0.5808 \pm 0.0333$ |
| GenLinCFA ($\epsilon = \epsilon_3$) | $0.8079 \pm 0.0073$ | $0.7463 \pm 0.0010$ | $0.7552 \pm 0.0215$ | $0.5517 \pm 0.0425$ |
| GenLinCFA ($\epsilon = \epsilon_4$) | $0.8205 \pm 0.0073$ | $0.7475 \pm 0.0012$ | $0.7376 \pm 0.0240$ | $0.5609 \pm 0.0375$ |
| GenLinCFA ($\epsilon = \epsilon_5$) | $\mathbf{0.8239 \pm 0.0012}$ | $0.7458 \pm 0.0009$ | $0.7375 \pm 0.0210$ | $0.4652 \pm 0.1008$ |
| LinCFA | $0.7780 \pm 0.0060$ | NA | NA | $0.4223 \pm 0.1131$ |
| PCA | $0.7268 \pm 0.0066$ | $0.7508 \pm 0.0039$ | $0.7424 \pm 0.0324$ | $0.5282 \pm 0.0150$ |
| LDA | NA | $0.7598 \pm 0.0087$ | $0.6624 \pm 0.0288$ | NA |
| Kernel PCA | $0.7763 \pm 0.0117$ | $0.7513 \pm 0.0006$ | $0.7360 \pm 0.0496$ | $0.5264 \pm 0.0162$ |
| Isomap | $0.6866 \pm 0.0153$ | $0.7519 \pm 0.0023$ | $0.7104 \pm 0.0374$ | $0.2761 \pm 0.0166$ |
| LLE | $0.6804 \pm 0.0281$ | $0.75139 \pm 0$ | $0.7440 \pm 0.0221$ | $0.3179 \pm 0.0186$ |
| Supervised PCA | $0.7521 \pm 0.0079$ | $0.7520 \pm 0.0011$ | $\mathbf{0.7616 \pm 0.0128}$ | $0.5665 \pm 0.0242$ |
| NCA | NA | $\mathbf{0.7626 \pm 0.0122}$ | $0.7520 \pm 0.0177$ | NA |
| UMAP | $0.5578 \pm 0.0458$ | $0.7218 \pm 0.0070$ | $0.7152 \pm 0.0196$ | $0.1826 \pm 0.0196$ |
| t-SNE | $0.4432 \pm 0.0881$ | $0.7458 \pm 0.0075$ | $0.4768 \pm 0.0361$ | $0.2833 \pm 0.0879$ |
| Autoencoders | $0.7237 \pm 0.0164$ | $0.7514 \pm 0.0008$ | $0.7360 \pm 0.0171$ | $0.4299 \pm 0.0115$ |
| Full | $0.7510 \pm 0.0035$ | $0.7486 \pm 0.0067$ | $0.7552 \pm 0.0130$ | $0.4902 \pm 0.0130$ |

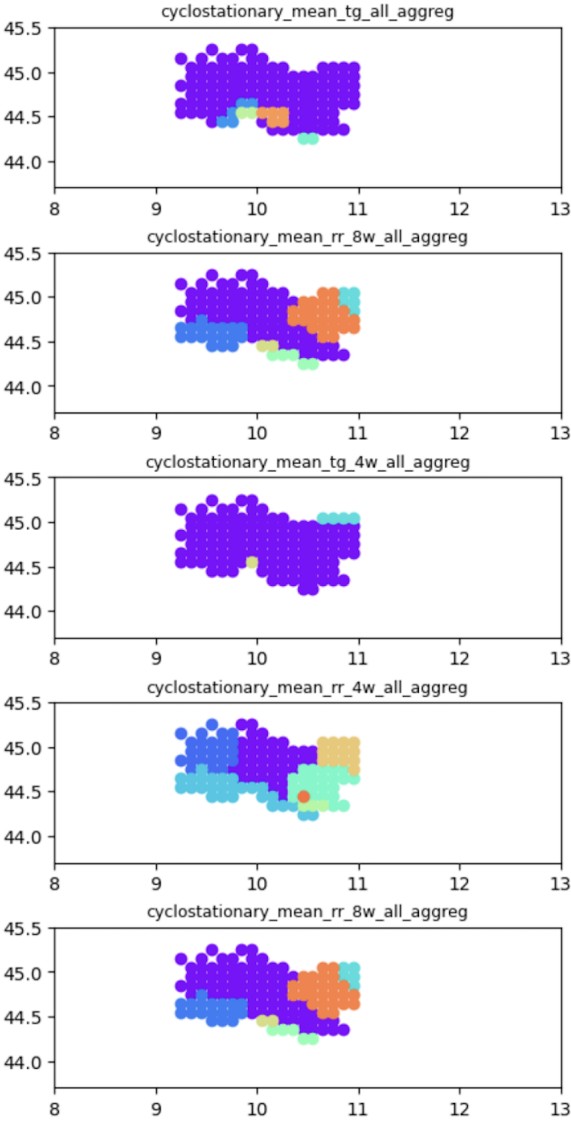

Figure 4: Aggregations on the considered basin using the NonLinCFA algorithm on five different types of variables related to temperature and precipitation, at different location. For each variable, reported in a different plot, points (i.e., features) are represented with the same color if they have been aggregated with their mean by the algorithm.

