# OpenReview forum: "Nonlinear Feature Aggregation: Two Algorithms driven by Theory"
_TMLR — Rejected by TMLR_

### Review · Reviewer_erGW · 2023-08-22

**Summary Of Contributions:**

The paper proposes a feature aggregation framework. The basic idea is to consider the increase of the expected error (measured by MSE or deviance) after the two features are aggregated (typically, taking mean). The proposed algorithm grows a group of the aggregated features iteratively as far as the estimated error increase is less than a given threshold. The analytical evaluation of the expected error increase is provided, and further, as a more general error criterion, the upper bound deviance is also provided. The performance is evaluated by using synthetic and real-world datasets based on the reduced dimension and prediction accuracy.

**Audience:**

Yes

**Claims And Evidence:**

Yes

**Requested Changes:**

- The feature grouping has also been studied in the context of sparse learning, though they are not discussed. The simplest approach is to apply hierarchical clustering beforehand (Park et al., Biostatistics2007). Famous approaches are cluster LASSO (She Electron. J. Statist 2010) and OSCAR (Bondell et al., Biometrics2008) and several follow up studies also exist (eg., Kamkar et al. 2016). The relation should be clarified.

- Algorithm 1 seemingly depends on the index order of features (When the index 1, ..., D is replaced, the result can change). It does not have a significant effect on the result? Further, the feature aggregation can be seen as a clustering problem, and in this sense, the partitioning procedure of Algorithm 1 is somewhat naive (just add a feature into a current group one by one). For example, I guess that a hierarchical clustering like approach might be applicable (iteratively merging closest pairs). The clustering procedure also does not have a significant effect on the results?

- More detailed discussion for approaches combining general dependency measure such HSIC with a linear projection should have been discussed. For example, HSIC-based (e.g., Fukumizu et al., JMLR2004, Masaeli, et al., ICML2010, Wu et al., NeurIPS2019) and mutual information-based (Suzuki et al., AISTATS2010) dimension reduction should have a close relation. Advantage compared with this type of approaches, which can incorporate nonlinear relation between y and aggregated feature (linearly projected features) unlike the proposed framework, should be revealed.

- Two methods of supervised PCA by (Bair et al 2006) and (Barshan et al 2011) are largely different methods (the former is based on classical PCA, and the later is based on HSIC). I think they should be discussed separately though they are categorized as the same approach in Sec 2.1, and further, in the experiments, which one is used?

- In the proof of theorem 4, the authors use the second order approximation, and then, why the upper bound still can be guaranteed? How the error by the the second order approximation is removed in the final upper bound?

- Why R2 score of the proposed method sometimes decreases with the number of features (such as Fig1(d)).

- What kernel is used in kernel pca and supervised pca (if it is one by Barshan et al 2011)?

Minor questions and comments:
- All the hyper-parameters are selected by the validation performance?

- The author mention that the typical setting of the aggregate function h is (x_1 + x_2) / 2. When x_1 is an already aggregated feature, the effect of x_2 becomes stronger than features aggregated in x_1?

- In Sec6.2 'as weel as' -> 'as well as'

- I do not find the definitions of \hat{w}_1, \hat{w}_2, \hat{w}. (least-squares estimator?)

Reference:

Fukumizu et al., Dimensionality Reduction for Supervised Learning with Reproducing Kernel Hilbert Spaces, JMLR 2004.

Masaeli, et al., From Transformation-Based Dimensionality Reduction to Feature Selection, ICML 2010.

Wu et al., Solving Interpretable Kernel Dimension Reduction, NeurIPS 2019.

Suzuki et al., Sufficient dimension reduction via squared-loss mutual information estimation, AISTATS 2010.

Park et al., Averaged gene expressions for regression, Biostatistics 2006

She, Sparse regression with exact clustering, Electron. J. Statist. 2010

Bondell et al., Simultaneous regression shrinkage, variable selection and clustering of predictors with OSCAR. Biometrics 2008

Kamkar et al., Stabilizing l1-norm prediction models by supervised feature grouping, Journal of Biomedical Informatics 2016

**Strengths And Weaknesses:**

< Strength >

- The overall framework is easy to use, and highly interpretable results can be obtained.

- Deriving the expected error based criterion for the feature aggregation seems novel and interesting.

< Weakness >

- Some related works such as the group regularization are not discussed.

- The feature aggregation procedure of the proposed algorithm is somewhat straightforward by which the quality of resulting grouping is not fully clear.

- The proposed framework is written in such a way that the aggregation function is arbitrary, and this is one of major differences from (Bonetti et al., 2023), but in the end, only the average based aggregation is used.

- Direct evaluation of accuracy of the identified aggregation is not provided. Evaluating accuracy on the group structure identification (how accurately aggregation is identified) is seemingly essential for the proposed framework, but it cannot be seen in the result.

---

> ### Author Response · Authors · 2023-09-15
> **Answer to review submitted by Reviewer erGW (Part 1)**
>
> Thank you for the comments and suggestions in the review. In the following we provide some answers for each of the points raised and the modifications that we will insert in the main paper after all reviews will be available.
>
> ### **Group regularization methods**
> As suggested in the review, we will add a discussion about feature grouping algorithms in the context of sparse learning and we will insert as references (Park et al., Biostatistics2007), (She Electron. J. Statist 2010), (Bondell et al., Biometrics2008) and (eg., Kamkar et al. 2016) in the related works section.
> Indeed, these methods identify groups of variables and assign the same coefficient to variables belonging to the same group, which is similar to aggregating them with their mean. Additionally, some theoretical guarantees resemble the ones proposed in (Bonetti et al., 2023). However, these approaches are designed in a regularization setting, where the main motivation is to regularize a sparse linear problem by adding a penalty on the number of non-zero coefficients and another penalty term that focuses on assigning similar weights to groups of features, to further regularize the supervised learning task. This way, these group regularization methods are designed as multi-objective algorithms that simultaneously try to optimize the regression error, the coefficient sparsity and their similarity. On the contrary, the algorithm proposed in (Bonetti et al., 2023) is specifically designed for dimensionality reduction through feature aggregation, with the final purpose of obtaining a set of reduced features that preserves the majority of the information on the target, without discarding the contribution of each feature but reducing their dimensions in an interpretable way (the mean). In this context, this method is specifically focused in a single task (aggregating two features) providing some guarantees on the mean squared error after the aggregation, with no explicit regularization involved.
>
> ### **Dependence on the ordering and partitioning strategy**
> As described in the review, the algorithms depend on the order of features. In this context, we assume the order of inputs to be randomly chosen, such that systematic biases due to the ordering are avoided. A heuristic procedure to avoid the dependency of the algorithms on the ordering of inputs would be to introduce an internal ordering. Considering for example correlation, one may fistly consider the pair of the two most correlated inputs and test if they exceed the threshold. If so, they could be added to a cluster and substituted with their mean. Iteratively proceeding in this way, until all features have been assigned to a set of the partition, produces an algorithm that becomes independent from the initial ordering of features. This would be a heuristic hierarchical clustering-like approach as suggested in the review, that still allows to aggregate pairs that are not the closest (in terms of correlation) if it is beneficial in terms of MSE.
>
> This choice does not lead to an *optimal* partition in general, which is the one that minimizes the mean squared error. Intuitively, with infinite samples, the MSE is minimized by considering each feature independently. This is confirmed by the asymptotic variance analysis, where a term $n$ shows that, with infinite samples, there is no decrease of variance with the aggregation. However, the identification of the optimal partition with finite samples is combinatorial, since all the possible partitions should be tested (as in any feature selection/aggregation method). The proposed algorithm adds one feature at a time in a cluster, therefore it has no guarantees of optimality. This is in line with classical machine learning approaches such as forward feature selection, that iteratively selects a promising feature, although a combination of other two features may be more informative.
>
> ### **Accuracy of the aggregations**
> As discussed above, since the analysis is focused on iteratively aggregating couples of inputs, the performance of the algorithms in terms of dimensionality reduction should be evaluated in terms of mean squared error. In this context, there exist a partition that leads to the minimum expected mean squared error. However, it would be combinatorial to find it. Therefore, the proposed algorithm identifies a partition in polynomial time which is not guaranteed to be optimal but that fastly aggregates features without much loss of information, letting the user the possibility to regulate the latter with the choice of the hyperparameter $\epsilon$.

---

> ### Author Response · Authors · 2023-09-15
> **Answer to review submitted by Reviewer erGW (Part 2)**
>
> ### **Aggregation function**
> As discussed in the review, the analysis and the subsequent algorithms proposed in the paper are based on a general zero-mean aggregation function. This leads to a broader theoretical analysis w.r.t. only considering the mean as aggregation function. However, as described in Section 2, the main applicative interest is to consider the mean as aggregation function, to preserve interpretability. For this reason, the majority of the experiments performed considers the mean as aggregation function, although some synthetic experiments have also been performed considering the sum of squares to aggregate. Therefore, this does not limits the theoretical analysis, which provides a result applicable with any aggregation function, leaving to the user the choice of considering more complex aggregations. This is also in line with (She, Electron. J. Statist 2010), where a customizable matrix $\mathbf{T}$ represents a customizable sparsity requirement. Additionally, the main novel contribution of the proposed algorithms w.r.t. (Bonetti et al, 2023) is to relax the linearity assumption of the underlying model, allowing the algorithms to be applied in a sound way in non-linear and regression contexts, independently from the aggregation function used.
>
> ### **General dependency measures**
> We will give more space to the approaches combining general dependency measures with linear projection in the *Non-linear dimensionality reduction* paragraph of Section 2.1, in order to clarify the motivation of the proposed algorithms. Indeed, as suggested in the review, these approaches consider general dependency measures such as HSIC and MI, rather than simple correlation, performing a linear projection based, for example, on the optimization of the conditional covariance operator or the squared mutual information. However, in the most general case, these approaches produce a set of reduced features that are a linear combination of possibly all the original features, with different coefficients. This is in contrast with the main motivation of the work in (Bonetti et al., 2023) and its extensions provided in this paper. Indeed, the focus of the proposed algorithms is to identify a partition of the original features to aggregate with a user-defined function, which preserves the interpretability, for example in the sense of (Kovalerchuk et al., 2021)*.
>
> Additionally, the correlation-based criterion considered in the proposed algorithms is an effect of the analysis, that is performed assuming linear regression. Therefore, the proposed algorithms perform, in the most general case, non-linear projections of non-linear features, driven by correlation. As reported in the conclusive section, the consideration of a general non-linear dependency measures such as HSIC and MI is subject to future work. Indeed, a further relaxation of the proposed algorithms would be to consider a broader family of ML models rather than linear regression and generalized (non)-linear models, where such measures are expected to arise from the analysis, replacing correlation.
>
> \* *Boris Kovalerchuk, Muhammad Aurangzeb Ahmad, and Ankur Teredesai. Survey of explainable machine learning with visual and granular methods beyond quasi-explanations. Interpretable artificial intelligence: A perspective of granular computing, pages 217–267, 2021.*

---

> ### Author Response · Authors · 2023-09-15
> **Answer to review submitted by Reviewer erGW (Part 3)**
>
> ### **Other comments**
>
> As suggested, to underline the different nature of the two supervised PCA methods mentioned, we will discuss the algorithm by Barshan et al. together with the methods based on general dependency measure referenced in the review and discussed in the previous point.
>
> The experiments consider the Supervised PCA algorithm by Barshan et al., with linear, polynomial and sigmoidal kernels (the same kernels have been considered for Kernel PCA).
>
> We will explicitely mention in the main paper that the upper bound provided is an approximated upper bound with a second order approximation.
>
> The R2 score associated to the proposed methods is slightly decreasing or it has larger confidence intervals in some synthetic experiments (mainly the ones with huge variance). This may be due to the fact that, in a much noisy experiment setting, a larger number of reduced features does not necessarily improve the model performance, but can lead to more noisy parameters estimate. For this reason, the scores associated to different values of the hyperparameter $\epsilon$ have been tested.
>
> Multiple values for the hyperparameter $\epsilon$ are reported for the proposed algorithms (LinCFA and NonLinCFA), selecting some of them in order to show different number of reduced features and the related performances. For all the benchmark dimensionality reduction methods, at each repetition of the algorithm with different seed, the best validation score and the related number of reduced features has been selected, considering between 1 and 50 reduced features.
>
> As underlined in the review, if at the first iteration we aggregate $x_1,x_2$ producing $\bar{x}=\frac{x_1+x_2}{2}$ and at the second iteration we aggregate $\bar{x},x_3$ producing $\hat{x}=\frac{\bar{x}+x_3}{2}$, the effect of $x_3$ becomes stronger than features aggregated in $\bar{x}$. Therefore, we propose to assign equal weight to each variable after the aggregation (producing in the example $\hat{x}=\frac{x_1+x_2+x_3}{3}$). This choice adheres less with the theoretical analysis, but we believe it is more interpretable, since each reduced feature is a single mean and not an iterative mean of means.
>
> As suggested, we will correct the typo in Section 6.2 and we will explicitely define $\hat{w},\hat{w}_1,\hat{w}_2$ as least squares estimators before they fistly appear in Equation 7.

---

### Review · Reviewer_Fa2z · 2023-10-23

**Summary Of Contributions:**

This paper presents two algorithms for dimension reduction, both extending an existing method that aggregates features in a linear fashion [Bonetti et al., 2023].

Some theoretical results are given.

Some experimental results are provided.

========================================================================

THE FOLLOWING  IS FOR THE REVISION, I.E., AFTER (THE FIRST ROUND OF) REBUTTAL:
========================================================================
I checked the revised paper. I tried to search at least three times
for the rebuttal for my comments and questions.
However, I couldn't find the response to my comments.
Also, reading through the revised paper, I don't think the revision addresses my questions
and  concerns. The interpretability issue still remains. The comparison is still quite limited
in range and depth; for example, for high dimensional data, only one gene expression data was added,
but its sample size is about 800, for which the revision claims that "... it is composed by
a very small number of sample (801)...". Actually, for gene expression data or more generally,
biomedical data, the sample size is often less than 100. Without a (clearly shown) ability
to handle such kind of data, I am not convinced by the utility of the proposed algorithms.
In terms of theoretical study, I found some derivations of the means and variants, but
I couldn't find the insight about such quantities or such derivations.

Taken together, I am not convinced that this paper is of (clearly shown) practical value or
theoretical importance. Thus, I cannot support its publication.

========================================================================
========================================================================

**Audience:**

Yes

**Broader Impact Concerns:**

No concerns.

**Claims And Evidence:**

No

**Requested Changes:**

Please see the Weaknesses part.

**Strengths And Weaknesses:**

Strengths:

- Two algorithms for dimension reduction and aggregation of features in nonlinear fashion, both extending an existing method that aggregates features in a linear fashion [Bonetti et al., 2023].

-Some theoretical results are given.
-Some experimental results are provided.

Weaknesses:
There are some issues or questions that should be clarified:

1. The main concern is about the interpretability.
The authors claim the linear feature aggregation method has an advantage of interpretabililty. The two algorithms presented in this paper are nonlinear extensions of the linear aggregation method;
however, with the use of the non-linear functions in the two algorithms, it is unclear if the interpretability can still be preserved. The authors need to clarify the interpretabily of the two algorithms.

2.  On Page 3, it is unclear what i represents in the notation of \phi_i(X) in the paragraph below Eq. (4).

3. On Page 3, it is stated that "... the proposed algorithms allow to decide the aggregation function that is more meaningful for a specific application,..."
however, the paper does not  show this ability clearly. It remains unclear if and how this is achieved.

4. On Page 4, it criticizes the svPCA since it states that "...but discarding the contribution of many features"
If the features couldn't contribute much to the variance used in PCA,
then it appears to be fine to discard such features.
It seems that the authors criticizes using sparse weights,
by saying that "... but discarding the contribution of many features."
Maybe I missed the point of the authors regarding using sparse weights.
Is there anything wrong with discarding non-contributing features?

5.  For Eq. (10), \Delta_{var}^{n \tend \infty} is not defined very clearly.
If n \tend \infty, then how can you still have (n-1) in the denominator?

6. On Page 11, the authors list some DR methods in the first paragraph of
Subsection 6.2 on Page 11, but it does not seem that the authors compare with these methods in Tables 1 and 2. (I saw some results about these methods in Appendix, but appears to be of limited scope). Also, some other methods for dimension reduction needs to be compared, for example, t-SNE, UMAP, autoencoder, and variational autoencoder.

7. On page 11, what does "a subset of test performances of three datasets ..." mean?

8. Experimental results are too limited.
The authors only used 3 datasets from Kaggle and UCI ML, including Finance,
Bankruptcy, and Parkinson, and 4 data sets from climate dataset, which are more or less similar.

However, more comparisons on datasets with much more dimensions are expected, in particular, cases with high p and small n, such as gene expression datasets.

9. In reporting the performance of the methods, for regression, only the performance for  linear regression (LR) is reported. Since LR may have limited performance, it would be  desirable to consider some nonlinear regression methods, e.g., SVR.

10. The authors  claim that the proposed method can preserve the interpretability of the reduced features, but there is no examples or experimental results illustrating the interpretability of the aggregated features.

Interpretability of the 2 algorithms needs to  be illustrated with experimental results.

---

> ### Author Response · Authors · 2023-11-02
> **Answer to review submitted by Reviewer Fa2z (Part 1)**
>
> Thank you for the comments and suggestions in the review. In the following, we provide some answers for each of the points raised and the modifications we have inserted in the main paper, which we will upload at the beginning of the discussion phase as suggested by the journal guidelines.
>
> 1) As described in the review, the two algorithms presented in this paper are nonlinear extensions of the linear aggregation method in (Bonetti et al, 2023). Although they also consider the possibility of using non-linear aggregation functions, the main novel contribution of the proposed algorithms w.r.t. (Bonetti et al, 2023) is the relaxation of the linearity assumption of the underlying model, allowing the algorithms to be applicable in a sound way in non-linear regression and classification contexts, independently from the aggregation function used.
>
>    Additionally, the analysis and the subsequent algorithms proposed in the paper are based on a general zero-mean aggregation function. This leads to a broader theoretical analysis w.r.t. only considering the mean as an aggregation function, and it gives more flexibility to the algorithms, giving the choice to the user to select the aggregation function that is more suitable for a specific application. However, as described in Section 2, the main applicative interest is still to consider the mean as an aggregation function to preserve the interpretability of the reduced features. Indeed, each reduced feature is the mean of a subset of the original features. For this reason, the majority of the experiments performed considers the mean as an aggregation function, although some synthetic experiments have also been performed considering the sum of squares to aggregate.
>
> 2) On Page 3, $\phi_i(X)$ represents the $i-th$ basis function derived from the $D$ original features. We reformulated the sentence in the paper to try to improve the clarity of the notation.
>
> 3) On Page 3, it is stated that "... the proposed algorithms allow to decide the aggregation function that is more meaningful for a specific application,...". As already mentioned in the first point, the meaning of this sentence is that we consider a generic zero-mean aggregation function in the analysis, allowing the user to select the aggregation function that is more meaningful for a specific application, which, in our focus, is the mean. We reformulated the sentence to underline that the algorithm does not do the selection of the aggregation function, which is considered as an input, as clearly shown in the pseudo-code of Section 4.
>
> 4) On Page 4, the sentence about svPCA "...but discarding the contribution of many features" is not aimed to criticize the use of sparse weights when features are non-contributing. However, in a general regression setting where all features may provide information about the target, sparse PCA can be seen as a variant of PCA that constrains up to $k$ coefficients to be non-null, discarding the other features that may still contain some relevant information. This is in line with what is usually done in feature selection, where the $k$ most relevant features are selected by the algorithms, and the loss of information is balanced by the advantage in terms of interpretability and computational complexity. We refined the sentence in the main paper to underline that it is not an issue in sparse environments, but it may result in a loss of information when all features are informative.
>
> 5) As underlined in the review, the asymptotic variance increase depends on the number of samples $n$. We added a sentence in the main paper that clarifies that the asymptotic analysis is performed to identify an expression of the decrease of variance that is manageable without considering the confidence intervals of each statistical estimator as the sample size increases. In this context, the dependence of the asymptotic increase of variance on the number of samples $n$ underlines that, increasing the number of samples, the variance of the bivariate regression gets smaller (as happens, for example, in the Central Limit Theorem), meaning in this case that we are less prone to aggregate. This is expected since a large number of samples implies less overfitting and more ability to estimate the two coefficients of the bivariate regression. Then, when the number of samples approaches infinity, the asymptotic increase of variance becomes zero, meaning that it is not profitable anymore to aggregate the two features since we have infinite samples and we can estimate the regression coefficients exactly. This is in line with what usually happens with dimensionality reduction and feature selection. Indeed, if we have infinite samples, there is no need to reduce the dimension or select subsets of features to contrast overfitting or the curse of dimensionality.

---

> ### Author Response · Authors · 2023-11-02
> **Answer to review submitted by Reviewer Fa2z (Part 2)**
>
> 6) The algorithms listed in Section 6.2 have all been considered (PCA, LDA, LinCFA, Kernel PCA, Isomap, LLE, Supervised PCA, NCA). Additionally, multiple repetitions of the proposed algorithms (NonLinCFA and GenLinCFA) with different hyperparameters have been considered. The extensive results can be found in Tables 7 and 8 of Appendix D. In the main paper, to avoid the insertion of large tables, the best performing configuration (in terms of the hyperparameter $\epsilon$) of the two proposed algorithms (NonLinCFA and GenLinCFA) are reported for each dataset considered, together with the results related to LinCFA and to the best performing among the other baselines.
>
>    As suggested in the review, we added t-SNE, UMAP, and autoencoders in the comparison without significant changes to the considerations made regarding the performances.
>
> 7) The sentence on page 11, "a subset of test performances of three datasets ..." means that, as already described in the previous point, we only reported the best-performing configuration of NonLinCFA and GenLinCFA, together with the results related to LinCFA and to the best performing among the other eleven baselines considered. We reformulated the sentence to try to improve its clarity.
>
> 8) The datasets from Kaggle and UCI ML have been considered to include some data available from classical ML repositories. In addition, the data from climate are particularly relevant for the setting of the problem and to validate the proposed algorithms with a real-world complex problem related to Earth sciences. Indeed, they already present a significant amount of features (about 1000 meteorological variables related to temperature and precipitation at different locations), and they are designed to predict a satellite target, whose measurements are only weekly and restricted to the last couple of decades. In this setting, it is particularly relevant to reduce the number of features, given the small number of samples available, and to preserve the physical interpretability of the reduced features in order to design a model that is trustable and visualizable by climate experts.
>
>    Additionally, as suggested in the review, to further inspect the performance of the proposed algorithms in comparison with the baselines, a dataset from the UCI ML repository related to gene expression has been added, considering a small number of samples (801) and a huge number of features (19966). Even in this case, the proposed algorithms are well-performing w.r.t. the considered baselines.
>
> 9) In the paper, we considered linear and logistic regression since it is in line with the theoretical analysis performed. However, the proposed algorithms are preprocessing approaches that do not rely on a specific model choice in practice. Therefore, as suggested in the review, we added SVR for regression and SVM for classification to further validate the proposed approaches. The results obtained, reported in Tables 9 and 10 of Appendix D, are in line with the ones obtained with linear supervised approaches.
>
> 10) As already discussed in the previous points and throughout the paper, the interpretability of the reduced features mainly resides in the choice of the basis functions and the aggregation function. In the experiments, the original features and the mean are considered, leading to interpretable reduced features since they are means of disjoint groups of original features.
>
>     Additionally, considering the climate example with temperature and precipitation measurements at different locations, reduced interpretable features can be shown in practice. As an example, we added Figure 4 in Appendix D. Considering five different features related to temperature and precipitation, we applied the NonLinCFA algorithm, constraining the algorithm to aggregate features only if they are at neighboring locations and they represent the same variable. In the figure is reported, represented with the same color, each set of variables that is aggregated with the mean by the algorithm. This is clearly interpretable since the resulting features are averages of the original features, which are obtained by partitioning the geographical area of interest into sub-regions and averaging on them.

---

> > ### Author Response · Authors · 2023-12-22
> > **About the visibility of the Answers**
> >
> > Dear Reviewer, thank you for the new comment, we have published our answers on the 2nd of November. We have now edited them since probably there were some issue with the visibility of our answer. We hope that now they are visible to you and that they clarify your concerns.

---

### Review · Reviewer_qGVi · 2023-11-07

**Summary Of Contributions:**

The authors propose algorithms for dimension reduction via non-/linear feature selection that have high prediction accuracy while maintaining feature interpretability. They claim that their techniques are grounded in theory and provide empirical efficacy of their algorithms by presenting some results on datasets from UCI repository and climate domain.

**Audience:**

Yes

**Claims And Evidence:**

No

**Requested Changes:**

Please address all the weaknesses

**Strengths And Weaknesses:**

Strengths
* The work tackles a key issue of how to effectively account for non-linear relationships between the features and the target.
* They provide some theoretical justification for their algorithm design.
* The experiments show that the quality of the downstream prediction task after applying their feature aggregation techniques are sometimes better and usually at par with previous state of the art while maintaining interpretability.

Weakness (Theory part):
* The theoretical discussion is surprisingly hard to read and understand.
* It is unclear why only study single and pairwise feature relationships. Do higher order relations (ie triples/quadruples/etc) contain no additional information? A discussion on that would be beneficial.
* Several quantities are not properly defined, eg. what is \Delta (Eq. 10), b’’ (Eq. 15), etc.
* There are proofs missing for Thm. 1 & 2
* The finite sample results, eg Thm. 5 (in the Appendix), are not presented in a readable way (partly because \Delta is not properly defined).

Weakness (Algorithms part):
* The biggest weakness of the presented algorithm is that there seems to be no control over the target dimension. Eg. the user cannot indicate that they need a dimension reduction to exactly say d=10 dimensions. Instead, the user has to make do with whatever reduced dimension the algorithm returns.

Weakness (Experiments part):
* The main claim of the paper is that feature interpretability is maintained after dimension reduction, however the key real-world experiments on this are missing.
* There is no “scale” to compare the details. For example, it is unclear what does attaining an accuracy of 0.75 mean? Is it a good number or a bad number? Perhaps including some experiments/discussion on the maximum possible accuracy attainable (perhaps when no dimension reduction is performed) may be informative.

---

> ### Author Response · Authors · 2023-11-14
> **Answer to review submitted by Reviewer qGVi (Part 1)**
>
> Thank you for the comments and suggestions in the review. In the following, we provide some answers for each of the points raised and the modifications we have inserted in the main paper, which we will upload shortly.
>
> **Theory part**
>
> * *About the clarity of the theoretical discussion.* We structured the paper to make the claims and theoretical discussions as clear as possible. In particular, in the *Contributions* paragraph, we firstly overview the structure of the paper and the algorithmic analysis that can be found. Then, in Section 2, the setting is formalized, and the paragraph called *Dimensionality Reduction via Aggregation* firstly describes the main idea of the paper, trying to ponder between the necessary formalism and the intuition. We have further elaborated this paragraph, following also the suggestions of the other Reviewers, in order to make it as clear as possible. Then, in Section 3, we present the methodology followed to derive the two proposed algorithms, formally introducing the structure of the analysis but still commenting on the terms and quantities that we consider. Finally, Section 4.1 and Section 5.1 report the main results of the theoretical analysis, where each theorem presented is accompanied by remarks that describe the intuitive interpretation that we give to the results. Additionally, to make the paper as clear as possible, many intermediate results and additional analysis that do not directly converge into the proposed algorithms are referred to the Appendix, including the majority of the proofs.
>
> * *About the study of pairwise feature relationships.* The main idea behind the theoretical analysis is to quantify if it is beneficial to aggregate two features, given the fact that the underline data generation process depends in principle on a huge set of $D$ features. As highlighted in the review, this choice does not lead to an optimal partition in general, since we should consider all possible aggregation from two to $D$ dimensions. However, this would be combinatorial, since all the possible partitions should be tested (as in classical feature selection methods). Therefore, the proposed algorithms add one feature at a time in a cluster considering a pairwise feature relationship, in line with classical machine learning approaches such as forward feature selection, that iteratively selects a promising feature, although a combination of other features may be more informative.
>
> * We have not explicitly defined some quantities that can be straightforwardly deduced from the context (e.g., $b''(\cdot)$ as the second derivative of $b(\cdot)$, or $\Delta$ as the difference of two quantities). However, we have now revised the paper, explicitly mentioning the meaning of these quantities.
>
> * Theorem 1 and Theorem 2 are proved in Appendix A, as referenced at the beginning of Section 4.1 of the main paper.
>
> * The finite-samples results, such as Theorem 5, are reported in the Appendix since they are more convolute expressions w.r.t. the asymptotic counterparts, which already allow to draw some interesting conclusions in our view. In general, we indicate $\Delta$ as a difference between two quantities, as should be clear from the statement of the theorems. However, we explicitly mentioned this in the revised version of the paper, to improve clarity.

---

> > ### Author Response · Authors · 2023-11-14
> > **Answer to review submitted by Reviewer qGVi (Part 2)**
> >
> > **Algorithmic part**
> > * The algorithms presented are constructed to iteratively aggregate couples of features depending on a hyperparameter $\epsilon$, which regulates the propensity to aggregate. As discussed in Section 4.2 and 5.2, when $\epsilon$ is large the algorithm is prone to aggregate, while setting it to small values makes the algorithm less prone to aggregate. Therefore, although it is not possible to set a desired number of reduced features, it is still possible to vary this hyperparameter and obtain different reduced dimensions. A possible variation that allows the user to set a desired number of reduced features would be to stop the algorithm when the desired number of features is reached, without further aggregations. However, we did not propose this variation in the main paper since we consider this to be less compliant with the theoretical analysis and with our presentation of the methodology.
> >
> > **Experiments part**
> >
> > * The interpretability of the reduced features mainly resides in the choice of the basis functions and the aggregation function. In all the real-world experiments performed, the original features are considered as inputs and the mean as aggregation function, leading to interpretable reduced features since they are means of disjoint groups of original features.
> >
> >     Additionally, considering the climate example with temperature and precipitation measurements at different locations, reduced interpretable features can be shown in practice. As an example, we added Figure 4 in Appendix D. Considering five different features related to temperature and precipitation, we applied the NonLinCFA algorithm, constraining the algorithm to aggregate features only if they are at neighboring locations and they represent the same variable. In the figure is reported, represented with the same color, each set of variables that is aggregated with the mean by the algorithm. This is clearly interpretable since the resulting features are averages of the original features, which are obtained by partitioning the geographical area of interest into sub-regions and averaging on them.
> >
> > * As suggested, in Table 1 and 2 of the main paper and in the full results of Table 7,8,9,10 of Appendix D, we added one raw (called *Full*), where the full regression score is reported, in order to have an additional benchmark of the regression performances considering all features, without any aggregation or reduction.
> >
> >     As requested by other Reviewers, we also added a Gene Expression dataset with a large number of features, additional dimensionality reduction baselines (UMAP, t-SNE, Autoencoders), and SVM for regression and classification.

---

### Author Response · Authors · 2023-11-15
**New version of the paper**

We have uploaded a new version of the paper, which contains some modifications and additional experiments, as suggested in the reviews. Details can be found in the answers we provided to each Review. The changes and additions are written in blue, to facilitate the tracking of the modifications.

---

### Decision · Action_Editor_dtnE · 2024-01-28

**Recommendation:** Reject

**Comment:**

The paper provided a valuable extension of Bonetti et al.'s (2023) linear feature aggregation method by addressing the critical challenge of effectively capturing non-linear relationships between features and the target variable. However, two of the final reviewers expressed strong reservations regarding the paper's acceptance due to concerns about the limited empirical comparisons with existing methods and insufficiently supported theoretical claims. It appears that significant revisions are necessary before I can recommend its acceptance in its current form

**Audience:**

The paper proposed some extension of the linear feature aggregation method (Bonetti et al., 2023) for high-dimensional data which may be of interest to the TMLR community.

**Claims And Evidence:**

As reviewers pointed out, the theoretical claims are not well supported.  The comparison with existing methods could be more extensive.

**Resubmission Of Major Revision:**

The authors may consider submitting a major revision at a later time.